# Fine-grained Generalisation Analysis of Inductive Matrix Completion

**Antoine Ledent**
TU Kaiserslautern
ledent@cs.uni-kl.de

**Rodrigo Alves**
TU Kaiserslautern
alves@cs.uni-kl.de

**Yunwen Lei**
University of Birmingham
y.lei@bham.ac.uk

**Marius Kloft**
TU Kaiserslautern
kloft@cs.uni-kl.de

## Abstract

In this paper, we bridge the gap between the state-of-the-art theoretical results for matrix completion with the nuclear norm and their equivalent in *inductive matrix completion*: (1) In the distribution-free setting, we prove sample complexity bounds improving the previously best rate of $rd^2$ to $d^{3/2}\sqrt{r}\log(d)$, where $d$ is the dimension of the side information and $r$ is the rank. (2) We introduce the (smoothed) *adjusted trace-norm minimization* strategy, an inductive analogue of the weighted trace norm, for which we show guarantees of the order $O(dr\log(d))$ under arbitrary sampling. In the inductive case, a similar rate was previously achieved only under uniform sampling and for exact recovery. Both our results align with the state of the art in the particular case of standard (non-inductive) matrix completion, where they are known to be tight up to log terms. Experiments further confirm that our strategy outperforms standard inductive matrix completion on various synthetic datasets and real problems, justifying its place as an important tool in the arsenal of methods for matrix completion using side information.

## 1 Introduction

Matrix completion (MC) is the machine learning problem of recovering the missing entries of a partially observed matrix. It is the go-to approach in various application domains such as recommender systems [1, 2] and social network analysis [3, 4, 5]. The SoftImpute algorithm [6, 7] is among the most popular MC methods. It solves the following convex problem encouraging low-rank solutions:

$$\min_{Z\in\mathbb{R}^{m\times n}} \frac{1}{2}\|P_\Omega(Z - G)\|_{\mathrm{Fr}}^2 + \lambda\|Z\|_*, \tag{1}$$

where $P_\Omega$ denotes the projection on the set $\Omega$ of observed entries, $G$ is the ground truth matrix, and $\|\cdot\|_*$ denotes the *nuclear norm* (i.e., the sum of the matrix's singular values).

Besides the incomplete matrix, additional information may be available in applications such as movie recommendation or drug interaction prediction [8, 9, 10, 11]. For instance in movie recommendation, one may have access to the movies' genres, their synopsis, the gender and occupation of the users, or a friendship network between the users. *Inductive matrix completion* (IMC) [11, 12, 13, 14] exploits such side information. It assumes that the side information is summarized in matrices $X \in \mathbb{R}^{m\times d_1}$ and $Y \in \mathbb{R}^{n\times d_2}$, with the row vectors representing the users and items, respectively. IMC then optimizes the following objective function

$$\min_{M\in\mathbb{R}^{d_1\times d_2}} \frac{1}{2}\|P_\Omega(XMY^\top - G)\|_{\mathrm{Fr}}^2 + \lambda\|M\|_*. \tag{2}$$

This model has been used in many domains also beyond movie recommendation [8, 10, 15].

In this paper, we contribute to a better theoretical understanding of IMC and related methods in the approximate recovery case. In this setting we obtain guarantees in terms of a bound on the expected

35th Conference on Neural Information Processing Systems (NeurIPS 2021).

loss which decreases with the number of samples. Our best results concern the *distribution-free* case, meaning that our bounds are valid for any sampling distribution. This is in sharp contrast to the vast areas of literature where one assumes the distribution is uniform [16, 17, 18]. Our analysis leads to substantial gains compared to the state of the art results [19, 20, 21], as we obtain near optimal bounds in situations where the state of the art bounds are vacuous, as is explained below.

Although for uniform sampling, near-tight exact recovery bounds of $O(rd \log(d) \log(n))$ exist[1] for IMC [16, 17], the approximate recovery case (especially in a *distribution-free* setting) is far less understood. The state-of-the-art distribution-free results for IMC were proved in [19, 20] (and in [21] for a kernel formulation of IMC) and, expressed in terms of generalisation error bounds, scale as

$$O\left(\mathbf{x}\mathbf{y}\mathcal{M}\sqrt{1/N}\right),\tag{3}$$

where $\mathbf{x} := \|X^\top\|_{2,\infty} = \max_u \|X_{\bullet,u}\|^2$ is the maximum norm of a left side information vector (row of $X$), $N$ is the number of available samples, and $\mathbf{y} := \|Y^\top\|_{2,\infty} = \max_v \|Y_{\bullet,v}\|^2$ is the maximum norm of a right side information vector (row of $Y$). This implies that reaching a given loss threshold $\epsilon$ requires $O(\mathbf{x}^2\mathbf{y}^2\mathcal{M}^2/\epsilon^2)$ entries, where $\mathcal{M}$ is a bound on the nuclear norm of $M$. In this case, we say that the 'sample complexity' is $O(\mathbf{x}^2\mathbf{y}^2\mathcal{M}^2)$. To understand how those bounds scale with the matrix dimensions, consider the simple case where $X$ and $Y$ are made up of blocks of identity matrices. In that case, we have $\mathbf{x} = \mathbf{y} = 1$, yielding a sample complexity of $O(\mathcal{M}^2)$. Since $\|M\|_*^2 \sim d^2 r$, this yields a bound of order $rd^2$.

Such bounds have a remarkable property: they do not depend on the size $n$ of the matrix and instead depend only on the size $d$ of the side information. This means that they capture the fact that valuable information can be extracted even for users and items for which no ratings are observed. On the other hand, these bounds have a strong dependence on the size $d$ of the side information. As an illustration, consider that they are vacuous when $X = I$ and $Y = I$, since the required number of entries $O(rd^2) = O(rn^2)$ then grows faster than the total number of entries $n^2$. This is despite the fact that in that situation, distribution-free bounds for standard matrix completion yield a sample complexity of $O(n^{3/2}\sqrt{r})$ for the standard regulariser [22] and $O(nr \log(n))$ for a modified regulariser (the smoothed weighted trace norm from [23]). Thus, these existing distribution-free IMC bounds are very far from tight. In fact, they are only meaningful when the size of the side information is *negligible* compared to the general scale of the problem, which is a significant limitation in terms of the elegance of the theory (mismatch with MC bounds, separate proof techniques for separate regimes) and in practice (real-life side information could be very high-dimensional, especially if it is extracted from a neural network [24] or from a wide variety of different sources). To reinforce that point, note that any side information with a strong cluster structure[2] would exhibit similar failings to the above mentioned identity side information case.

In this work, we bridge the gap between the state-of-the art in matrix completion and inductive matrix completion with the trace norm by providing distribution-free bounds for IMC which combine both of the following advantages: (1) a lack of dependence in the size of the original matrices, and (2) a more refined dependence on the size of the side information: the dependence on $d$ in our bounds is almost the same as the dependence on $n$ (the size of the matrix) for the state-of-the-art MC results. More precisely, our first contribution is to provide a bound of order $O(d^{3/2}\sqrt{r} \log(d))$ for the standard regulariser (2). The proof builds on techniques from [22, 25], but is substantially more involved due to the complicated dependence structure generated by the side information. As our second contribution, we construct analogues of the ideas of [23, 26] for the IMC setting: we begin by showing a bound of order $O(rd \log(d))$ for a class of distributions with certain uniformity assumptions (our "uniform inductive marginals"), and then design a new "adjusted trace norm regulariser" for the problem (2) with similar properties to the weighted trace norm [26, 23] in MC. Instead of simply renormalising rows and columns of $M$ as in previous work, our method requires rescaling the core matrix $M$ along data-dependent orientations that capture interplay between the sampling distribution and the side information matrices $X, Y$.

Our contributions are summarised as follows.

---

[1]with some orthogonality assumptions on the side information

[2]where the users and items are approximately split into 'communities', see also Appendix A

1. We provide distribution-free generalisation bounds for the inductive matrix completion model (2) (assuming a fixed upper bound on the nuclear norm) which scale like $O(d^{3/2}\sqrt{r}\log(d))$ where $r$ is a soft relaxation of the rank.

2. In the case of uniform or approximately uniform sampling, we provide a bound of order $O(rd\log(d))$ for *approximate recovery*.

3. We introduce a modified version of the IMC objective (2), which we refer to as *adjusted trace norm regularsation* (ATR). An empirical version E-ATR is also introduced and both achieve bounds of order $O(rd\log(d))$ in the distribution-free setting.

4. We experimentally demonstrate on synthetic data that our adjusted regulariser outperforms the standard IMC objective (2) in many situations.

5. We incorporate our method into a model involving a non-inductive term and evaluate it on real-life datasets, demonstrating substantially improved performance.

This paper is organized as follows. In Section 2 we review some related work. In Section 3 we introduce our main results. Finally, in Section 4 we present our experimental results.

## 2   Related work

In both MC and IMC, the existing literature consists of several main branches differing in their main assumptions: *exact recovery* versus *approximate recovery* and *uniform sampling* versus *distribution-free bounds*. In *exact recovery*, the matrix is assumed deterministic, and we want to recover its missing entries exactly [17, 16, 27, 28]. In *approximate recovery*, the matrix is assumed noisy, and we want to recover its missing entries only approximately, within some interval around their expectation [19, 20, 21, 18, 29]. Approximate recovery theory is typically expressed in terms of uniform generalisation bounds over a function class using a matrix-norm constraint. Assuming that the entries are sampled from a specific distribution (e.g., uniform), one typically can achieve much faster rates than *distribution-free* theory regardless of the distribution. The typical sample complexity of standard MC under uniform sampling is $O(nr\log^2(n))$ for exact recovery (proved in the series of breakthrough papers [27, 28, 30]) and $O(nr\log(n))$ for approximate recovery [23]. In [31, 32], an improved rate of $nr\log(n)\log(r)$ (for exact recovery) was shown.

The most closely related papers to ours are [22] and [23], which both work only on standard matrix completion *without side information*. In [22], a bound of order $O(n^{3/2}\sqrt{r})$ was obtained in the distribution-free setting for matrix completion with the trace norm, whilst in [23], rates of $O(rn\log(n))$ are shown for sampling with uniform marginals and for a smoothed version of the weighted trace norm regulariser in the distribution-free case. We almost perfectly extend most of the results from both papers to the inductive case, which requires many technical modifications.

Within the IMC framework the closest works are those which also deal with approximate recovery in the non uniform sampling case: [21, 33, 19, 20]. Their bounds, presented in many different contexts, translate to sample complexities of type $O(rd^2)$. Other celebrated works in the theoretical study of IMC include: [16] and [17], which showed rates of order $d^2r^3\log(d)$ and $rd\log(d)\log(n)$ respectively for exact recovery with uniform sampling, together with other important contributions (see appendix). In the case of exact recovery, the rate of $rd\log(d)\log(n)$ was obtained only under the assumption that the side information matrices have orthonormal columns. Some bounds use a completely different regulariser (such as the max norm) to achieve better rates [34, 35] etc. These works also do not involve side information.

In Figures 1 and 2, we summarize state-of-the-art (s.o.t.a.) results in both MC and IMC. Note the problem of exact recovery in the distribution-free case is ill-defined (hence the N/As in our table). In approximate recovery bounds, we omit a factor of $1/\epsilon^2$, where $\epsilon$ is the tolerance threshold in terms of expected loss), as this factor is present in all approximate recovery bounds [3]. In exact recovery bounds, the rate is the order of magnitude of the threshold past which exact recovery occurs with high probability.

---

[3]To our best knowledge, all results show a decline in population expected loss of the order of $\sqrt{1/N}$ where $N$ is the sample size

Table 1: Matrix completion results (trace norm-based only)

| MC | Unif.Sampling | Distr.-free | Weighted version |
|---|---|---|---|
| Exact | $nr \log^2(n)$ ([27, 28, 30]) $nr \log(n) \log(r)$ ([31, 32]) | N/A | N/A |
| Approx. | $nr \log(n)$ ([23, 22]) | $n^{3/2} \sqrt{r}$ ([22]) | $rn \log(n)$ ([23]) |

Table 2: Inductive matrix completion results (trace norm-based only)

| IMC | Unif.Sampling | Distr.-free | Weighted version |
|---|---|---|---|
| Exact | $rd \log(d) \log(n)$ ([17, 18]) $d^2 r^3 \log(d)$ ([16]) | N/A | N/A |
| Approx. (s.o.t.a.) | $rd^2$ ([21, 33, 19]) | $rd^2$ ([21, 33, 19]) | None |
| Approx. (ours) | $rd \log(d)$ (Ours) | $d^{3/2} \sqrt{r} \log(d)$ (Ours) | $rd \log(d)$ (Ours) |

Other related works include (IMCNF) [19, 20], which proposed the following model:

$$\min \quad \frac{1}{2} \sum_{(i,j) \in \Omega} |G_{i,j} - (XMY^\top + Z)_{i,j}|^2 + \lambda_1 \|M\|_* + \lambda_2 \|Z\|_*, \tag{4}$$

where $\lambda_1, \lambda_2$ are regularisation parameters, $G_{i,j}$ denotes the observed entries and the predictors take the form $(XMY^\top + Z)$. This model relies on the cross-validated hyperparameters $\lambda_1, \lambda_2$ to balance the importance of the side information. The authors also showed results based on a combination of a bound for the inductive term $XMY^\top$ and a bound for the non inductive term $Z$. The non inductive terms in the bounds are similar to [22], whilst the bounds for the inductive term are proved from scratch and have also later appeared in a different form in [21, 33] together with a kernel formulation of IMC. In Subsection 4.2 we combine our framework with this strategy to reach competitive results.

In [36], the authors introduce a model consisting of a sum of mutually orthogonal IMC terms together with an explicit optimization strategy in the specific case where the available side information consists in partitions of the users and items into communities. In [37], the authors further extend the model to learn the community membership functions together with the ground truth matrix, based only on the sampled entries. The case of a single IMC term where the side information is in the form of a community partition is useful to develop intuition into the equivalent roles of $d_1, d_2$ in our bounds versus $m, n$ in MC bounds. Whilst generalization bounds were proved in [36] with a similar scaling as our bound from Thm 3.1 (and in particular are better than the state-of-the-art IMC bounds if applied to this situation), they only apply to the specific case of community side information. In this work (Theorem 3.1) we achieve the first IMC bounds which cover the whole range of possible side information matrices $X, Y$, whilst providing the correct scaling (up to log terms) in the case of identity or community side information. Community side information has also been studied in other discrete contexts where individual behaviour is assumed to be a noisy realisation of community side information [38, 39].

Another work is [18] which introduces a joint model that imposes a nuclear norm-based constraint on both $M$ and $XMY^\top$ through a modification of the objective. The authors prove bounds for their method which match the state of the art in IMC [17, 19] and MC [22] when the side information is perfect and useless respectively. The dependence on the side information is better in our case. Further discussion of that paper is included in the appendix. Of course, there are also many other works which propose modified optimization problems for the Recommender Systems task through other rank-sparsity inducing regularisers [35, 34, 40] and even exploiting other ground truth structure besides the low-rank property [41, 42].

## 3   Main results

**Notation:** We observe $N$ entries of a ground truth matrix $G \in \mathbb{R}^{m \times n}$ which are sampled i.i.d (with replacement) through an arbitrary distribution $p$: we draw $(i,j) \in \{1,..,m\} \times \{1,...,n\}$ with probability $p_{i,j}$ where $\sum_{i,j} p_{i,j} = 1$. The sampled entries $\xi^1, \xi^2, \ldots, \xi^N \in \{1, 2, \ldots, m\} \times$

$\{1, 2, \ldots, n\}$ form a multiset $\Omega$: our setting allows for the observations to be noisy with a different noise distribution for each entry, but purely for notation convenience we often treat the issue as if there is no noise when no ambiguity is possible. When written explicitly, the noise is denoted by $\zeta$. For a function $f : \mathbb{R} \to \mathbb{R}$ we will write $\sum_{(i,j)\in\Omega} f(G_{i,j})$ for the sum of the images of the observations, counted as many times as necessary [4]. We assume we are given side information matrices $X \in \mathbb{R}^{m \times d_1}$ and $Y \in \mathbb{R}^{n \times d_2}$. The maximum $L^2$ norm of a row of $X$ (resp. $Y$) is denoted by $\mathbf{x}$ (resp. $\mathbf{y}$). The minimums are denoted by $\underline{x}$ and $\underline{y}$ respectively. The row vectors of $X$ (resp. $Y$) are also written $x_i$ for $i \leqslant m$ (resp. $y_j$ for $j \leqslant n$). For any matrices $A, B$, $A \leqslant B$ means that $B - A$ is positive semi-definite, $\|A\|$ denotes the spectral norm of $A$ and $\|A\|_*$ denotes the nuclear norm of $A$. We have one fixed loss function $l$ used throughout the paper which is both *Lipschitz* with constant $\ell$ and bounded by $b$. For convenience we also frequently write $d$ instead of $\max(d_1, d_2)$. In the appendix, we provide a complete table of notations (Table K.1) that includes all notations introduced throughout the paper.

We now present our results, starting with the distribution-free bound for the standard regulariser, then moving on to the improved bounds under uniform sampling, and finally to our adjusted trace norm regulariser and the theoretical improvements it provides.

### 3.1 Distribution-free guarantees for the standard IMC objective

For a constant $\mathcal{M} \in \mathbb{R}$, we define the function class: $\mathcal{F}_\mathcal{M} = \{XMY^\top : \|M\|_* \leqslant \mathcal{M}\}$, which contains all predictors $XMY^\top$ where $M$ has its spectral norm bounded by $\mathcal{S}$. Our first main result is a uniform generalisation bound for the loss minimiser within this function class. Below we use the shorthand $l(A)$ (resp. $\hat{l}_S(A)$ or even $l_S(A)$) for $\mathbb{E}_{(i,j)\sim p}(l(A_{i,j}, G_{i,j} + \zeta_{i,j}))$ (resp. $\sum_{(i,j)\in\Omega} l(A_{i,j}, G_{i,j} + \zeta_{i,j})/N$), the overall expected (resp. empirical) loss associated to matrix $A \in \mathbb{R}^{m \times n}$. In particular, in the noiseless setting, $\inf_{Z\in\mathcal{F}_\mathcal{M}} l(Z) = 0$ as long as $\|G\|_* \leqslant \mathcal{M}$.

**Theorem 3.1.** *Fix any target matrix $G$ and distribution $p$. Define $\hat{Z}_S = \arg\min(\hat{l}_S(Z) : Z \in \mathcal{F}_\mathcal{M})$. For any $\delta \in (0, 1)$, with probability (w.p.) $\geqslant 1 - \delta$ over the draw of the training set $\Omega$ we have*

$$l(\hat{Z}) \leqslant \inf_{Z\in\mathcal{F}_\mathcal{M}} l(Z) + C\left[\sqrt{\frac{\ell b \mathbf{x}\mathbf{y}\mathcal{M}\sqrt{d}}{N}}\Psi + \frac{b}{\sqrt{N}} + \frac{\mathbf{x}\mathbf{y}\ell\mathcal{M} + \ell}{N}\log(2d)\right] + 4b\sqrt{\frac{\log(2/\delta)}{2N}}, \quad (5)$$

*where $C$ is a universal constant, $b$ is a bound on the loss, $\ell$ is the Lipschitz constant of the loss $l$, and $\Psi = \left[\sqrt{\log(2d)} + \sqrt{\log(N(20\mathcal{M}^2\ell\sqrt{d}[\mathbf{x}^2\mathbf{y}^2]/b + 1)}\right]$ is a logarithmic quantity. Furthermore, in expectation over the training set we have:*

$$l(\hat{Z}) \leqslant \inf_{Z\in\mathcal{F}_\mathcal{M}} l(Z) + C\left[\sqrt{\frac{\ell b \mathbf{x}\mathbf{y}\mathcal{M}\sqrt{d}}{N}}\Psi + \frac{b}{\sqrt{N}} + \frac{\mathbf{x}\mathbf{y}\ell\mathcal{M} + \ell}{N}\log(2d)\right] + 20b\sqrt{\frac{1}{N}}. \quad (6)$$

The proof is provided in Appendix A. Assuming that $\ell, b$ are treated as constants, the above bound on the generalisation gap $l(\hat{Z}) - \inf_{Z\in\mathcal{F}_\mathcal{M}} l(Z)$ scales like

$$O\left(\frac{\mathbf{x}\mathbf{y}\mathcal{M}}{N}\log(d) + \sqrt{\frac{\mathbf{x}\mathbf{y}\mathcal{M}\sqrt{d}}{N}}\left[\sqrt{\log(d)} + \sqrt{\log(N)} + \sqrt{\log(\mathbf{x}\mathbf{y}\mathcal{M})}\right]\right). \quad (7)$$

If we further think of the maximum entry of the core matrix $M$ as bounded by a constant, $\mathcal{M}$ scales like $\sqrt{d_1 d_2 r}$ where $r$ is the rank of $M$. Assuming the rescaling is also set so that $\mathbf{x}, \mathbf{y}$ are constants, the above yields a sample complexity of

$$O\left(\frac{\sqrt{d_1 d_2}\sqrt{dr}\log(d)}{\epsilon^2}\right),$$

---

[4]More rigorously the observations are i.i.d of the form $(\xi^o, \bar{\xi}^o)$ with $\xi^o \in \{1, 2, \ldots, m\} \times \{1, 2, \ldots, n\}$ and $\bar{\xi}^o \in \mathbb{R}$ and write $\sum_{o=1}^N f(\bar{\xi}_o)$ instead of $\sum_{(i,j)\in\Omega} f(G_{i,j})$, and it should be assumed that the "ground truth" values $G$ (are defined so as to) minimize $\mathbb{E}(l(G_\xi, \bar{\xi}))$ for our loss function $l$ over the joint distribution of $\xi, \bar{\xi}$

where $\epsilon$ is the tolerance threshold. Indeed, the $\sqrt{\log(N)}$ term can be treated via the following simple observation: If $N \geqslant \Theta \log(\Theta)$ and $\Theta$ is sufficiently large then

$$\frac{N}{\log(N)} \geqslant \frac{\Theta \log(\Theta)}{\log(\Theta) + \log(\log(\Theta))} \geqslant \frac{\Theta \log(\Theta)}{2 \log(\Theta)} \geqslant \Theta/2.$$

**Remark on the proof technique:** The proof of the result in [22] relies on a lemma of Latala (lemma A.1) from [43] for random matrices *with i.i.d. entries* and an elegant decomposition of the entries into two groups: (1) entries that have been sampled many times, and (2) entries that have not been sampled too often. On group 1, the partial sums of the Rademacher variables concentrate trivially, whilst on group 2, the entries are well spread out and Lemma A.1 limits the spectral norm similarly to the uniform case. The proof is about carefully balancing those two contributions.

In our inductive situation, using the same split can only yield bounds of the type (3) which are well known and vacuous when the side information is of comparable size to the matrix. Our key idea to fix this issue is that instead of distinguishing frequently and less frequently sampled *entries*, we split between high and low energy *orientations* corresponding to pairs $(X_{\bullet,u}, Y_{\bullet,v})$ of *columns* of the side information matrices. To achieve this aim, we use the rotational invariance of the trace operator and equivalently express the Rademacher averages in inductive space ($\mathbb{R}^{d_1 \times d_2}$). However, the entries of the resulting matrix are certainly not independent, which makes it impossible to apply the concentration results from [43]. Instead, we must rely again on the matrix Bernstein inequality F.4. Obtaining a covariance structure that is amenable to application of this result requires performing an iterative procedure involving series of *distribution dependent* rotational transformations of the side information and other estimates at each step.

### 3.2 Generalisation bounds for the trace norm regulariser under a uniformity assumption

We now move to our second main contribution, which is a broad generalisation of most of the results of [23] to the *inductive* case. In this direction, we begin with a result for approximate recovery in inductive matrix completion with the standard nuclear norm regulariser. Although this first result (proved in Appendix B) is original to the best of our knowledge, it is not surprising since a similar result is known in the exact recovery case. However, it is an excellent way to introduce notation which will be necessary in the rest of the paper.

**Proposition 3.1.** *Let us write $\mathcal{F}_{\mathcal{M}}$ for the function class corresponding to matrices of the form $XMY^\top$ with $\|M\|_* \leqslant \mathcal{M}$. Let $M_S = \arg\min_{\|M\|_* \leqslant \mathcal{M}} \sum_{\xi \in \Omega} l((XMY^\top)_\xi, G_\xi + \zeta_\xi)$ be the trained matrix $M$ and $M_* = \arg\min_{\|M\|_* \leqslant \mathcal{M}} \mathbb{E}l((XMY^\top)_\xi, G_\xi + \zeta_\xi)$ be the optimal $M$ when $M$ is restricted by $\|M\|_* \leqslant \mathcal{M}$. Write also $Z_S = XM_SY^\top$ and $Z_* = XM_*Y^\top$.*

*Write $\mathcal{K} := \max\left[\sqrt{d_1 \frac{\|X^\top X\|}{m} \frac{\|Y\|_{\mathrm{Fr}}^2}{n}}, \sqrt{d_2 \frac{\|Y^\top Y\|}{n} \frac{\|X\|_{\mathrm{Fr}}^2}{m}}\right]$. Under uniform sampling, w.p. $\geqslant 1 - \delta$:*

$$l(Z_S) - l(Z_*) \leqslant \frac{8\ell\mathcal{K}\sqrt{rd}(1 + \sqrt{\log(2d)})}{\sqrt{N}} + \frac{12\ell}{N}\mathcal{M}\mathbf{xy}(1 + \log(2d)) + b\sqrt{\frac{\log(2/\delta)}{2N}}, \quad (8)$$

*where $\sqrt{r} = \mathcal{M}/\sqrt{d_1 d_2}$ and $b$ is a bound on the loss. Furthermore, the above result holds under the following more general "uniform inductive marginals" condition (analogous to the "uniform marginals"):*

$$\forall i, \quad \sum_{i,j} p_{i,j} \|y_j\|^2 = \frac{\|Y\|_{\mathrm{Fr}}^2}{mn} \quad and \quad \forall j, \quad \sum_{i,j} p_{i,j} \|x_i\|^2 = \frac{\|X\|_{\mathrm{Fr}}^2}{mn}. \quad (9)$$

**Remarks:** If $\|x_i\|$ and $\|y_j\|$ are constant over $i$ and $j$, then the above conditions (9) reduce to a requirement of uniform marginal probabilities. Note that $\sqrt{r} = (\mathcal{M}/\sqrt{d_1 d_2})$ acts as a soft relaxation of the rank of $\mathcal{M}$ since if $M \in \mathcal{F}_{\mathcal{M}}$ and the entries of $M$ are bounded by 1 then $\mathrm{rank}(M) \leqslant r$. If $X = I$ and $Y = I$, then conditions (9) reduce to the uniform marginals condition from [23].

In particular, we see that in the case of identity side information, we require $O(dr \log(r))$ samples to reach a given accuracy. However, the result above is deeper when the side information is non trivial. Indeed, the quantity $\max(\sqrt{\|X^\top X\|\|Y\|_{\mathrm{Fr}}^2}, \sqrt{\|Y^\top Y\|\|X\|_{\mathrm{Fr}}^2})$, which equals $d = \max(d_1, d_2)$ in the

case of identity (or equal-size community) side information, is sensitive to the relative orientation of the columns of $X$ and $Y$: if the side information $X$ and $Y$ are properly scaled and approximately of rank $\rho$, then this quantity will approach $\rho$. We discuss this in more details in the appendix.

To prove the above result, we will show a slightly more general result below (Prop 3.2). In order to capture the interaction between the side information and the data-distribution, we must define a distribution-dependent inner product $\langle \cdot, \cdot \rangle_l$ (resp. $\langle \cdot, \cdot \rangle_r$) on the column space of $X$ (resp. $Y$):

For two vectors $u^1, u^2 \in \mathbb{R}^m$ (resp. $v^1, v^2 \in \mathbb{R}^n$) we define $\langle u^1, u^2 \rangle_l = \sum_{i=1}^m u_i^1 u_i^2 q_i$ (resp. $\langle v^1, v^2 \rangle_r = \sum_{j=1}^n v_j^1 v_j^2 \kappa_j$) where the $q_i$s and $\kappa_j$s are defined by

$$q_i = \sum_{j=1}^n p_{i,j} \|y_j\|^2 \quad \forall i \leqslant m \qquad \kappa_j = \sum_{i=1}^m p_{i,j} \|x_i\|^2 \quad \forall j \leqslant n. \tag{10}$$

We now define the vector $\sigma^1 \in \mathbb{R}^{d_1}$ (resp. $\sigma^2 \in \mathbb{R}^{d_2}$) as the vector of singular values of the matrix $X$ (resp. $Y$) *with respect to (w.r.t) the inner product* $\langle \cdot, \cdot \rangle_l$ *(resp.* $\langle \cdot, \cdot \rangle_r$). In other words, the entries of $\sigma^1 \in \mathbb{R}^{d_1}$ (resp. $\sigma^2 \in \mathbb{R}^{d_2}$) are the square roots of the eigenvalues of the symmetric matrix $L := X^\top \operatorname{diag}(q) X \in \mathbb{R}^{d_1 \times d_1} = \sum_{i,j} p_{i,j} x_i x_i^\top \|y_j\|^2$ (resp. $R := Y^\top \operatorname{diag}(\kappa) Y = \sum_{i,j} p_{i,j} y_j y_j^\top \|x_i\|^2 \in \mathbb{R}^{d_2 \times d_2}$). We also write $\sigma_*^1 = \max(\sigma^1)$ and $\sigma_*^2 = \max(\sigma^2)$.

**Proposition 3.2.** *With the same notation as in Proposition 3.1, w.p.* $\geqslant 1 - \delta$ *over the draw of the training set* $\Omega$:

$$l(Z_S) - l(Z_*) \leqslant \frac{8\ell}{\sqrt{N}} \mathcal{M} \max(\sigma_*^1, \sigma_*^2)(1 + \sqrt{\log(2d)}) + \frac{12\ell}{N} \mathcal{M} \mathbf{xy}(1 + \log(2d)) + b\sqrt{\frac{\log(2/\delta)}{2N}}.$$

**Remarks:** Note that both $\sigma^1$ and $\sigma^2$ scale as the product of the scaling of $X$ and $Y$. The above result shows that if the distribution is only approximately uniform (sampling probabilities within a given ratio), then the bound is only penalised proportionally to this ratio: for identity side information, $[\sigma_*^1]^2$ is the maximum user (marginal) probability which scales like $1/d_1$ for approximately uniform marginals. Similarly $\sigma_*^2 \sim 1/d_2$, yielding a sample complexity bound of order $dr \log(d)$ as expected.

### 3.3 Proposed adjusted regularisers and notation

In this section, we introduce our adjusted trace norm regulariser and its variants. We first recall that in standard (non-inductive) matrix completion, the weighted trace norm [26, 23] of a matrix $Z$ is defined as $\|\sqrt{D} Z \sqrt{E}\|_*$ where $D \in \mathbb{R}^{m \times m}$ (resp. $E \in \mathbb{R}^{n \times n}$) are diagonal matrices whose diagonal entries contain the marginal row (resp. column) sampling probabilities. Regularising the weighted trace norm instead of the standard trace norm increases performance [26] and leads to better theoretical guarantees. In this work we extend those advantages to the setting where side information is available.

**Notation**: Recall $\Gamma = \sum_{i,j} p_{i,j} \|x_i\|^2 \|y_j\|^2$. Our method is based on a careful distribution-dependent rescaling of the matrix $M$. The idea is that we must look at the principal directions (singular vectors) of the side information matrices, but computed with respect to a distribution-sensitive inner product: when computing inner products of vectors in the column space of $x$, components corresponding to highly users which are more likely to be sampled must be weighted more. Accordingly, we diagonalise the matrix $L = X^\top \operatorname{diag}(q) X$ (resp. $L = Y^\top \operatorname{diag}(\kappa) Y$ from above to write it $P^{-1} D P$ (resp. $Q^{-1} E Q$). We also define empirical versions of those quantities: $\widehat{\Gamma} = \frac{1}{N} \sum_{i,j} h_{i,j} \|x_i\|^2 \|y_j\|^2$ where $h_{i,j}$ is the number of times that entry $(i,j)$ was sampled: $h_{i,j} = \sum_{o=1}^N 1_{\xi_o = (i,j)} = \#(\Omega \cap \{(i,j)\})$; $\hat{q}_i = \sum_j \frac{h_{i,j}}{N} \|y_j\|^2$, $\hat{\kappa}_j = \sum_i \frac{h_{i,j}}{N} \|x_i\|^2$, $\widehat{L} = X^\top \operatorname{diag}(\hat{q}) X$, $\widehat{R} = Y^\top \operatorname{diag}(\hat{\kappa}) Y$, and their diagonalisations $\widehat{P}^{-1} \widehat{D} \widehat{P}$ and $\widehat{Q}^{-1} \widehat{E} \widehat{Q}$. We can now write our predictors

$$XMY^\top = XP^{-1}D^{\frac{1}{2}}[D^{-\frac{1}{2}}PMQ^{-1}E^{-\frac{1}{2}}]E^{\frac{1}{2}}QY^\top = X\widehat{P}^{-1}\widehat{D}^{\frac{1}{2}}[\widehat{D}^{-\frac{1}{2}}M\widehat{E}^{-\frac{1}{2}}]\widehat{E}^{\frac{1}{2}}\widehat{Q}Y^\top. \tag{11}$$

The simplest version of our proposed algorithm is to regularise $[D^{-\frac{1}{2}}PMQ^{-1}E^{-\frac{1}{2}}]$ instead of $M$.

However, some extra technical modifications may be necessary: If some users or items have extremely small sampling probability, the corresponding entries of $D^{-\frac{1}{2}}$ and $E^{-\frac{1}{2}}$ will be very large. To obtain good bounds, we tackle this issue by forcing the entries of $D, \widehat{D}, E, \widehat{E}$ to be bounded below, which

we achieve via smoothing: fixing a parameter $\alpha \in [0,1]$, we define $\tilde{D} = \alpha D + (1-\alpha)\Gamma I/d_1$ and $\tilde{E} = \alpha E + (1-\alpha)\Gamma I/d_2$ where $I$ is the identity matrix. Similarly, $\check{D} = \alpha\hat{D} + (1-\alpha)\hat{\Gamma}I/d_1$ and $\check{E} = \alpha\hat{E} + (1-\alpha)\hat{\Gamma}I/d_2$.

We also define accordingly $M' = D^{\frac{1}{2}}PMQ^{-1}E^{\frac{1}{2}}$; $\widehat{M} = \hat{D}^{\frac{1}{2}}\hat{P}M\hat{Q}^{-1}\hat{E}^{\frac{1}{2}}$; $\widetilde{M} = \tilde{D}^{\frac{1}{2}}PMQ^{-1}\tilde{E}^{\frac{1}{2}}$; and $\widecheck{M} = \check{D}^{\frac{1}{2}}\hat{P}M\hat{Q}^{-1}\check{E}^{\frac{1}{2}}$; as well as similarly $\tilde{X} = XP^{-1}\tilde{D}^{-\frac{1}{2}}$, $X' = XP^{-1}D^{-\frac{1}{2}}$, $\hat{X} = X\hat{P}^{-1}\hat{D}^{-\frac{1}{2}}$, $\check{X} = X\hat{P}^{-1}\check{D}^{-\frac{1}{2}}$, $\tilde{Y} = YP^{-1}\tilde{E}^{-\frac{1}{2}}$, $Y' = YP^{-1}D^{-\frac{1}{2}}$, $\hat{Y} = Y\hat{Q}^{-1}\hat{D}^{-\frac{1}{2}}$, $\check{Y} = Y\hat{Q}^{-1}\check{D}^{-\frac{1}{2}}$. Thus $XMY^\top = X'M'[Y']^\top = \tilde{X}\widetilde{M}\tilde{Y}^\top = \hat{X}\widehat{M}\hat{Y}^\top = \check{X}\widecheck{M}\check{Y}^\top$.

**Proposed models:** We then propose a variety of adjusted regularisation strategies as follows by replacing the regularisation of $M$ by that of $M'$, $\widetilde{M}$, $\widehat{M}$ or $\widecheck{M}$ depending on whether the ground truth distribution is known and whether smoothing is desired. For instance, in the smoothed, empirical case, we will solve the following optimization problem:

$$\min_M \frac{1}{N}\sum_{\xi\in\Omega} l((XMY^\top)_\xi, G_\xi + \zeta_\xi) + \lambda\|\check{D}^{\frac{1}{2}}\hat{P}M\hat{Q}^{-1}\check{E}^{\frac{1}{2}}\|_*. \tag{12}$$

**Remark:** Similarly to the matrix case the smoothing parameter $\alpha$ is set to $\frac{1}{2}$ in all theorem statements [5]. In the experiments, we vary $\alpha$ as indicated.

We will prove results for the empirical risk minimiser belonging to the following function classes:

$$\tilde{\mathcal{F}}_r := \left\{XMY^\top : \|\widetilde{M}\|_* \leqslant \sqrt{r}\Gamma\right\} \qquad \check{\mathcal{F}}_r := \left\{XMY^\top : \|\widecheck{M}\|_* \leqslant \sqrt{r}\hat{\Gamma}\right\}, \tag{13}$$

corresponding to the smoothed and smoothed empirical versions of our algorithm. Note that the factors of $\Gamma$ are added purely for convenience in the final formula, so that we can understand the final formulae in terms of a soft concept of "rank". Indeed we have

$$\|\tilde{D}^{\frac{1}{2}}\|_{\mathrm{Fr}}^2 \leqslant d_1\frac{\Gamma}{2d_1} + \frac{1}{2}\|\sqrt{\mathrm{diag}(q)}X\|_{\mathrm{Fr}}^2 = (1/2)\Gamma + (1/2)\sum_{i,u}X_{i,u}^2\sum_j p_{i,j}\|y_j\|^2 = \Gamma, \tag{14}$$

and similarly $\|\tilde{E}^{\frac{1}{2}}\|^2 \leqslant \Gamma$. Thus if $\|PMQ^{-1}\|_\infty \leqslant 1$ and $\mathrm{rank}(M) \leqslant \rho$, we have $\|\widetilde{M}\|_* \leqslant \sqrt{\rho}\|\widetilde{M}\|_{\mathrm{Fr}} \leqslant \sqrt{\rho}\sqrt{\sum_{u,v}[\tilde{D}_u^{\frac{1}{2}}]^2[\tilde{E}_v^{\frac{1}{2}}]^2[PMQ^{-1}]_{i,j}^2} \leqslant \sqrt{\rho}\|PMQ^{-1}\|_\infty\sqrt{\sum_{u,v}[\tilde{D}_u^{\frac{1}{2}}]^2[\tilde{E}_v^{\frac{1}{2}}]^2} \leqslant \sqrt{\rho}\Gamma$. Similarly, $\|\widecheck{M}\|_* \lesssim \sqrt{\rho}\hat{\Gamma}$ under the condition $\|\hat{P}M\hat{Q}^{-1}\|_\infty \lesssim 1$.

### 3.4 Generalisation bounds for the smoothed adjusted trace norm

Although knowing the distribution is not realistic, it is instructive to see that one can obtain guarantees of order $O(dr\log(d))$ for the function class $\tilde{\mathcal{F}}_r$ as a reasonably straightforward extension of the ideas developed for Proposition 3.2. The proof is provided in Appendix C.

**Proposition 3.3.** *Let $\widetilde{M}_S = \arg\min_{\|\widetilde{M}\|\leqslant\sqrt{r}\Gamma}\sum_{\xi\in\Omega}l((\tilde{X}\widetilde{M}\tilde{Y}^\top)_\xi, G_\xi + \zeta_\xi)$ be the trained matrix $\widetilde{M}$ and $\tilde{Z}_* = \arg\min_{Z\in\tilde{\mathcal{F}}_r}\mathbb{E}l(Z_\xi, G_\xi + \zeta_\xi)$ be the optimal $\tilde{Z}$ when the predictors are restricted to the class $\tilde{\mathcal{F}}_r$. Let also $\tilde{Z}_S = \tilde{X}\widetilde{M}_S\tilde{Y}^\top$. We have w.p. $\geqslant 1 - \delta$:*

$$l(\tilde{Z}_S) - l(\tilde{Z}_*) \leqslant \frac{16\ell\sqrt{\Gamma}\sqrt{r}\sqrt{d}(1 + \sqrt{\log(2d)})}{\sqrt{N}} + \frac{24\ell\mathbf{xy}\sqrt{d_1 d_2 r}(1 + \log(2d))}{N} + b\sqrt{\frac{\log(2/\delta)}{2N}}.$$

### 3.5 Generalisation bounds for the smoothed empirically adjusted trace norm

Below is a more challenging result (proof in Appendix D) which concerns the function class $\check{\mathcal{F}}_r$ corresponding to the empirically smoothed regulariser.

**Theorem 3.2.** *Fix any target matrix $G$ and distribution $p$. Define $\check{Z}_S = \arg\min(\hat{l}_S(Z) : Z \in \check{\mathcal{F}}_r)$ where $\hat{l}_S(Z) = \frac{1}{N}\sum_{\xi\in\Omega}l(Z_\xi, G_\xi + \zeta_\xi)$. For any $\delta \in (0,1)$, w.p. $\geqslant 1 - \delta$*

$$l(\check{Z}) \leqslant \inf_{Z\in\tilde{\mathcal{F}}_r} l(Z) + C\left[\ell\sqrt{r}\gamma(\mathbf{x} + \mathbf{y})^2 + b\right]\sqrt{\frac{\gamma^2 d\log(\frac{d}{\delta})}{N}}, \tag{15}$$

---

[5]It is trivial to extend the proofs to arbitrary $\alpha$ at the cost of a factor of $1/\min(\alpha, 1 - \alpha)$.

*where $\gamma = \frac{\mathbf{x}^2 \mathbf{y}^2}{\underline{x}^2 \underline{y}^2}$ and $C$ is a universal constant. In particular, in expectation over the draw of the training set we have*

$$l(\check{Z}) \leqslant \inf_{Z \in \widetilde{\mathcal{F}}_r} l(Z) + 2C \left[ \ell \sqrt{r} \gamma (\mathbf{x} + \mathbf{y})^2 + b \right] \sqrt{\frac{\gamma^2 d \log(d)}{N}}. \tag{16}$$

The significance of this result is that even in the case of an arbitrary distribution, minimizing the smoothed empirical adjusted nuclear norm $\|\widetilde{M}\|_*$ results in sample complexity bounds of order $dr \log(d)$, meaning that our distribution-dependent transformations have completely removed the negative effects of non-uniformity on the sample complexity. Note the proof requires careful technical variations compared to the proof of the comparable results in [23]. As an example, Lemma E.1 is the equivalent of Lemma 2 in page 8 of the supplementary in [23] (whose proof is far shorter).

### 3.6 Variations on the optimization problems

As in the related literature ([19, 22] etc.), we worked with a *bounded loss*, and expressed our results for the *loss minimizer* within a function class defined by *explicit norm constraints*. However, it is also possible to modify the results (under some boundedness assumptions) to make them apply to lagrangian formulations such as (18) (1), (2). In typical contexts where the entries are known to be bounded, this can even be done with the square loss. As an example, we consider the following immediate corollary of Proposition 3.3 and its global version C.1 (appendix):

**Corollary 3.4.** *Assume that all of the entries of the ground truth are bounded by a constant $C$, and that they are observed without noise. Let $Z_\# = X M_\# Y^\top$ be the solution to the following optimization problem:*

$$\min_M \quad \|\widetilde{D}^{\frac{1}{2}} P M Q^{-1} \widetilde{E}^{\frac{1}{2}}\|_* \qquad \text{subject to} \qquad [XMY^\top]_\xi = G_\xi \quad \forall \xi \in \Omega. \tag{17}$$

*Let $\Phi_C(x) = \mathrm{sign}(x) \min(|x|, C)$. For any $\ell$-Lipschitz loss $l$, we have (with probability $\geqslant 1 - \delta$)*

$$l(\Phi_C(Z_\#)) \leqslant \frac{8 \ell \sqrt{\Gamma} \sqrt{d r_G} (1 + \sqrt{\log(2d)})}{\sqrt{N}} + \frac{12 \ell \mathbf{x} \mathbf{y} \sqrt{d_1 d_2 r_G} (1 + \log(2d))}{N} + 2C\ell \sqrt{\frac{\log(2/\delta)}{2N}}.$$

*where $r_G$ is the smallest $r$ such that the ground truth $G$ satisfies $G \in \widetilde{\mathcal{F}}_r$.*

A further result which applies in the presence of noise is provided in Appendix H.

## 4 Experimental verification

In this section, we experimentally validate the advantages of our adjusted regularisation strategies described in Subsection 3.3. In all experiments, we work with the square loss.

### 4.1 Experiments on synthetic data

We construct square data matrices in $\mathbb{R}^{n \times n}$ with a given rank $r \leqslant d$ for several combinations of $n, d, r$. We provide each model with $d$-dimensional side information spanning the row and column spaces. The sampling distribution is a power-type law depending on $\Lambda$ such that $\Lambda = 0$ yields uniform sampling (details in appendix). We compare three approaches: (1) Standard inductive matrix completion with the side information matrices $X, Y$ (IMC) (2) Our smoothed adjusted regulariser $\lambda \|\widetilde{M}\|$ (for several values of $\alpha$) (ATR)[6]; and finally (3) our smoothed empirically adjusted regulariser $\lambda \|\widetilde{M}\|$ (for several values of $\alpha$) (E-ATR). For each $n \in \{100, 200\}$ we evaluate the following $d, r$ combinations: $(30, 4)$, $(50, 6)$ and $(80, 10)$. In order to study a meaningful data-sparsity regime, in each case we sampled $dr\omega$ entries where $\omega \in \{1, 2, 3, 4, 5\}$. We show the most representative results here. More comprehensive results are provided in the supplementary material.

We observe that our methods outperform standard inductive matrix completion by significant margins in many regimes, even in the case of uniform sampling. Furthermore, the empirical version of our model actually often performs better than the exact one, which matches the observations made in [23] in the case of standard matrix completion. More detailed results are reported in the appendix.

---

[6]Note that in this synthetic context, it is actually possible to compute $\widetilde{M}$ since the distribution is known.

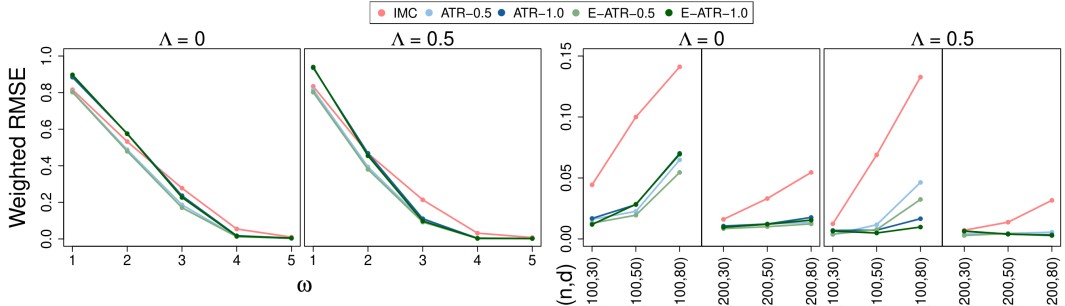

Figure 1: Left: performance as a function of the data sparsity parameter $\omega$ for $n, d, r = 200, 80, 10$. Right: Performance on different $n, d, r$ combinations for $\omega = 4$. Legend: parameter to the right is $\alpha$.

Table 3: Results of real-world datasets (RMSE)

|  | SoftImpute [6] | IMCNF [19] | E-ATR-0.5 | E-ATR-0.75 | E-ATR-1.0 |
|---|---|---|---|---|---|
| **Douban** | 0.9582 | 0.8197 | 0.7691 | **0.7614** | 0.8779 |
| **LastFM** | 2.4109 | 1.7612 | **1.6159** | 1.6943 | 2.3371 |
| **MovieLens** | 0.9280 | 0.9252 | **0.9056** | 0.9139 | 0.9262 |

## 4.2 Real data experiments

We evaluate the performance of our model on three real life datasets: Douban, LastFM and MovieLens (further described in the supplementary). In real data we work with the following adjusted version of the model in [19]:

$$\min_{M,Z} \frac{1}{N} \sum_{(i,j)\in\Omega} l(XMY^\top + Z, G_{i,j} + \zeta_{i,j}) + \lambda_1 \|\breve{D}^{\frac{1}{2}} \widehat{P} M \widehat{Q}^{-1} \breve{E}^{\frac{1}{2}}\|_* + \lambda_2 \|\breve{D}_I^{\frac{1}{2}} Z \breve{E}_I^{\frac{1}{2}}\|_* \quad (18)$$

where $\breve{D}, \breve{E}$ are defined as above based on the side information matrices $X, Y$, and $\breve{D}_I, \breve{E}_I$ are defined as $\breve{D}, \breve{E}$ except based on the side information matrices $(I, I)$. In particular, $\|\breve{D}_I^{1/2} Z \breve{E}_I^{1/2}\|_* = \|\breve{Z}\|_*$ is the smoothed weighted trace norm of $Z$ in the sense of [23]. We report results in Table 3 and note our method outperforms both SoftImpute and IMCNF, especially with appropriate smoothing.

## 5 Conclusion

In this paper, we have provided the first distribution-free bounds for approximate recovery in inductive matrix completion with the trace norm with the following two desirable properties: (1) being non vacuous for identity or community side information and (2) being completely independent of the size of the matrix. We further presented an adjusted regularisation strategy which relies on a careful rescaling along distribution-dependent directions that captures the interaction between the side information matrices and the sampling distribution. Our bounds, which concern both the standard regulariser (rate $O(d^{3/2}\sqrt{r}\log(d))$) and our adjusted version (rate $O(dr\log(d))$) are almost exactly what one would obtain by replacing the size of the matrix with the size of the side information in the standard matrix completion bound. Thus, we have bridged the large gap between the theoretical guarantees for matrix completion and inductive matrix completion.

## Broader impact

The work in this paper is theoretical and without any foreseeable significant societal impact.

## Acknowledgements

The authors acknowledge support by the German Research Foundation (DFG) awards KL 2698/2-1 and KL 2698/5-1, as well as by the Federal Ministry of Science and Education (BMBF) awards 031L0023A, 01IS18051A, and 031B0770E.

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
