# Supplementary Material for "Fine-grained Generalisation Analysis of Inductive Matrix Completion"

**Antoine Ledent**
TU Kaiserslautern
ledent@cs.uni-kl.de

**Rodrigo Alves**
TU Kaiserslautern
alves@cs.uni-kl.de

**Yunwen Lei**
University of Birmingham
y.lei@bham.ac.uk

**Marius Kloft**
TU Kaiserslautern
kloft@cs.uni-kl.de

## A  Proof of Theorem 3.1

**Preliminary discussion:**
The following lemma from [1] was used in the proof in the non inductive case [2].

**Lemma A.1** (Latała, 2005). *Let $X$ be a random matrix with independent, zero mean entries, we have*

$$\mathbb{E}(\|X\|) \leqslant C_\ell \left( \max_i \sqrt{\sum_j \mathbb{E}(X_{i,j}^2)} + \max_j \sqrt{\sum_i \mathbb{E}(X_{i,j}^2)} + \sqrt[4]{\sum_{i,j} \mathbb{E}(X_{i,j}^4)} \right),$$

*where $C_\ell$ is a universal constant.*

The proof of the result in [2] relies on this Lemma, which applies to random matrices *with i.i.d. entries* and an elegant decomposition of the entries into two groups: (1) entries that have been sampled many times, and (2) entries that have not been sampled too often. On group 1, the partial sums of the Rademacher variables concentrate trivially (as the function is constant there), whilst on group 2, the entries are well spread out and Lemma A.1 limits the spectral norm similarly to the uniform case. The idea of the proof is to carefully tune those two contributions by adjusting the threshold involved in the split.

In our inductive situation, directly using a similar splitting strategy can only yield bounds with non-logarithmic dependence on $n$, or bounds of the type of equation (3) (which are well known and vacuous when the side information is of comparable size to the matrix). To understand the problem intuitively, it is helpful to think of the case of 'community side information', where users and items can be divided into equally-sized groups ('communities') by partition functions $c_U : \{1, 2, \ldots, m\} \to \{1, 2, \ldots, d_1\}$ and $c_I : \{1, 2, \ldots, n\} \to \{1, 2, \ldots, d_2\}$ respectively, with the rating of $(i, j)$ depending only on the groups $c_U(i)$ and $c_I(j)$ to which $i$ and $j$ belong respectively. If the side information consists in indicator functions of the communities, simply applying known results for standard matrix completion yields distribution-free bounds of order $O(d^{3/2}\sqrt{r})$ (in this case $d = \max(d_1, d_2)$ will be equal to the max number of communities), whilst applying existing IMC results only yields bounds of order $rd^2$.

Comparing the proof techniques in the MC and "IMC with communities" cases with this example in mind, it becomes clear that the split should no longer be into frequently sampled entries and less frequently sampled entries, but into frequently sampled communities and less frequently sampled communities. To generalise this to arbitrary $X, Y$, we must define a concept of "frequently sampled" combinations $(X_{\cdot, u}, Y_{\cdot, v})$ of columns of the side information matrices. In practice this corresponds to a split between entries of $X^\top R_N Y$ (where $[R_N]_{i,j}$ contains the sum of the Rademacher variables corresponding to entry $i, j$) by high or low variance: we use the rotational invariance of the trace operator and equivalently express the Rademacher averages in inductive space. However, the entries of the resulting matrix are certainly not independent, which makes it impossible to apply the

35th Conference on Neural Information Processing Systems (NeurIPS 2021).

concentration results from [1]. Instead, we must rely again on the matrix Bernstein inequality F.4. Obtaining a covariance structure that is amenable to application of this result requires performing an iterative procedure involving series of *distribution dependent* rotational transformations of the side information and other estimates at each step.

*Proof of Theorem 3.1.* The theorem follows immediately from the classic result (Theorem F.1) as well as its variation F.2 together with the Rademacher complexity bound below (Theorem A.1). □

Recall that for any $x_1, \ldots, x_N$ and any function class $\mathcal{F}$ we can define the (data dependent) Rademacher complexity $\widehat{\mathfrak{R}}_{(x_1,\ldots,x_n)}(\mathcal{F})$ as

$$\widehat{\mathfrak{R}}_{(x_1,\ldots,x_n)}(\mathcal{F}) := \mathbb{E}_\sigma \sup_{f \in \mathcal{F}} \frac{1}{N} \sum_{i=1}^{N} \sigma_i f(x_i), \tag{A.1}$$

where the $\sigma_i$'s are i.i.d. Rademacher random variables (i.e. $\mathbb{P}(\sigma_i = 1) = \mathbb{P}(\sigma_i = -1) = 0.5$).

**Theorem A.1.** *Let $X \in \mathbb{R}^{m \times d_1}$ and $Y \in \mathbb{R}^{n \times d_2}$ be side information matrices. Consider the function class*

$$\mathcal{F}_\mathcal{M} := \left\{ XMY^\top \,\middle|\, \|M\|_* \leqslant \mathcal{M} \right\}$$

*We have the following bound on the expected Rademacher complexity of $l \circ \mathcal{F}_\mathcal{M}$:*

$$\mathbb{E}_{x_1,\ldots,x_N} \mathfrak{R}\left(l \circ \mathcal{F}_\mathcal{M}\right) \leqslant b\sqrt{\frac{2\pi}{N}} + \frac{16\mathbf{xy}\ell\mathcal{M} + \ell}{N} \log(2d) + \sqrt{\frac{10\ell b\mathbf{xy}\mathcal{M}\sqrt{d}}{N}}\Psi, \tag{A.2}$$

*where $\Psi = \left[ \sqrt{\log(2d)} + \sqrt{\log(N(20\mathcal{M}^2\ell\sqrt{d}[\mathbf{x}^2\mathbf{y}^2]/b + 1)} \right]$ is a logarithmic quantity.*

*In other words,*

$$\mathbb{E}\,\mathfrak{R}\left(l \circ \mathcal{F}_\mathcal{M}\right) = \widetilde{O}\left( \sqrt{\frac{\ell b\mathbf{xy}\mathcal{M}\sqrt{d}}{N}} + \frac{\ell\mathbf{xy}\mathcal{M}}{N} + \frac{b}{\sqrt{N}} \right) \tag{A.3}$$

Before we proceed with the proof, we need to establish a few lemmas.

**Lemma A.2** (Variation on Lemma 8 in [3])**.** *Let $r \in \mathbb{N}$ and suppose we are given $r$ fixed matrices $E_1, E_2, \ldots, E_r \in \mathbb{R}^{m \times n}$ with the property that $|E_u|_{i,j} \leqslant B$ for all $u, i, j$. Now consider the following function class for a constant $C \in \mathbb{R}^+$:*

$$\mathcal{F}_C := \left\{ \sum_{u=1}^{r} \lambda_u E_u : |\lambda_u| \leqslant C \quad \forall u \leqslant r \right\}. \tag{A.4}$$

*For any $\epsilon > 0$ there exists a cover $\mathcal{C}_\epsilon \subset \mathcal{F}$ with the following two properties:*

1. *For any $Z \in \mathcal{F}$ there exists a $\widetilde{Z} \in \mathcal{C}_\epsilon$ such that for all $(i,j) \in [m] \times [n]$ we have $|Z_{i,j} - \widetilde{Z}_{i,j}| \leqslant \epsilon$*

2.
$$|\mathcal{C}_\epsilon| \leqslant \left[ \frac{2CBr}{\epsilon} + 1 \right]^r \tag{A.5}$$

*Proof.* We consider the following discretised version of $\mathcal{F}$ for an $\epsilon'$ which will be determined later:

$$\mathcal{D}_{\epsilon'} := \left\{ \sum_{u=1}^{r} p_u \epsilon' E_u \,\middle|\, (\forall u) \quad p_u \in \mathbb{Z} \quad \wedge \quad |p_u \epsilon'| \leqslant C \right\} \tag{A.6}$$

Let $Z \in \mathcal{F}$. We can write $Z = \sum_{u=1}^{r} \lambda_u E_u$ for some $\lambda_u$s. Let $\widetilde{Z} = \sum_{u=1}^{r} \text{sign}(\lambda_u) \left\lfloor \frac{|\lambda_u|}{\epsilon'} \right\rfloor \epsilon' E_u$.

Note that $\widetilde{Z} \in \mathcal{D}_{\epsilon'}$. Furthermore, for any $i, j$ we have

$$\left| Z_{i,j} - \widetilde{Z}_{i,j} \right| = \left| \sum_{u=1}^{r} [E_u]_{i,j} \left[ \lambda_u - \text{sign}(\lambda_u) \left\lfloor \frac{|\lambda_u|}{\epsilon'} \right\rfloor \epsilon' \right] \right| \tag{A.7}$$

$$\leqslant B \sum_{u=1}^{r} \epsilon' = Br\epsilon'. \tag{A.8}$$

Thus, setting $\epsilon' = \frac{\epsilon}{Br}$, we obtain that $\mathcal{C}_\epsilon := \mathcal{D}_{\epsilon'}$ is indeed a uniform $\epsilon$-cover of $\mathcal{F}$ w.r.t. to the $L^\infty$ norm (over the whole sample space $[m] \times [n]$).

Finally, it is trivial to calculate that

$$|\mathcal{C}_{\epsilon'}| = \left[ \frac{2C}{\epsilon'} + 1 \right]^r = \left[ \frac{2CBr}{\epsilon} + 1 \right]^r, \tag{A.9}$$

as expected.

$\square$

The following useful result is an immediate consequence of the McDiarmid inequality. A similar result was presented in [4] (cf. Theorem 11 page 469) for the expected Rademacher complexity.

**Lemma A.3.** *For any fixed $x_1, \ldots, x_N$ and any function class $\mathcal{F}$ mapping to $[-1, 1]$ we have with probability $\geqslant 1 - \delta$ over the draw of the Rademacher variables $\sigma_1, \ldots, \sigma_N$,*

$$\left| \sup_{f \in \mathcal{F}} \frac{1}{N} \sum_{i=1}^{N} \sigma_i f(x_i) - \widehat{\mathfrak{R}}_{(x_1, \ldots, x_n)}(\mathcal{F}) \right| \leqslant \sqrt{\frac{2 \log(2/\delta)}{N}}. \tag{A.10}$$

We now present the following result, of great importance to the proof of Theorem A.1, and which may be of independent interest. It may be viewed as a modification of Dudley's Entropy theorem F.6 entertwined with Talagrand's concentration Lemma.

**Proposition A.4.** *Let $\mathcal{F}_1, \mathcal{F}_2 : \mathcal{X} \to \mathbb{R}$ be two function classes, let $l : \mathbb{R}^2 \to [-1, 1]$ be a bounded loss function with Lipschitz constant $\ell$.*

*Assume that the function class $\mathcal{F}_1$ has the property for all $\epsilon$, it has a uniform cover of size $\mathcal{N}(\mathcal{F}_1, \epsilon)$, where $\mathcal{N}(\mathcal{F}_1, \epsilon)$ is some function of $\epsilon$. That is to say, there is a cover $\mathcal{C}(\epsilon)$ of size $\mathcal{N}(\mathcal{F}_1, \epsilon)$ such that for all $f_1 \in \mathcal{F}_1$ there exists $\tilde{f}_1 \in \mathcal{C}(\epsilon)$ such that for all $x \in \mathcal{X}$ we have*

$$\left| f_1(x) - \tilde{f}_1(x) \right| \leqslant \epsilon. \tag{A.11}$$

*Define the function class $\mathcal{F} = \{ f_1 + f_2 | f_1 \in \mathcal{F}, f_2 \in \mathcal{F}_2 \}$.*

*For all $\epsilon$ and for any training set $x_1, \ldots, x_N$, we have the following bound on the (expected) Rademacher complexity of the function class $l \circ \mathcal{F}$:*

$$\widehat{\mathfrak{R}}(l \circ \mathcal{F}) \leqslant \ell\epsilon + 2\ell\widehat{\mathfrak{R}}(\mathcal{F}_2) + \sqrt{\frac{\log(\mathcal{N}(\mathcal{F}_1, \epsilon))}{N}} + \sqrt{\frac{2\pi}{N}}. \tag{A.12}$$

*In particular, the above also holds for the expected Rademacher complexity after taking expectations.*

**Remark:** The requirement on the cover $\mathcal{C}(\epsilon)$ is quite strong: we require that one fixed cover be an $\epsilon$-cover w.r.t. the $l^\infty$ norm for *any* training set. However, this condition can be satisfied when the function class considered is parametric and globally Lipschitz, as is the case in our application of the result to the proof of Theorem A.1.

*Proof.* Fix an $\epsilon > 0$ and let $\mathcal{C}(\epsilon)$ be a uniform $\epsilon$ cover of $\mathcal{F}_1$. By the Lipschitz property we have for any $\sigma = (\sigma_1, \sigma_2 \ldots, \sigma_N)$:

$$\sup_{f \in \mathcal{F}} \frac{1}{N} \sum_{i=1}^{N} \sigma_i l(f(x_i), y_i) \tag{A.13}$$

$$= \sup_{f_1 \in \mathcal{F}} \sup_{f_2 \in \mathcal{F}_2} \frac{1}{N} \sum_{i=1}^{N} \sigma_i l(f(x_i), y_i) \tag{A.14}$$

$$= \sup_{f_1 \in \mathcal{F}} \sup_{f_2 \in \mathcal{F}_2} \frac{1}{N} \sum_{i=1}^{N} \sigma_i l(\tilde{f}_1(x_i) + [f_1 - \tilde{f}_1](x_i) + f_2(x_i), y_i) \tag{A.15}$$

$$\leqslant \sup_{f_1 \in \mathcal{F}} \sup_{f_2 \in \mathcal{F}_2} \frac{1}{N} \sum_{i=1}^{N} \sigma_i l(\tilde{f}_1(x_i) + f_2(x_i), y_i) \tag{A.16}$$

$$+ \sup_{f_1 \in \mathcal{F}} \sup_{f_2 \in \mathcal{F}_2} \frac{1}{N} \sum_{i=1}^{N} \sigma_i \left[ l([f_1 - \tilde{f}_1](x_i) + \tilde{f}_1(x_i) + f_2(x_i), y_i) - l(\tilde{f}_1(x_i) + f_2(x_i), y_i) \right] \tag{A.17}$$

$$\leqslant \sup_{f_1 \in \mathcal{F}} \sup_{f_2 \in \mathcal{F}_2} \frac{1}{N} \sum_{i=1}^{N} \sigma_i l(\tilde{f}_1(x_i) + f_2(x_i), y_i) + \ell\epsilon, \tag{A.18}$$

where at the last line we have used the fact that $l$ is $\ell$-Lipschitz and that $\mathcal{C}(\epsilon)$ is an $L^\infty$ cover for any dataset, including $x_1, x_2, \ldots, x_N$.

Now, observe that for all $\tilde{f}_1 \in \mathcal{C}(\epsilon)$ and for our fixed training set $x_1, \ldots, x_N$ we can apply Lemma A.3 to the function class

$$l_{\tilde{f}_1} \circ \mathcal{F}_2 := \left\{ [l(\tilde{f}_1(x_i) + f_2(x_i)), y_i]_{i=1}^{N} \quad | f_2 \in \mathcal{F}_2 \right\}.$$

Thus, for any $\delta > 0$, we have w.p. $\geqslant 1 - \delta$ over the draw of the Rademacher variables,

$$\left| \sup_{f_2 \in \mathcal{F}_2} \frac{1}{N} \sum_{i=1}^{N} \sigma_i l(\tilde{f}_1(x_i) + f_2(x_i), y_i) - \mathbb{E}_\sigma \sup_{f_2 \in \mathcal{F}_2} \frac{1}{N} \sum_{i=1}^{N} \sigma_i l(\tilde{f}_1(x_i) + f_2(x_i), y_i) \right| \tag{A.19}$$

$$= \left| \sup_{f_2 \in \mathcal{F}_2} \frac{1}{N} \sum_{i=1}^{N} \sigma_i l(\tilde{f}_1(x_i) + f_2(x_i), y_i) - \widehat{\mathfrak{R}}(l_{\tilde{f}_1} \circ \mathcal{F}_2) \right| \tag{A.20}$$

$$\leqslant \sqrt{\frac{2 \log(2/\delta)}{N}} \tag{A.21}$$

where at the second line we have simply defined $\widehat{\mathfrak{R}}(l_{\tilde{f}_1} \circ \mathcal{F}_2) := \mathbb{E}_\sigma \sup_{f_2 \in \mathcal{F}_2} \frac{1}{N} \sum_{i=1}^{N} \sigma_i l(\tilde{f}_1(x_i) + f_2(x_i), y_i)$.

Now, composing inequality (A.21) with a union bound over all possible choices of $\tilde{f}_1 \in \mathcal{C}(\epsilon)$ we have that for all $\delta > 0$, w.p. $\geqslant 1 - \delta$, every $\tilde{f}_1 \in \mathcal{C}(\epsilon)$ satisfies

$$\left| \sup_{f_2 \in \mathcal{F}_2} \frac{1}{N} \sum_{i=1}^{N} \sigma_i l(\tilde{f}_1(x_i) + f_2(x_i), y_i) - \widehat{\mathfrak{R}}(l_{\tilde{f}_1} \circ \mathcal{F}_2) \right| \tag{A.22}$$

$$\leqslant \sqrt{\frac{\log(\mathcal{N}(\mathcal{F}_1, \epsilon)) + 2 \log(2/\delta)}{N}} \tag{A.23}$$

$$\leqslant \sqrt{\frac{2 \log(2/\delta)}{N}} + \sqrt{\frac{\log(\mathcal{N}(\mathcal{F}_1, \epsilon))}{N}}. \tag{A.24}$$

Now, note that for any choice of $\tilde{f}_1$, we can apply the Talagrand contraction Lemma(cf. [4] (Theorem 12 page 469), [5] (corollary 3.17) , [6](Lemma 8 page 1 of supplementary)) to the function class $\mathfrak{R}(l_{\tilde{f}_1} \circ \mathcal{F}_2)$ to obtain (for any $\tilde{f}_1$):

$$\widehat{\mathfrak{R}}(l_{\tilde{f}_1} \circ \mathcal{F}_2) \leqslant 2\ell\widehat{\mathfrak{R}}(\mathcal{F}_2). \tag{A.25}$$

Plugging Equations (A.24) and A.25 back into equation (A.18), we have that w.p. $\geqslant 1 - \delta$,

$$\sup_{f \in \mathcal{F}} \frac{1}{N} \sum_{i=1}^{N} \sigma_i l(f(x_i), y_i) \leqslant \ell\epsilon + 2\ell\widehat{\mathfrak{R}}(\mathcal{F}_2) + \sqrt{\frac{2 \log(2/\delta)}{N}} + \sqrt{\frac{\log(\mathcal{N}(\mathcal{F}_1, \epsilon))}{N}}. \tag{A.26}$$

The only thing left to do is a simple integration over $\delta$: let $X$ denote the random variable

$$X := \sup_{f \in \mathcal{F}} \frac{1}{N} \sum_{i=1}^{N} \sigma_i l(f(x_i), y_i) - \ell\epsilon - 2\ell\widehat{\mathfrak{R}}(\mathcal{F}_2) - \sqrt{\frac{\log(\mathcal{N}(\mathcal{F}_1, \epsilon))}{N}}. \tag{A.27}$$

By equation (A.26) we have for all $\varepsilon > 0$

$$\mathbb{P}(X \geqslant \varepsilon) \leqslant 2 \exp\left(-\frac{\varepsilon^2 N}{2}\right). \tag{A.28}$$

Integrating over $\varepsilon$ we obtain

$$\mathbb{E}(X) \leqslant \int_0^\infty 2 \exp\left(-\frac{\varepsilon^2 N}{2}\right) d\varepsilon \tag{A.29}$$

$$= \frac{2\sqrt{2}}{\sqrt{N}} \int_0^\infty \exp(-\theta^2) d\theta = \sqrt{\frac{2\pi}{N}}. \tag{A.30}$$

Plugging this equation (A.29) back into the definition of $X$ (eq. (A.27)) we obtain:

$$\widehat{\mathfrak{R}}(l \circ \mathcal{F}) \leqslant \ell\epsilon + 2\ell\widehat{\mathfrak{R}}(\mathcal{F}_2) + \sqrt{\frac{\log(\mathcal{N}(\mathcal{F}_1, \epsilon))}{N}} + \sqrt{\frac{2\pi}{N}}, \tag{A.31}$$

as expected.

$\square$

*Proof of Theorem A.1.* Let $\xi_1, \ldots, \xi_n$ be sampled i.i.d from the sampling distribution $\mathcal{D}$ on $\{1, 2, \ldots, m\} \times \{1, 2, \ldots, n\}$. Let $s_1, s_2, \ldots, s_N$ be iid Rademacher random variables. For any element of $\xi \in \{1, 2, \ldots, m\} \times \{1, 2, \ldots, n\}$ we also write $e_\xi$ for the matrix with all entries equal to $0$ except the entry corresponding to $\xi$, which is set to $1$.

Define the Rademacher matrix $R_N := \sum_{o=1}^{N} e_{\xi_o} s_o$. Define also $U = X^\top R_N Y$. This is a random variable.

We begin with the following easy observations:

$$\begin{aligned}
\text{Tr}\left(\mathbb{E}(UU^\top)\right) = \text{Tr}\left(\mathbb{E}(U^\top U)\right) &= \sum_{u=1}^{d_1} \sum_{v=1}^{d_2} \mathbb{E}(U_{u,v}^2) \\
&= \sum_{u,v} \sum_{i,j} p_{i,j}(X_{i,u})^2 (Y_{jv})^2 = N \sum_{i,j} p_{i,j} \|x_i\|^2 \|y_j\|^2 \\
&= N\Gamma. \tag{A.32}
\end{aligned}$$

Note also that for any $M$, $\langle XMY^\top, R_N \rangle = \langle M, U \rangle$.

We will now need to iteratively define a sequence of matrices $U^k, \bar{U}^k, \bar{M}^k, \bar{V}^k \in \mathbb{R}^{d_1 \times d_2}$ and $\mathcal{T}_k$ for $k = 0, 1, \ldots, K$ for some stopping time $K$. The whole construction depends on a real parameter $p > 0$ which will be chosen later. It is important to note that although the construction of $U^k, \bar{U}^k, \bar{M}^k, \bar{V}^k$ also depends on the sampling distribution $\mathcal{D}$, it is a deterministic construction and does not depend on the data (the same is true of $\bar{M}^k$ for a given core matrix $M$).

$\mathcal{T}_k$ is a sequence of reals defined by $\mathcal{T}_k = \mathbb{E}(\|\bar{U}^k\|_{\text{Fr}}^2)$

First, we set $\bar{U}^0 = \bar{V}^0 = U, \bar{M}^0 = M$ (and $\mathcal{T}_0 = N\Gamma$).

Assuming that $\bar{U}^k$ and $\bar{M}^k$ have been defined already, we define the next iteration as follows.

We first obtain an orthogonal matrix $A^k \in \mathbb{R}^{d_1 \times d_1}$ (resp. $B^k \in \mathbb{R}^{d_2 \times d_2}$) which diagonalises $\mathbb{E}(\bar{U}^k(\bar{U}^k)^\top)$ (resp. $\mathbb{E}((\bar{U}^k)^\top \bar{U}^k)$) so that $\mathbb{E}(\bar{U}^k(\bar{U}^k)^\top) = (A^k)^{-1} D_1 A^k$ and $\mathbb{E}((U^k)^\top(U^k)) = (B^k)^{-1} D_2 B^k$ for some diagonal matrices $D_1, D_2$.

Now, we define

$$\bar{V}^{k+1} = A^k \bar{U}^k B^k \tag{A.33}$$

$$\bar{M}^{k+1} = A^k \bar{M}^k B^k. \tag{A.34}$$

Now, by construction of the matrices $A^{k+1}$ and $B^{k+1}$, the matrices $\mathbb{E}((\bar{V}^{k+1})[\bar{V}^{(k+1)}]^\top)$ and $\mathbb{E}([\bar{V}^{(k+1)}]^\top \bar{V}^{k+1})$ are both diagonal. We now split according to two cases: Case 1:

$$\mathbb{V}\mathrm{ar}(\bar{V}_{u,v}^{k+1}) \leqslant p \qquad \forall u, v \tag{A.35}$$

Case 2: equation (A.35) does not hold, i.e. there exists $u_{k+1}, v_{k+1} \in \mathbb{R}^{d_1 \times d_2}$ with $\mathbb{V}\mathrm{ar}(\bar{V}_{u_{k+1},v_{k+1}}^{k+1}) > p$.

In case 1, we end the procedure and set $K = k$. In case 2, we set

$$\bar{U}^{k+1} = \bar{V}^{k+1} - e_{u_{k+1},v_{k+1}} \bar{V}_{u_{k+1},v_{k+1}}^{k+1} \tag{A.36}$$

(i.e. $\bar{U}^{k+1}$ is identical to $\bar{V}^{k+1}$ on all entries except $(u_{k+1}, v_{k+1})$ where it is set to zero.)

The procedure repeats until case 1 occurs. Note that since the only operations on $\bar{M}$ are from equation (A.34) we have that $\bar{M}^k = A^{k-1}A^{k-1}\ldots, A^0 M B^0 B^1 \ldots B^{k-1} = \bar{A}^{k-1} M \bar{B}^{k-1}$ where $\bar{A}^{k-1}$ (resp. $\bar{B}^{k-1}$) denotes the orthogonal matrix $A^{k-1}A^{k-1}\ldots A^0$ (resp. $B^0 B^1 \ldots B^{k-1}$). Finally, we define

$$U^k = \prod_{i=0}^{k-1}[A^i]^{-1} \bar{U}^k \prod_{i=k-1}^{0}[B^i]^{-1} = [\bar{A}^{k-1}]^{-1}\bar{U}^k[\bar{B}^{k-1}]^{-1}. \tag{A.37}$$

Now, observe that by the rotational invariance of the Frobenius norm and the nuclear norms:

$$\|\bar{M}^k\|_* = \|M\|_* \tag{A.38}$$
$$\mathbb{E}(\|\bar{V}^{k+1}\|_{\mathrm{Fr}}^2) = \mathbb{E}(\|\bar{U}^k\|_{\mathrm{Fr}}^2) = \mathbb{E}(\|U^k\|_{\mathrm{Fr}}^2) = \mathcal{T}_k \tag{A.39}$$

and therefore for all $k \leqslant K - 1$:

$$\mathcal{T}_{k+1} = \mathbb{E}(\|U^{k+1}\|_{\mathrm{Fr}}^2) = \mathcal{T}_k - \mathbb{V}\mathrm{ar}(V_{u_{k+1},v_{k+1}}^{k+1}) \leqslant \mathcal{T}_k - p. \tag{A.40}$$

In particular, since $\mathcal{T}_0 = \mathbb{E}(\|U\|_{\mathrm{Fr}}^2) = \Gamma N$ is finite, the procedure must finish in finite time $K$ with

$$K \leqslant \frac{\Gamma N}{p}. \tag{A.41}$$

Now, $U^k$ is of course only the reexpression of $\bar{U}^k$ in the original orthogonal basis: in particular by the rotational invariance of the Frobenius inner product we have

$$\langle M, U^k \rangle = \langle \bar{M}^k, \bar{U}^k \rangle.$$

Further, we can express the recurrence relations (A.36) and (A.33) directly in this original orthogonal basis in terms of transformations on the $U^k$s:

$$U^{k+1} = [\bar{A}^k]^{-1}\bar{U}^{k+1}[\bar{B}^k]^{-1} \tag{A.42}$$

$$= [\bar{A}^k]^{-1}\left[\bar{V}^{k+1} - e_{u_{k+1},v_{k+1}}\bar{V}_{u_{k+1},v_{k+1}}^{k+1}\right][\bar{B}^k]^{-1} \tag{A.43}$$

$$= [\bar{A}^k]^{-1}\left[A^k\bar{U}^k B^k - e_{u_{k+1},v_{k+1}}\langle A^k\bar{U}^k B^k, e_{u_{k+1},v_{k+1}}\rangle\right][\bar{B}^k]^{-1} \tag{A.44}$$

$$= [\bar{A}^k]^{-1}\left[A^k\bar{A}^{k-1}U^k\bar{B}^{k-1}B^k - e_{u_{k+1},v_{k+1}}\langle A^k\bar{U}^k B^k, e_{u_{k+1},v_{k+1}}\rangle\right][\bar{B}^k]^{-1} \tag{A.45}$$

$$= [\bar{A}^k]^{-1}\left[\bar{A}^k U^k \bar{B}^k - e_{u_{k+1},v_{k+1}}\langle \bar{A}^k U^k \bar{B}^k, e_{u_{k+1},v_{k+1}}\rangle\right][\bar{B}^k]^{-1} \tag{A.46}$$

$$= U^k - \langle \bar{A}^k U^k \bar{B}^k, e_{u_{k+1},v_{k+1}}\rangle[\bar{A}^k]^{-1}e_{u_{k+1},v_{k+1}}[\bar{B}^k]^{-1} \tag{A.47}$$

$$= U^k - \langle U^k, [\bar{A}^k]^{-1}e_{u_{k+1},v_{k+1}}[\bar{B}^k]^{-1}\rangle[\bar{A}^k]^{-1}e_{u_{k+1},v_{k+1}}[\bar{B}^k]^{-1} \tag{A.48}$$

$$= U^k - \langle U^k, E_k\rangle E_k, \tag{A.49}$$

where at the second line (A.43) we have used equation (A.36), at the third line (A.44) we have used equation (A.33), at the fourth line (A.45) we have used equation (A.37), at the fifth line (A.46) we

have used equation (A.37) again as well as a simplification via the definitions of $\bar{A}^k$ and $\bar{B}^k$, at the seventh line (A.48) we have used properties of the Frobenius inner product, and at the eighth and last line (A.49) we have defined $E_k = [\bar{A}^k]^{-1} e_{u_{k+1}, v_{k+1}} [\bar{B}^k]^{-1}$. Note again crucially that the $E_k$s are *deterministic* matrices.

Now, we write $\mathcal{P}$ for the (projection) operator $\mathcal{P}_k : \mathbb{R}^{d_1 \times d_2} \to \mathbb{R}^{d_1 \times d_2} : W \mapsto \langle W, E_k \rangle$. Then equation (A.49) can be written

$$U^{k+1} = (I - \mathcal{P}_k) U^k, \tag{A.50}$$

where $I$ denotes the identity operator from $\mathbb{R}^{d_1 \times d_2}$ to itself. Iterating, we obtain for all $k$

$$U^k = \prod_{i=0}^{k-1} (I - \mathcal{P}_i) U. \tag{A.51}$$

Note that both $\mathcal{P}_k$ and $(I - \mathcal{P}_k)$ are self-adjoint. Hence, we can write

$$\langle M, U^k \rangle = \left\langle M, \prod_{i=0}^{k-1} (I - \mathcal{P}_i) U \right\rangle \tag{A.52}$$

$$= \left\langle \prod_{i=k-1}^{0} (I - \mathcal{P}_i) M, U \right\rangle \tag{A.53}$$

$$= \langle M^k, U \rangle, \tag{A.54}$$

where at the last line we have defined $M^k = \prod_{i=k-1}^{0} (I - \mathcal{P}_i) M$.

Now, note that we can write

$$M^k = \prod_{i=k-1}^{0} (I - \mathcal{P}_i) M \tag{A.55}$$

$$= M - \sum_{u=0}^{k-1} \mathcal{P}_u \prod_{i=k-1}^{u+1} (I - \mathcal{P}_i) M \tag{A.56}$$

$$= M - \sum_{u=0}^{k-1} E_u \left\langle E_u, \prod_{i=k-1}^{u+1} (I - \mathcal{P}_i) M \right\rangle \tag{A.57}$$

$$= M - \sum_{u=0}^{k-1} E_u \lambda_u^k(M), \tag{A.58}$$

where we have defined $\lambda_u^k(M) := \left\langle E_u, \prod_{i=k-1}^{u+1} (I - \mathcal{P}_i) M \right\rangle$. Note that $\|E_u\|_{\mathrm{Fr}} = \|E_u\| = 1$ and since each operator $(I - \mathcal{P}_i)$ is a projection and in particular a contraction with respect to the Frobenius norm we have that $\|\prod_{i=k-1}^{u+1} (I - \mathcal{P}_i) M\|_{\mathrm{Fr}} \leqslant \|M\|_{\mathrm{Fr}} \leqslant \|M\|_*$. Hence for any $M$ with $\|M\|_* \leqslant \mathcal{M}$ we have for any $u < k \leqslant K$:

$$\left| \lambda_u^k(M) \right| \leqslant \mathcal{M}. \tag{A.59}$$

We note that by construction, the matrix $\bar{V}^{K+1} = A^k \bar{U}^k B^k = A^k [\bar{A}^{k-1}] U^k \bar{B}^{k-1} B^k$, has the property that $\mathbb{E}((\bar{V}^{K+1})[\bar{V}^{(K+1)}]^\top)$ and $\mathbb{E}([\bar{V}^{(K+1)}]^\top \bar{V}^{K+1})$ are both diagonal, and

$$\mathbb{V}\mathrm{ar}(\bar{V}_{u,v}^{k+1}) \leqslant p \qquad \forall u, v \tag{A.60}$$

Thus, we have

$$\left\| [U^K][U^K]^\top \right\| = \left\| \mathbb{E}((\bar{V}^{K+1})[\bar{V}^{(K+1)}]^\top) \right\| \leqslant p d_2 \leqslant p d, \tag{A.61}$$

$$\left\| [U^K]^\top [U^K] \right\| = \left\| \mathbb{E}([\bar{V}^{(K+1)}]^\top \bar{V}^{K+1}) \right\| \leqslant p d_1 \leqslant p d. \tag{A.62}$$

We now have the tools to proceed with the proof of the equation (A.2).

We define the following function classes:

$$\mathcal{F}_1 := \left\{ \sum_{k=0}^{K-1} \lambda_k X E_k Y^\top \,\middle|\, |\lambda_k| \leqslant \mathcal{M} \right\} \tag{A.63}$$

$$\mathcal{F}_2 := \left\{ X \left[ \prod_{i=K-1}^{0} (I - \mathcal{P}_i) M \right] Y^\top \,\middle|\, \|M\|_* \leqslant \mathcal{M} \right\}. \tag{A.64}$$

By the constructions above and in particular equation (A.59) we have $\mathcal{F} \subset \mathcal{F}_1 + \mathcal{F}_2$. Furthermore, also by the construction of $U^k$ etc., we can bound the Rademacher complexity of $\mathcal{F}_2$:

$$\mathbb{E}_{\xi_1,\dots,\xi_N} (\mathfrak{R}(\mathcal{F}_2)) = \mathbb{E} \sup_{\|M\|_* \leqslant \mathcal{M}} \left\langle X \left[ \prod_{i=K-1}^{0} (I - \mathcal{P}_i) M \right] Y^\top, R_N \right\rangle \tag{A.65}$$

$$= \mathbb{E} \sup_{\|M\|_* \leqslant \mathcal{M}} \left\langle \left[ \prod_{i=K-1}^{0} (I - \mathcal{P}_i) M \right], X^\top R_N Y \right\rangle \tag{A.66}$$

$$= \mathbb{E} \sup_{\|M\|_* \leqslant \mathcal{M}} \left\langle \left[ \prod_{i=K-1}^{0} (I - \mathcal{P}_i) M \right], U \right\rangle \tag{A.67}$$

$$= \mathbb{E} \sup_{\|M\|_* \leqslant \mathcal{M}} \left\langle M, U^K \right\rangle \tag{A.68}$$

$$\leqslant \mathcal{M} \mathbb{E} \left( \|U^K\| \right) \tag{A.69}$$

where as usual $\|\cdot\|$ denotes the spectral norm.

Now, observe that

$$U^K = \prod_{i=0}^{K-1} (I - \mathcal{P}_i) U = \sum_{o=1}^{N} \prod_{i=0}^{K-1} (I - \mathcal{P}_i) X^\top e_{\xi^o} Y \tag{A.70}$$

$$= \sum_{o=1}^{N} s_o \prod_{i=0}^{K-1} (I - \mathcal{P}_i) x_{\xi_1^o} y_{\xi_2^o}^\top, \tag{A.71}$$

which is a sum of i.i.d centred random matrices. Thus we can apply Proposition (F.4) to it. The value of "$M$" in that proposition is clearly bounded by $\mathbf{xy}$ (indeed, for all $i, j$, $\|x_i y_j^\top\|_{\mathrm{Fr}} = \|x_i y_j^\top\| \leqslant \mathbf{xy}$, the operator $\prod_{i=0}^{K-1} (I - \mathcal{P}_i)$ is a contraction with respect to the Frobenius norm, and the spectral norm is certainly bounded by the Frobenius norm). A bound on the value of "$\sigma$" from Proposition (F.4) follows from our iterative construction and in particular from equations (A.61) which ensure that "$\sigma$" is bounded by $\sqrt{pd}$:

$$\sum_{o=1}^{N} \rho_o^2 \leqslant \sqrt{pd}. \tag{A.72}$$

It follows by an application of Proposition (F.4) to equation (A.69) that

$$N \mathbb{E}_{\xi_1,\dots,\xi_N} (\mathfrak{R}(\mathcal{F}_2)) \leqslant \mathcal{M} \mathbb{E} \left( \|U^K\| \right) \tag{A.73}$$

$$\leqslant \sqrt{8/3}(1 + \sqrt{\log(2d)}) \mathcal{M} \sqrt{pd} + \mathcal{M} \frac{8\mathbf{xy}}{3} (1 + \log(2d))). \tag{A.74}$$

On the other hand, a simple application of Lemma A.2 tells us that $\mathcal{F}_1$ admits a uniform $L^\infty$ cover $\mathcal{C}_{1/N}$ (w.r.t. the whole sample space), of granularity $1/N$ with

$$\mathcal{N}_\infty(\mathcal{F}_1, 1/N) = |\mathcal{C}_{1/N}| \leqslant [2N\mathcal{M}\mathbf{xy}K + 1]^K \leqslant [N(2\mathcal{M}\mathbf{xy}K + 1)]^K, \tag{A.75}$$

since the maximum entry of $E_u$ is bounded by $\mathbf{xy}$ for any $u$.

By Proposition A.4 (rescaled taking into account the bound $b$ on the loss function) we have for any training set

$$\widehat{\mathfrak{R}}_N(l \circ \mathcal{F}) \leqslant \ell\epsilon + 2\ell\widehat{\mathfrak{R}}_N(\mathcal{F}_2) + b\sqrt{\frac{\log(\mathcal{F}_\infty(\mathcal{F}_1, 1/N))}{N}} + b\sqrt{\frac{2\pi}{N}}. \tag{A.76}$$

Taking expectations with respect to the training set on both sides and then applying equation (A.75) and (A.74) we obtain:

$$\mathbb{E}[\widehat{\mathfrak{R}}_N(l \circ \mathcal{F})] \leqslant \frac{\ell}{N} + 2\ell\mathbb{E}(\widehat{\mathfrak{R}}_N(\mathcal{F}_2)) + b\sqrt{\frac{\log(\mathcal{F}_\infty(\mathcal{F}_1, 1/N))}{N}} + b\sqrt{\frac{2\pi}{N}} \tag{A.77}$$

$$\leqslant \frac{\ell}{N} + \frac{2\ell\mathcal{M}}{N}\left[\sqrt{8/3}(1 + \sqrt{\log(2d)})\sqrt{pd} + \frac{8\mathbf{xy}}{3}(1 + \log(2d)))\right] \tag{A.78}$$

$$+ b\sqrt{\frac{K\log(N(2\mathcal{M}\mathbf{xy}K + 1))}{N}} + b\sqrt{\frac{2\pi}{N}} \tag{A.79}$$

$$\leqslant b\sqrt{\frac{2\pi}{N}} + \frac{\ell}{N} + \frac{10\ell\mathcal{M}}{N}\sqrt{\log(2d)}\sqrt{pd} + \frac{16\mathbf{xy}\ell\mathcal{M}}{N}\log(2d) \tag{A.80}$$

$$+ b\sqrt{\frac{\Gamma\log(N(2\mathcal{M}\mathbf{xy}\Gamma N/p + 1))}{p}} \tag{A.81}$$

$$\leqslant b\sqrt{\frac{2\pi}{N}} + \frac{\ell}{N} + \frac{10\ell\mathcal{M}}{N}\sqrt{\log(2d)}\sqrt{pd} + \frac{16\mathbf{xy}\ell\mathcal{M}}{N}\log(2d) \tag{A.82}$$

$$+ b\sqrt{\frac{\mathbf{x}^2\mathbf{y}^2\log(N(2\mathcal{M}\mathbf{xy}[\mathbf{x}^2\mathbf{y}^2]N/p + 1))}{p}}, \tag{A.83}$$

where at line (A.81) we have plugged in the bound for $K$ from equation (A.41) and at line (A.83) we have used the fact that $\Gamma \leqslant \mathbf{x}^2\mathbf{y}^2$.

We can finally set the value of $p$, to balance the two contributions in equation (A.81) above: we set

$$p := \frac{\mathbf{xy}Nb}{10\mathcal{M}\ell\sqrt{d}}, \tag{A.84}$$

which plugged into equation (A.83) gives

$$\mathbb{E}[\mathfrak{R}_N(l \circ \mathcal{F})] \tag{A.85}$$

$$\leqslant b\sqrt{\frac{2\pi}{N}} + \frac{16\mathbf{xy}\ell\mathcal{M} + \ell}{N}\log(2d) + \frac{10\ell\mathcal{M}}{N}\sqrt{\log(2d)}\sqrt{pd}+ \tag{A.86}$$

$$b\sqrt{\frac{\mathbf{x}^2\mathbf{y}^2\log(N(2\mathcal{M}[\mathbf{x}^3\mathbf{y}^3]N/p + 1))}{p}} \tag{A.87}$$

$$\leqslant b\sqrt{\frac{2\pi}{N}} + \frac{16\mathbf{xy}\ell\mathcal{M} + \ell}{N}\log(2d)+ \tag{A.88}$$

$$\sqrt{\frac{10\ell b\mathbf{xy}\mathcal{M}\sqrt{d}}{N}}\left[\sqrt{\log(2d)} + \sqrt{\log(N(20\mathcal{M}^2\ell\sqrt{d}[\mathbf{x}^2\mathbf{y}^2]/b + 1)}\right], \tag{A.89}$$

as expected.

$\square$

## B  Proof of Propositions 3.1 and 3.2

Proposition 3.2 is included in the wordier version B.1 and proved below.

**Proposition B.1.** *W.p.$\geqslant 1 - \delta$ for all $M$ with $\|M\| \leqslant \mathcal{M}$:*

$$\mathbb{E}\left[l((XMY^\top)_\xi, G_\xi)\right] - \frac{1}{N}\sum_{\xi \in \Omega} l((XMY^\top)_\xi, G_\xi) \tag{B.1}$$

$$\leqslant \frac{4\ell}{\sqrt{N}}\mathcal{M}\max(\sigma_*^1, \sigma_*^2)(1 + \sqrt{\log(2d)}) + \frac{6\ell}{N}\mathcal{M}\mathbf{xy}(1 + \log(2d)) + b\sqrt{\frac{\log(2/\delta)}{2N}},$$

*thus as long as $N \geqslant 9[\mathbf{xy}/\max(\sigma_*^1, \sigma_*^2)]^2(1 + \log(2d))$, we have with probability $\geqslant 1 - \delta$ over the draw of the training set $S$*

$$\mathbb{E}\left[l((XMY^\top)_\xi, G_\xi)\right] - \frac{1}{N}\sum_{\xi \in \Omega} l((XM_SY^\top)_\xi, G_\xi) \tag{B.2}$$

$$\leqslant \frac{6\ell\mathcal{M}\max(\sigma_*^1, \sigma_*^2)(1 + \sqrt{\log(2d)})}{\sqrt{N}} + b\sqrt{\frac{\log(2/\delta)}{2N}}.$$

*Proof of Proposition B.1.* We will show the following bound on the Rademacher complexity of the function class $\mathcal{F}_\mathcal{M} := \{XMY^\top : \|M\| \leqslant \mathcal{M}\}$

$$\mathbb{E}(\mathfrak{R}) \leqslant \frac{1}{\sqrt{N}}\mathcal{M}\sqrt{\frac{8}{3}}\max(\sigma_*^1, \sigma_*^2)(1 + \sqrt{\log(2d)}) + \frac{1}{N}\mathcal{M}\frac{8}{3}\mathbf{xy}(1 + \log(2d)) \tag{B.3}$$

and for $N \geqslant 9[\mathbf{xy}/\max(\sigma_*^1, \sigma_*^2)]^2(1 + \log(2d))$:

$$\mathbb{E}(\mathfrak{R}) \leqslant \frac{3\mathcal{M}\max(\sigma_*^1, \sigma_*^2)(1 + \sqrt{\log(2d)})}{\sqrt{N}}, \tag{B.4}$$

The claims then follow from Theorem F.1, together with Talagrand's contraction Lemma.

Now, by the circular properties of the trace and the duality between the nuclear and spectral norms, writing $F$ for the matrix with $F_{i,j} := \sum_{o=1}^N \sigma_o 1_{\xi^0=(i,j)}$,

$$\left[\frac{1}{N}\langle XMY^\top, F\rangle\right] = \frac{1}{N}\operatorname{Tr}((XMY^\top)^\top F) = \frac{1}{N}\operatorname{Tr}(YM^\top X^\top F) = \frac{1}{N}\operatorname{Tr}(X^\top FYM^\top)$$

$$= \frac{1}{N}\langle X^\top FY, M\rangle \leqslant \|M\|_*\|X^\top FY\|. \tag{B.5}$$

$$\mathfrak{R}(\mathcal{F}_\mathcal{M}) = \mathbb{E}\sup_{\|M\|_* \leqslant \mathcal{M}}\left[\frac{1}{N}\langle XMY^\top, F\rangle\right]$$

$$\leqslant \frac{\mathcal{M}}{N}\mathbb{E}(\|X^\top FY\|). \tag{B.6}$$

The term $\mathbb{E}(\|X^\top FY\|)$ can be written as $\sum_{o=1}^N \sigma_o x_{\xi_1^o} y_{\xi_2^o}^\top = \sum_{o=1}^N \sigma_o x_{i_o} y_{j_o}^\top$, thus, we can prove concentration inequalities for it using the non commutative Bernstein inequality (Proposition (F.4)).

We first note that for all $i, j$, $\|x_i y_j^\top\| \leqslant \mathbf{xy}$. Furthermore, we have $\mathbb{E}_{(i,j)\sim p}\left(\|[x_i y_j^\top][x_i y_j^\top]^\top\|\right) = \|\sum_{i,j} p_{i,j} x_i y_j^\top y_j x_i^\top\| = \|\sum_{i,j} p_{i,j} x_i x_i^\top \|y_j\|^2\| = \|\sum_i x_i x_i^\top q_i\| = \|\widetilde{L}\| = (\sigma_*^1)^2$, and similarly, $\mathbb{E}_{(i,j)\sim p}\left(\|[x_i y_j^\top]^\top[x_i y_j^\top]\|\right) = (\sigma_*^2)^2$.

Using this together with Proposition (F.4) we obtain

$$\mathbb{E}(\|X^\top FY\|) \leqslant \sqrt{N}\sqrt{\frac{8}{3}}\max(\sigma_*^1, \sigma_*^2)(1 + \sqrt{\log(2d)}) + \frac{8}{3}\mathbf{xy}(1 + \log(2d)). \tag{B.7}$$

Plugging this back into equation (B.6), we obtain

$$\mathbb{E}(\mathfrak{R}) \leqslant \frac{1}{\sqrt{N}}\mathcal{M}\sqrt{\frac{8}{3}}\max(\sigma_*^1, \sigma_*^2)(1 + \sqrt{\log(2d)}) + \frac{1}{N}\mathcal{M}\frac{8}{3}\mathbf{xy}(1 + \log(2d)) \tag{B.8}$$

(which yields (B.3)) and as long as $N \geqslant 9[\mathbf{xy}/\max(\sigma_*^1, \sigma_*^2)]^2(1 + \log(2d))$,

$$\mathbb{E}(\mathfrak{R}) \leqslant \frac{1}{\sqrt{N}}\mathcal{M}\sqrt{\frac{8}{3}}\max(\sigma_*^1, \sigma_*^2)(1 + \sqrt{\log(2d)}) + \frac{1}{\sqrt{N}}\mathcal{M}\max(\sigma_*^1, \sigma_*^2)\sqrt{1 + \log(2d)}$$

$$\leqslant \frac{3\mathcal{M}\max(\sigma_*^1, \sigma_*^2)(1 + \sqrt{\log(2d)})}{\sqrt{N}}, \tag{B.9}$$

as expected. This establishes equation (B.4) and the claim follows from Talagrand's concentration lemma and the Rademacher Theorem F.1. $\qquad\square$

Proposition 3.1 follows from the more general result below.

**Proposition B.2.** *Let us write $\mathcal{F}_\mathcal{M}$ for the function class corresponding to matrices of the form $XMY^\top$ with $\|M\|_* \leqslant \mathcal{M}$. Assume uniform sampling and write $\mathcal{K} := \max\left[\sqrt{d_1 \frac{\|X^\top X\|}{m} \frac{\|Y\|_{\mathrm{Fr}}^2}{n}}, \sqrt{d_2 \frac{\|Y^\top Y\|}{n} \frac{\|X\|_{\mathrm{Fr}}^2}{m}}\right].$*

*We have with probability $\geqslant 1 - \delta$, for all $M \in \mathcal{F}_\mathcal{M}$:*

$$\mathbb{E}\left[l((XMY^\top)_\xi, G_\xi + \zeta_\xi)\right] - \frac{1}{N}\sum_{\xi\in\Omega} l((XMY^\top)_\xi, G_\xi + \zeta_\xi)$$

$$\leqslant \frac{4\ell\mathcal{K}\sqrt{rd}(1 + \sqrt{\log(2d)})}{\sqrt{N}} + \frac{6\ell}{N}\mathcal{M}\mathbf{xy}(1 + \log(2d)) + b\sqrt{\frac{\log(2/\delta)}{2N}}, \tag{B.10}$$

*where $\sqrt{r} = (\mathcal{M}/\sqrt{d_1 d_2})$ and $b$ is a bound on the loss.*

*Similarly, as long as*

$$N \geqslant 9\left[\frac{\sqrt{d}\mathbf{xy}}{\mathcal{K}}\right]^2 (1 + \log(2d)) \tag{B.11}$$

*we have with probability $\geqslant 1 - \delta$ over the draw of the training set $S$, for all a $M \in \mathcal{F}_\mathcal{M}$:*

$$\mathbb{E}\left[l((XMY^\top)_\xi, G_\xi + \zeta_\xi)\right] - \frac{1}{N}\sum_{\xi\in\Omega} l((XMY^\top)_\xi, G_\xi + \zeta_\xi)$$

$$\leqslant \frac{6\ell(\mathcal{M}/\sqrt{mn})\max(\sqrt{\|X^\top X\|\|Y\|_{\mathrm{Fr}}^2}, \sqrt{\|Y^\top Y\|\|X\|_{\mathrm{Fr}}^2})(1 + \sqrt{\log(2d)})}{\sqrt{N}} + b\sqrt{\frac{\log(2/\delta)}{2N}}$$

$$= \frac{6\ell\mathcal{K}\sqrt{rd}(1 + \sqrt{\log(2d)})}{\sqrt{N}} + b\sqrt{\frac{\log(2/\delta)}{2N}}. \tag{B.12}$$

*Furthermore, the above result holds under the following more general "uniform inductive marginals" condition (analogous to the "uniform marginals"):*

$$\forall i, \quad \sum_{i,j} p_{i,j}\|y_j\|^2 = \frac{\|Y\|_{\mathrm{Fr}}^2}{mn} \quad and \quad \forall j, \quad \sum_{i,j} p_{i,j}\|x_i\|^2 = \frac{\|X\|_{\mathrm{Fr}}^2}{mn}. \tag{B.13}$$

*Proof of Proposition B.2.* In this case, let us simply compute the values of $\sigma_*^1$ and $\sigma_*^2$. We have, by definition, $q_i = \sum_j p_{i,j}\|y_j\|^2$, thus under conditions (B.13), $q_i = \frac{\|Y\|_{\mathrm{Fr}}^2}{mn}$ for all $i$, and therefore

$$(\sigma_*^1)^2 = \|\widetilde{L}\| = \frac{\|X^\top X\|\|Y\|_{\mathrm{Fr}}^2}{mn}. \tag{B.14}$$

Similarly, we have $\kappa_j = \frac{\|X\|_{\mathrm{Fr}}^2}{mn}$ for all $j$ and

$$(\sigma_*^2)^2 = \|\widetilde{R}\| = \frac{\|Y^\top Y\|\|X\|_{\mathrm{Fr}}^2}{mn}. \tag{B.15}$$

Plugging equations (B.14) and (B.15) into the first result (B.2) yields inequality (B.12) as expected. $\qquad\square$

**Remark:** The sample complexity provided by Proposition B.2 above scales like $O((1/\epsilon^2)[r\mathcal{K}^2 d\log(d)])$ where $\epsilon$ is the tolerance in terms of expected loss. In the case of identity side information we recover the result of $O([rd\log(d)]/\epsilon^2)$ from [7]. In the inductive case, the result is similar but with the correction term offered by $\mathcal{K}^2$, which makes the bound better when the side information has lower effective dimension.

For instance, suppose $d_1 = d_2$, $m = n$ and the dimensions of $X$ and $Y$ are both $k \ll d$, and the top left $k \times k$ entries of $X$ and $Y$ form an identity matrix, with all other entries of $X$ and $Y$ being zero. Suppose also we are in the uniform sampling scenario. We then have that $\mathcal{K}^2 = k^2/d^2$, yielding a sample complexity $O([drk^2/d^2 \log(d)]/\epsilon^2) = O([kr\frac{k}{d}\log(d)]/\epsilon^2)$, which is counter-intuitively tight because of the extra factor of $\frac{k}{d}$. Indeed, it would appear the problem is similar to the uniform sampling case with identity side information and a $k \times k$ matrix, which should yield a bound of $O(kr\log(k))$, but not better.

However, this factor comes from the scale parameter $\epsilon$. Indeed, recall that the expected error is computed with respect to the sampling distribution in both cases. In this example, every entry $(i, j)$ where either $x_i = 0$ or $y_j = 0$ is known to be equal to zero. This means that we only need $\epsilon d^2/k^2$ accuracy on the non zero entries to reach $\epsilon$ accuracy overall. However, only $k^2/d^2$ entries are usable (corresponding to $x_i \neq 0$ and $x_j \neq 0$). This means if we were using an optimal strategy, we would actually have a sample complexity of $O(\frac{k^2}{d^2}k\log(k))$. Our own sample complexity is actually slightly worse than that due to the smoothing procedure, which ensures stability and theoretical guarantees, but deprives us of a small part of the advantages of the weighting and adjustment. It is worth noting that this slight limitation is similar to an analogous weakness in the results of [7]: indeed, even in the MC case treated in that reference, the smoothed weighted trace norm [1] (which requires knowledge of the distribution) yields bounds of order $O(rn\log(n))$. That is the case even if the (known) distribution happens to be supported on a subset of the matrix with size $\tilde{n} \times \tilde{n}$ where $\tilde{n} \ll n$, despite the fact that a direct application of the result to the smaller matrix would yield better bounds in this case. It is interesting but challenging to consider the possibility of extending both our results and those of [7] to cover for these effects.

## C   Proof of Proposition 3.3

Proposition 3.3 follows from the wordier result below:

**Proposition C.1** (Long version of proposition 3.3). *W.p.* $\geqslant 1 - \delta$, *for all* $M \in \widetilde{\mathcal{F}}_r$:

$$\mathbb{E}\left[l((XMY^\top)_\xi, G_\xi + \zeta_\xi)\right] - \frac{1}{N}\sum_{\xi \in \Omega} l((XMY^\top)_\xi, G_\xi + \zeta_\xi) \tag{C.1}$$

$$\leqslant \frac{8\ell\sqrt{\Gamma}\sqrt{r}\sqrt{d}(1 + \sqrt{\log(2d)})}{\sqrt{N}} + \frac{12\ell\mathbf{xy}\sqrt{d_1 d_2 r}(1 + \log(2d))}{N} + b\sqrt{\frac{\log(2/\delta)}{2N}}.$$

*Further, as long as* $N \geqslant \min(d_1, d_2)\frac{18\mathbf{x}^2\mathbf{y}^2}{\Gamma}(1 + \log(2d))$, *we have with probability* $\geqslant 1 - \delta$ *over the draw of the training set* $S$ *for all* $M \in \widetilde{\mathcal{F}}_r$

$$\mathbb{E}\left[l((XMY^\top)_\xi, G_\xi)\right] - \frac{1}{N}\sum_{\xi \in \Omega} l((XMY^\top)_\xi, G_\xi)$$

$$\leqslant \frac{12\ell\sqrt{\Gamma}\sqrt{r}\sqrt{d}(1 + \sqrt{\log(2d)})}{\sqrt{N}} + b\sqrt{\frac{\log(2/\delta)}{2N}}, \tag{C.2}$$

*Proof.* This follows from a careful application of the Proposition B.1 to a modified problem where the side information matrices $X$ and $Y$ are replaced by $XP^{-1}\widetilde{D}^{-\frac{1}{2}}$ and $YQ^{-1}\widetilde{E}^{-\frac{1}{2}}$.

Let $\theta(\mathbf{x})$, $\theta(\sigma^1_*)$ (etc.) denote the value taken by $\mathbf{x}$, $\sigma^1_*$ (etc.) after the substitution above. Thus, we only need to show that replacing the values of the quantities appearing in formula (B.2) by their new values (computed below gives the formula (C.2)).

---

[1]The exact, non-empirical version

We have $\theta(\mathbf{x}) = \|[XP^{-1}\widetilde{D}^{-\frac{1}{2}}]^\top\|_{2,\infty} \leqslant \mathbf{x}\|\widetilde{D}^{-\frac{1}{2}}\| \leqslant \mathbf{x}\sqrt{2\frac{d_1}{\Gamma}}$. And similarly, $\theta(\mathbf{y}) \leqslant \mathbf{y}\sqrt{2\frac{d_2}{\Gamma}}$.
We also have $\theta(\mathcal{M}) = \sqrt{r}\Gamma$.

One trickier computation is that of $\theta(\sigma_*^1)$ and $\theta(\sigma_*^2)$:

$\theta(\sigma_*^1)$ is the spectral norm of the matrix $\theta(X) = XP^{-1}\widetilde{D}^{-\frac{1}{2}}$ evaluated with respect to the post-substitution inner product $\langle,\rangle_{\theta(l)}$. Note that the new values $\theta(q_i)$ and $\theta(\kappa_i)$ for $\kappa_j$ and $q_j$ have the following properties:

$$\theta(q_i) = \sum_j p_{i,j}\|\theta(y_j)\|^2$$
$$= \sum_j p_{i,j}\|y_j Q^{-1}\widetilde{E}^{\frac{1}{2}}\|^2$$
$$\leqslant \sum_j p_{i,j}\|y_j\|^2\|\widetilde{E}^{\frac{1}{2}}\|^2$$
$$\leqslant \frac{2q_i d_2}{\Gamma}, \tag{C.3}$$

and similarly

$$\theta(\kappa_j) \leqslant \frac{2\kappa_j d_1}{\Gamma}.$$

In particular, for any vector $v \in \mathbb{R}^m$ we have

$$\|v\|_{\theta(l)}^2 = \langle v, v\rangle_{\theta(l)} = v^\top \operatorname{diag}(\theta(q))v \leqslant v^\top \operatorname{diag}(q)v\frac{2d_2}{\Gamma} \leqslant \|v\|_l^2\frac{2d_2}{\Gamma}, \tag{C.4}$$

and similarly for vectors in $\mathbb{R}^n$ with a factor of $\frac{2d_1}{\Gamma}$.

As a result we can compute:

$$\theta(\sigma_*^1)^2 = \|\theta(X)^\top \operatorname{diag}(\theta(q))(\theta(X))\|$$
$$= \|(XP^{-1}\widetilde{D}^{-\frac{1}{2}})^\top \operatorname{diag}(\theta(q))(XP^{-1}\widetilde{D}^{-\frac{1}{2}})\|$$
$$\leqslant \frac{2d_2}{\Gamma}\|(XP^{-1}\widetilde{D}^{-\frac{1}{2}})^\top \operatorname{diag}(q)(XP^{-1}\widetilde{D}^{-\frac{1}{2}})\|$$
$$= \frac{2d_2}{\Gamma}\|\widetilde{D}^{-\frac{1}{2}}P[P^{-1}DP]P^{-1}\widetilde{D}^{-\frac{1}{2}}\| = \frac{2d_2}{\Gamma}\|2I\|$$
$$\leqslant \frac{4d_2}{\Gamma}, \tag{C.5}$$

and similarly

$$\theta(\sigma_*^1)^2 \leqslant \frac{4d_1}{\Gamma}. \tag{C.6}$$

Plugging the post substitution values computed above into each of the relevant expressions in Proposition B.1, we obtain first that w.p. $\geqslant 1 - \delta$:

$$\frac{4\ell}{\sqrt{N}}\theta(\mathcal{M})\max(\theta(\sigma_*^1),\theta(\sigma_*^2))(1 + \sqrt{\log(2d)}) + \frac{6\ell}{N}\theta(\mathcal{M})\theta(\mathbf{xy})(1 + \log(2d)) \tag{C.7}$$

$$\leqslant \frac{4\ell}{\sqrt{N}}\Gamma\sqrt{r}\max(\sqrt{\frac{4d_2}{\Gamma}},\sqrt{\frac{4d_1}{\Gamma}})(1 + \sqrt{\log(2d)}) + + \frac{12\ell\sqrt{d_1 d_2}/\Gamma}{N}\sqrt{r}\Gamma\mathbf{xy}(1 + \log(2d))$$
$$\tag{C.8}$$

$$= \frac{8\ell\sqrt{\Gamma}\sqrt{r}\sqrt{d}(1 + \sqrt{\log(2d)})}{\sqrt{N}} + \frac{12\ell\sqrt{rd_1 d_2}}{N}\mathbf{xy}(1 + \log(2d)) \tag{C.9}$$

$$\tag{C.10}$$

as expected.

And then also that (w.p. $\geqslant 1 - \delta$) $\mathbb{E}\left[l((XMY^\top)_\xi, G_\xi)\right] - \frac{1}{N}\sum_{\xi\in\Omega} l((XMY^\top)_\xi, G_\xi) - b\sqrt{\frac{\log(2/\delta)}{2n}}$ is bounded above by

$$\frac{6\ell\theta(\mathcal{M})\max(\theta(\sigma_*^1), \theta(\sigma_*^2))(1 + \sqrt{\log(2d)})}{\sqrt{N}} = \frac{6\ell\sqrt{\Gamma}\sqrt{r}\max(\sqrt{\frac{4d_2}{\Gamma}}, \sqrt{\frac{4d_1}{\Gamma}})(1 + \sqrt{\log(2d)})}{\sqrt{N}}$$

$$= \frac{12\ell\sqrt{\Gamma}\sqrt{r}\sqrt{d}(1 + \sqrt{\log(2d)})}{\sqrt{N}},$$

with the condition that $N$ needs to be larger than

$$9\theta([\mathbf{xy}/\max(\sigma_*^1, \sigma_*^2)])^2(1 + \log(2d))$$

$$= 9\left[\mathbf{xy}\sqrt{\frac{2d_1}{\Gamma}}\sqrt{\frac{2d_2}{\Gamma}}/\sqrt{\frac{2}{\Gamma}}\sqrt{d}\right]^2 \sqrt{r}\Gamma(1 + \log(2d))$$

$$= \min(d_1, d_2)\frac{18\mathbf{x}^2\mathbf{y}^2}{\Gamma}(1 + \log(2d)), \qquad (\text{C.11})$$

as expected. $\qquad\square$

## D   Proof of Theorem 3.2

Theorem 3.2 follows from the longer version below.

**Theorem D.1.** *Fix any target matrix $G$ and distribution $p$. Define $\check{Z}_S = \arg\min(\hat{l}_S(Z) : Z \in \check{\mathcal{F}}_r)$. For any $\delta \in (0, 1)$, with probability $\geqslant 1 - \delta$ over the draw of the training set we have*

$$l(\check{Z}) \leqslant \inf_{\check{\mathcal{F}}_r} l(Z) + \left[48\ell\sqrt{r}\gamma(\mathbf{x} + \mathbf{y})^2 + 2b\right]\sqrt{\frac{2\log(\frac{12d}{\delta})[\gamma(d+3) + \gamma^2]}{N}}, \qquad (\text{D.1})$$

*where $\gamma = \frac{\mathbf{x}^2\mathbf{y}^2}{x^2 y^2}$. In particular, in expectation over the draw of the training set we have*

$$l(\check{Z}) \leqslant \inf_{\check{\mathcal{F}}_r} l(Z) + \left[96\ell\sqrt{r}\gamma(\mathbf{x} + \mathbf{y})^2 + 4b\right]\sqrt{\frac{2\log(12d)[\gamma(d+3) + \gamma^2]}{N}}. \qquad (\text{D.2})$$

*Proof of Theorem D.1.* The lemmas which are used are proved below.

We write $Z^*$ for an element of $\arg\min_{\check{\mathcal{F}}_r} l(Z)$. First, by applying Proposition 3.3, we have that $N \geqslant \sqrt{\min(d_1, d_2)}18\gamma(1 + \log(2d))$, we have with probabiltiy $\geqslant 1 - \delta/3$:

$$l(\check{Z}) - \hat{l}_S(\check{Z}) \leqslant \frac{12\ell\sqrt{\Gamma}\sqrt{r}\sqrt{d}(1 + \sqrt{\log(2d)})}{\sqrt{N}} + b\sqrt{\frac{\log(6/\delta)}{2N}}. \qquad (\text{D.3})$$

Define $C(S) = \max\left(0, \left\|\frac{1}{\sqrt{r_*}\Gamma}\widetilde{M_*}\right\|_* - 1\right)$. Note that $(1 - C(S))Z^* \in \check{\mathcal{F}}_r$. Thus, using Lemma E.4 we also have similarly with probability $\geqslant 1 - \delta/3$:

$$\hat{l}_S((1 - C(S))Z^*) - l((1 - C(S))Z^*) \qquad (\text{D.4})$$

$$\leqslant \frac{24\ell\sqrt{\Gamma}\gamma\sqrt{r}\sqrt{d}(1 + \sqrt{\log(2d)})}{\sqrt{N}} + b\sqrt{\frac{\log(6/\delta)}{2N}},$$

as long as $N \geqslant 8\gamma^2 + \gamma[8d + 20][\log(2d) + \log(\frac{6}{\delta})]$. By definition, since $(1 - C(S))Z^* \in \check{\mathcal{F}}_r$ we also have

$$\hat{l}_S(\check{Z}) - \hat{l}_S((1 - C(S))Z^*) \leqslant 0. \qquad (\text{D.5})$$

Next, by Lemma E.3, as long as $N \geqslant 2\log(\frac{6d}{\delta})[\gamma(d+3) + \gamma^2]$, with probability $\geqslant 1 - \delta/3$ over the draw of the training set:

$$l((1 - C(S))Z_*) - l(Z_*)$$

$$\leqslant \ell \|\widetilde{M_*}\|_* \left[ \frac{1}{\underline{x}^2} + \frac{1}{\underline{y}^2} \right] \sqrt{\frac{2 \log(\frac{12d}{\delta})[\gamma(d+3) + \gamma^2]}{N}}$$

$$\leqslant \ell \sqrt{r} \Gamma \left[ \frac{1}{\underline{x}^2} + \frac{1}{\underline{y}^2} \right] \sqrt{\frac{2 \log(\frac{12d}{\delta})[\gamma(d+3) + \gamma^2]}{N}}. \tag{D.6}$$

Combining all of the above, we get that as long as $N \geqslant 2 \log(\frac{6d}{\delta})[\gamma(d+3) + \gamma^2]$ and $N \geqslant \sqrt{\min(d_1, d_2)} 18\gamma(1 + \log(2d))$, we have

$$l(\check{Z}) - l(Z_*) \tag{D.7}$$

$$\leqslant l(\check{Z}) - \hat{l}_S(\check{Z}) + \hat{l}_S(\check{Z}) - \hat{l}_S((1 - C(S))Z^*) + \tag{D.8}$$

$$\hat{l}_S((1 - C(S))Z^*) - l((1 - C(S))Z^*) + l((1 - C(S))Z_*) - l(Z_*)$$

$$\leqslant \frac{12\ell\sqrt{\Gamma}\sqrt{r}\sqrt{d}(1 + \sqrt{\log(2d)})}{\sqrt{N}} + b\sqrt{\frac{\log(6/\delta)}{2N}} \tag{D.9}$$

$$+ \frac{24\ell\gamma\sqrt{\Gamma}\sqrt{r}\sqrt{d}(1 + \sqrt{\log(2d)})}{\sqrt{N}} + b\sqrt{\frac{\log(6/\delta)}{2N}} \tag{D.10}$$

$$+ \ell\sqrt{r}\Gamma \left[ \frac{1}{\underline{x}^2} + \frac{1}{\underline{y}^2} \right] \sqrt{\frac{2 \log(\frac{12d}{\delta})[\gamma(d+3) + \gamma^2]}{N}} \tag{D.11}$$

$$\leqslant \frac{48\ell\gamma\sqrt{\Gamma}\sqrt{r}\sqrt{d}(1 + \sqrt{\log(2d)})}{\sqrt{N}} + 2b\sqrt{\frac{\log(6/\delta)}{2N}} \tag{D.12}$$

$$+ \ell\sqrt{r}\gamma(\mathbf{x}^2 + \mathbf{y}^2)\sqrt{\frac{2 \log(\frac{12d}{\delta})[\gamma(d+3) + \gamma^2]}{N}} \tag{D.13}$$

$$\leqslant \frac{48\ell\gamma\sqrt{\Gamma}\sqrt{r}\sqrt{d}(1 + \sqrt{\log(2d)})}{\sqrt{N}} \tag{D.14}$$

$$+ \left[ \ell\sqrt{r}\gamma(\mathbf{x}^2 + \mathbf{y}^2) + 2b \right] \sqrt{\frac{2 \log(\frac{12d}{\delta})[\gamma(d+3) + \gamma^2]}{N}} \tag{D.15}$$

$$\leqslant \left[ 48\ell\sqrt{r}\gamma(\mathbf{x} + \mathbf{y})^2 + 2b \right] \sqrt{\frac{2 \log(\frac{12d}{\delta})[\gamma(d+3) + \gamma^2]}{N}}. \tag{D.16}$$

Furthermore, the conditions on $N$ can now be dropped since the RHS is greater than $b$ whenever $N$ fails to satisfy either of them.

The expectation version of the theorem follows directly from Lemma F.5. $\qquad \square$

# E  Lemmas for the proof of Theorem 3.2

**Proposition E.1.** *For any $\delta \in (0, 1)$, with probability $\geqslant 1 - \delta$, we have*

$$\frac{1}{\sqrt{2}} \leqslant \left\| \widetilde{D}^{\frac{1}{2}} P \widehat{P}^{-1} \check{D}^{-\frac{1}{2}} \right\| \leqslant \sqrt{2}, \tag{E.1}$$

*as long as $N \geqslant 8\gamma^2 + \gamma[8d_1 + 20][\log(2d_1) + \log(\frac{1}{\delta})]$.*

*Similarly, for any $\delta \in (0, 1)$, with probability $\geqslant 1 - \delta$, we have*

$$\frac{1}{\sqrt{2}} \leqslant \left\| \widetilde{E}^{\frac{1}{2}} Q \widehat{Q}^{-1} \check{E}^{-\frac{1}{2}} \right\| \leqslant \sqrt{2}, \tag{E.2}$$

*as long as $N \geqslant [8\gamma^2 + \gamma[8d_2 + 20]][\log(2d_2) + \log(\frac{1}{\delta})]$.*

*Proof.* We will write $T$ for the matrix $\widetilde{D}^{\frac{1}{2}}P\widehat{P}^{-1}\widecheck{D}^{-\frac{1}{2}}$ whose spectral norm we want to bound.

We consider the matrix

$$\mathcal{T} := \widetilde{D}^{-\frac{1}{2}}P\widehat{P}^{-1}\widecheck{D}\widehat{P}P^{-1}\widetilde{D}^{-\frac{1}{2}} = (T^{-1})^{\top}(T^{-1}). \tag{E.3}$$

We can write $\mathcal{T}$ as a sum of independent random matrices as follows:

$$
\begin{aligned}
\mathcal{T} :&= \frac{1}{N}\sum_{\xi\in\Omega}\widetilde{D}^{-\frac{1}{2}}P\left[\frac{1}{2}x_{\xi_1}x_{\xi_1}^{\top}\|y_{\xi_2}\|^2 + \frac{1}{2d_1}\|x_{\xi_1}\|^2\|y_{\xi_2}\|^2 I\right]P^{-1}\widetilde{D}^{-\frac{1}{2}} \\
&= \frac{1}{N}\sum_{i,j}h_{i,j}\widetilde{D}^{-\frac{1}{2}}P\left[\frac{1}{2}x_i x_i^{\top}\|y_j\|^2 + \frac{1}{2d_1}\|x_i\|^2\|y_j\|^2 I\right]P^{-1}\widetilde{D}^{-\frac{1}{2}} \\
&= \frac{1}{N}\sum_{o=1}^{N}\Lambda_o, \tag{E.4}
\end{aligned}
$$

where $\Omega$ is the multi-set containing all the iid sampled entries and $\Lambda = \widetilde{D}^{-\frac{1}{2}}P\left[\frac{1}{2}x_{\xi_1^o}x_{\xi_1^o}^{\top}\|y_{\xi_2^o}\|^2 + \frac{1}{2d_1}\|x_{\xi_1}\|^2\|y_{\xi_2}\|^2 I\right]P^{-1}\widetilde{D}^{-\frac{1}{2}}$ and the $\xi^o$ $(o = 1,\dots,N)$ are the sampled entries.

Now, we can compute the expectation of $\mathcal{T}$ and $\Lambda$ as follows:

$$\mathbb{E}(\mathcal{T}) = \mathbb{E}(\Lambda_{\xi}) = \sum_{i,j}p_{i,j}\widetilde{D}^{-\frac{1}{2}}P\left[\frac{1}{2}x_i x_i^{\top}\|y_j\|^2 + \frac{1}{2d_1}\|x_i\|^2\|y_j\|^2 I\right]P^{-1}\widetilde{D}^{-\frac{1}{2}} \tag{E.5}$$

$$= \widetilde{D}^{-\frac{1}{2}}PP^{-1}\widetilde{D}PP^{-1}\widetilde{D}^{-\frac{1}{2}} = I. \tag{E.6}$$

Now, note that for any $(i,j) \in \{1,2,\dots,m\}\times\{1,2,\dots,n\}$ we have

$$\|\Lambda_{(i,j)}\| = \left\|\widetilde{D}^{-\frac{1}{2}}P\left[\frac{1}{2}x_i x_i^{\top}\|y_j\|^2 + \frac{1}{2d_1}\|x_i\|^2\|y_j\|^2 I\right]P^{-1}\widetilde{D}^{-\frac{1}{2}}\right\| \leqslant (\frac{1}{2}\mathbf{x}^2\mathbf{y}^2 + \frac{1}{2d_1}\widehat{\Gamma})\|\widetilde{D}\|^{-1}$$

$$\leqslant (\frac{1}{2}\mathbf{x}^2\mathbf{y}^2 + \frac{1}{2d_1}\widehat{\Gamma})\frac{2d_1}{\widecheck{\Gamma}} \leqslant \frac{\mathbf{x}^2\mathbf{y}^2}{\underline{x}^2\underline{y}^2} + 1 = \gamma + 1 \tag{E.7}$$

By abuse of notation, we write below $\Lambda$ for the random variable $\Lambda_{\xi}$ where $\xi \in \{1,2,\dots,m\}\times\{1,2,\dots,n\}$ is distributed according to $p$.

We now begin to bound $\|\mathbb{E}((\Lambda - \mathbb{E}(\Lambda))(\Lambda - \mathbb{E}(\Lambda))^{\top})\|$. We first note that

$$
\begin{aligned}
\left\|\mathbb{E}\left((\Lambda - \mathbb{E}(\Lambda))(\Lambda - \mathbb{E}(\Lambda))^{\top}\right)\right\| &= \left\|\mathbb{E}(\Lambda\Lambda^{\top}) - \mathbb{E}(\Lambda)\mathbb{E}(\Lambda)^{\top}\right\| \\
&= \left\|\mathbb{E}(\Lambda\Lambda^{\top}) - I\right\| \leqslant \left\|\mathbb{E}(\Lambda\Lambda^{\top})\right\|. \tag{E.8}
\end{aligned}
$$

Thus, we now note that by equation (E.4):

$$\mathbb{E}(\Lambda\Lambda^{\top}) = \tag{E.9}$$

$$\sum_{i,j}p_{i,j}\widetilde{D}^{-\frac{1}{2}}P\left[\frac{\|y_j\|^2}{2}x_i x_i^{\top} + \frac{\|x_i\|^2\|y_j\|^2}{2d_1}I\right]P^{-1}\widetilde{D}^{-1}P\left[\frac{\|y_j\|^2}{2}x_i x_i^{\top} + \frac{\|x_i\|^2\|y_j\|^2}{2d_1}I\right]^{\top}P^{-1}\widetilde{D}^{-\frac{1}{2}}.$$

From this it follows that

$$\|\mathbb{E}(\Lambda\Lambda^{\top})\| \leqslant \left\|\sum_{i,j}p_{i,j}\widetilde{D}^{-\frac{1}{2}}P\left[\frac{1}{2}x_i x_i^{\top}\|y_j\|^2\right]P^{-1}\widetilde{D}^{-1}P\left[\frac{1}{2}x_i x_i^{\top}\|y_j\|^2\right]P^{-1}\widetilde{D}^{-\frac{1}{2}}\right\| \tag{E.10}$$

$$+ \left\|\sum_{i,j}p_{i,j}\widetilde{D}^{-\frac{1}{2}}P\left[\frac{1}{2d_1}\|x_i\|^2\|y_j\|^2 I\right]P^{-1}\widetilde{D}^{-1}P\left[\frac{1}{2}x_i x_i^{\top}\|y_j\|^2\right]P^{-1}\widetilde{D}^{-\frac{1}{2}}\right\|$$

$$+ \left\| \sum_{i,j} p_{i,j} \widetilde{D}^{-\frac{1}{2}} P \left[ \frac{1}{2} x_i x_i^\top \|y_j\|^2 \right] P^{-1} \widetilde{D}^{-1} P \left[ \frac{1}{2d_1} \|x_i\|^2 \|y_j\|^2 I \right] P^{-1} \widetilde{D}^{-\frac{1}{2}} \right\|$$

$$+ \left\| \sum_{i,j} p_{i,j} \widetilde{D}^{-\frac{1}{2}} P \left[ \frac{1}{2d_1} \|x_i\|^2 \|y_j\|^2 I \right] P^{-1} \widetilde{D}^{-1} P \left[ \frac{1}{2d_1} \|x_i\|^2 \|y_j\|^2 I \right] P^{-1} \widetilde{D}^{-\frac{1}{2}} \right\|.$$

We bound each of the four terms above separately:

For the first (and key) term, we have:

$$\frac{1}{4} \left\| \sum_{i,j} p_{i,j} \widetilde{D}^{-\frac{1}{2}} P \left[ x_i x_i^\top \|y_j\|^2 \right] P^{-1} \widetilde{D}^{-1} P \left[ x_i x_i^\top \|y_j\|^2 \right] P^{-1} \widetilde{D}^{-\frac{1}{2}} \right\|$$

$$\leqslant \frac{1}{4} \left\| \sum_{i,j} p_{i,j} \widetilde{D}^{-\frac{1}{2}} P \left[ x_i x_i^\top \|y_j\|^2 \right] P^{-1} \widetilde{D}^{-\frac{1}{2}} \right\| \sup_{i,j} \left\| \widetilde{D}^{-\frac{1}{2}} P \left[ x_i x_i^\top \|y_j\|^2 \right] P^{-1} \widetilde{D}^{-\frac{1}{2}} \right\|$$

$$\leqslant \frac{1}{4} \frac{2d_1}{\Gamma} \mathbf{x}^2 \mathbf{y}^2 \left\| \sum_{i,j} p_{i,j} \widetilde{D}^{-\frac{1}{2}} P \left[ x_i x_i^\top \|y_j\|^2 \right] P^{-1} \widetilde{D}^{-\frac{1}{2}} \right\|$$

$$= \frac{d_1}{2\Gamma} \mathbf{x}^2 \mathbf{y}^2 \left\| \widetilde{D}^{-\frac{1}{2}} P P^{-1} D P P^{-1} \widetilde{D}^{-\frac{1}{2}} \right\| = \frac{d_1}{2\Gamma} \mathbf{x}^2 \mathbf{y}^2 \| D \widetilde{D}^{-1} \| \leqslant \frac{d_1}{\underline{x}^2 \underline{y}^2} \mathbf{x}^2 \mathbf{y}^2 = d_1 \gamma. \quad \text{(E.11)}$$

For the second term we have

$$\left\| \sum_{i,j} p_{i,j} \widetilde{D}^{-\frac{1}{2}} P \left[ \frac{1}{2d_1} \|x_i\|^2 \|y_j\|^2 I \right] P^{-1} \widetilde{D}^{-1} P \left[ \frac{1}{2} x_i x_i^\top \|y_j\|^2 \right] P^{-1} \widetilde{D}^{-\frac{1}{2}} \right\|$$

$$\leqslant \frac{\mathbf{x}^2 \mathbf{y}^2}{2d_1} \left\| \widetilde{D}^{-\frac{1}{2}} I \widetilde{D}^{-\frac{1}{2}} \right\| \left\| \sum_{i,j} p_{i,j} \widetilde{D}^{-\frac{1}{2}} P \left[ \frac{1}{2} x_i x_i^\top \|y_j\|^2 \right] P^{-1} \widetilde{D}^{-\frac{1}{2}} \right\|$$

$$\leqslant \frac{\mathbf{x}^2 \mathbf{y}^2}{2d_1} \frac{2d_1}{\Gamma} \frac{1}{2} \left\| \widetilde{D}^{-\frac{1}{2}} P P^{-1} D P P^{-1} \widetilde{D}^{-\frac{1}{2}} \right\| \leqslant \frac{\mathbf{x}^2 \mathbf{y}^2}{\underline{x}^2 \underline{y}^2} = \gamma. \quad \text{(E.12)}$$

For the third term we obtain similarly:

$$\left\| \sum_{i,j} p_{i,j} \widetilde{D}^{-\frac{1}{2}} P \left[ \frac{1}{2} x_i x_i^\top \|y_j\|^2 \right] P^{-1} \widetilde{D}^{-1} P \left[ \frac{1}{2d_1} \|x_i\|^2 \|y_j\|^2 I \right] P^{-1} \widetilde{D}^{-\frac{1}{2}} \right\| \leqslant \frac{\mathbf{x}^2 \mathbf{y}^2}{\underline{x}^2 \underline{y}^2} = \gamma. \quad \text{(E.13)}$$

Finally for the fourth term we have:

$$\left\| \sum_{i,j} p_{i,j} \widetilde{D}^{-\frac{1}{2}} P \left[ \frac{1}{2d_1} \|x_i\|^2 \|y_j\|^2 I \right] P^{-1} \widetilde{D}^{-1} P \left[ \frac{1}{2d_1} \|x_i\|^2 \|y_j\|^2 I \right] P^{-1} \widetilde{D}^{-\frac{1}{2}} \right\|$$

$$\leqslant \left\| \frac{\mathbf{x}^2 \mathbf{y}^2 \widetilde{D}^{-1}}{2d_1} \right\|^2 \leqslant \left\| \frac{\mathbf{x}^2 \mathbf{y}^2 2d_1}{2d_1 \Gamma} \right\|^2 \leqslant \gamma^2. \quad \text{(E.14)}$$

Plugging equations (E.11), (E.12), (E.13) and (E.14) into equations (E.10) and (E.8) we finally obtain:

$$\left\| \mathbb{E} \left( (\Lambda - \mathbb{E}(\Lambda))(\Lambda - \mathbb{E}(\Lambda))^\top \right) \right\| \leqslant \left\| \mathbb{E}(\Lambda \Lambda^\top) \right\| \leqslant \gamma(d_1 + 2) + \gamma^2. \quad \text{(E.15)}$$

We now apply the non-communtative Bernstein inequality (F.3) to $\mathcal{T} - \mathbb{E}(\mathcal{T})$ which is the average of $N$ i.i.d. instances of $\Lambda$. With the notation from Proposition F.3 we have $M = \gamma + 1$ (from equation (E.7)), $\nu^2 = \sum_{o=1}^{N} \frac{1}{N^2} [\gamma(d_1 + 2) + \gamma^2] = \frac{1}{N} [\gamma(d_1 + 2) + \gamma^2]$ (from equation (E.15)), $n = m = d_1$ and we obtain (for all $\tau$):

$$\mathbb{P} \left( \|\mathcal{T} - \mathbb{E}(\mathcal{T})\| \geqslant \tau \right) \leqslant (2d_1) \exp \left( -\frac{\tau^2/2}{\nu^2 + M\tau/3} \right)$$

$$\leq (2d_1) \exp\left(-\frac{N\tau^2/2}{[\gamma(d_1+2)+\gamma^2]+(\gamma+1)\tau/3}\right) \tag{E.16}$$

Setting $\tau = \frac{1}{2}$ we obtain, as long as $N \geq [8\gamma^2 + \gamma[8d_1 + 20]][\log(2d_1) + \log(\frac{1}{\delta})]$:

$$\mathbb{P}\left(\|\mathcal{T} - \mathbb{E}(\mathcal{T})\| \geq \frac{1}{2}\right) \leq (2d_1) \exp\left(-\frac{\tau^2/2}{\nu^2 + M\tau/3}\right)$$

$$\leq (2d_1) \exp\left(-\frac{N}{8[\gamma(d_1+2)+\gamma^2]+2(\gamma+1)}\right)$$

$$\leq (2d_1) \exp\left(-\frac{N}{8\gamma^2 + \gamma[8d_1+20]}\right)$$

$$\leq \delta. \tag{E.17}$$

Thus, we now know that as long as $N \geq 8\gamma^2 + \gamma[8d_1 + 20][\log(2d_1) + \log(\frac{1}{\delta})]$ we have with probability $\geq 1 - \delta$ that

$$\|\mathcal{T} - \mathbb{E}(\mathcal{T})\| \leq \frac{1}{2}. \tag{E.18}$$

This already implies that $\|\mathcal{T}\| \leq 1 + 0.5 \leq 2$ and therefore $\|T^{-1}\| \leq \sqrt{2}$, leaving us only the second inequality to prove.

We will show that inequality (E.18) actually implies inequality (E.1).

To that effect, recall from equation (E.3) that $\mathcal{T} = (T^{-1})^\top (T^{-1}) = G^{-1}$ where $G = TT^\top$. Thus we have $G = [I + (\mathcal{T} - I)]^{-1}$. Rewriting this as $G[I + (\mathcal{T} - I)] = I$ and taking spectral norms on both sides we obtain

$$\|G\|\sigma_{\inf}([I + (\mathcal{T} - I)]) \leq 1, \tag{E.19}$$

where for any symmetric matrix $A$, $\sigma_{\inf}(A)$ denotes the smallest eigenvalue of $A$.

Now note that by inequality (E.18), for any unit vector $v$, we have

$$v^\top [I + (\mathcal{T} - I)] v = 1 - v^\top (\mathcal{T} - I) v \geq 1 - \|(\mathcal{T} - I)\| \geq 1 - \frac{1}{2} = \frac{1}{2}. \tag{E.20}$$

Thus the smallest eigenvalue of $\|[I + (\mathcal{T} - I)]\|$ is bounded below by $\frac{1}{2}$, i.e.

$$\sigma_{\inf}([I + (\mathcal{T} - I)]) \geq \frac{1}{2}. \tag{E.21}$$

Plugging inequality (E.21) back into identity (E.19), we obtain:

$$\|G\| \leq 2. \tag{E.22}$$

Finally, recall that $G = TT^\top$ and thus $\|G\| = \|T\|^2$, which together with inequality (E.22) finally implies

$$\|T\| \leq \sqrt{2}, \tag{E.23}$$

as expected. $\qquad\square$

**Lemma E.2.** *Let* $\widetilde{M} \in \mathbb{R}^{d_1 \times d_2}$ *be a fixed matrix with* $\|M\|_* = 1$. *For any* $\delta \in (0,1)$ *we have that w.p.* $\geq 1 - \delta$, *(as long as* $N \geq 2\log(\frac{2d}{\delta})[\gamma(d+3)+\gamma^2]$*):*

$$\|\widetilde{M}\|_* = \|\check{D}^{\frac{1}{2}}\hat{P}P^{-1}\tilde{D}^{-\frac{1}{2}}\widetilde{M}\tilde{E}^{-\frac{1}{2}}Q\hat{Q}^{-1}\check{E}^{\frac{1}{2}}\|_* \leq \|\widetilde{M}\|_* \left[1 + \sqrt{\frac{2\log(\frac{4d}{\delta})[\gamma(d+3)+\gamma^2]}{N}}\right], \tag{E.24}$$

*where* $d := \max(d_1, d_2)$.

*Proof.* Writing $\widetilde{M}$ for the matrix $\check{D}^{\frac{1}{2}}\widehat{P}P^{-1}\widetilde{D}^{-\frac{1}{2}}\widetilde{M}\widetilde{E}^{-\frac{1}{2}}Q\widehat{Q}^{-1}\check{E}^{\frac{1}{2}}$ we want to control, we have by the properties of the trace norm:

$$\|\widetilde{M}\|_* = \max_{A,B}\left(\frac{1}{2}\left[\|A\|_{\mathrm{Fr}}^2 + \|B\|_{\mathrm{Fr}}^2\right] : AB^\top = \widetilde{M}\right)$$

Let $\check{A}$, $\check{B}$ denote the matrices which realize the maximum above. Now note that we have $[\check{D}^{\frac{1}{2}}\widehat{P}P^{-1}\widetilde{D}^{-\frac{1}{2}}]^{-1}\check{A}\check{B}[\check{E}^{-\frac{1}{2}}Q\widehat{Q}^{-1}\widehat{E}^{\frac{1}{2}}]^{-1} = [\check{D}^{\frac{1}{2}}\widehat{P}P^{-1}\widetilde{D}^{-\frac{1}{2}}]^{-1}\widetilde{M}[\check{E}^{-\frac{1}{2}}Q\widehat{Q}^{-1}\widehat{E}^{\frac{1}{2}}]^{-1}$, i.e. $\widetilde{A}\widetilde{B} = \widetilde{M}$ where

$$\begin{aligned}\widetilde{A} &:= [\check{D}^{\frac{1}{2}}\widehat{P}P^{-1}\widetilde{D}^{-\frac{1}{2}}]^{-1}\check{A} \quad \text{and} \\ \widetilde{B} &:= [\check{E}^{\frac{1}{2}}\widehat{Q}Q^{-1}\widetilde{E}^{-\frac{1}{2}}]^{-1}\check{B}.\end{aligned} \tag{E.25}$$

In particular, we have

$$\begin{aligned}\|\widetilde{M}\|_* &= \max_{A,B}\left(\frac{1}{2}\left[\|A\|_{\mathrm{Fr}}^2 + \|B\|_{\mathrm{Fr}}^2\right] : AB = \widetilde{M}\right) \\ &\geqslant \frac{1}{2}\left[\|\widetilde{A}\|_{\mathrm{Fr}}^2 + \|\widetilde{B}\|_{\mathrm{Fr}}^2\right].\end{aligned} \tag{E.26}$$

Now, we can express $\check{A}$ and $\check{B}$ as $[\check{D}^{\frac{1}{2}}\widehat{P}P^{-1}\widetilde{D}^{-\frac{1}{2}}]\widetilde{A}$ and $[\check{E}^{\frac{1}{2}}\widehat{Q}Q^{-1}\widetilde{E}^{-\frac{1}{2}}]\widetilde{B}$ respectively, and thus we have

$$\begin{aligned}\|\widetilde{M}\|_* &= \frac{1}{2}\left[\|\check{A}\|_{\mathrm{Fr}}^2 + \|\check{B}\|_{\mathrm{Fr}}^2\right] \\ &= \frac{1}{2}\left[\|[\check{D}^{\frac{1}{2}}\widehat{P}P^{-1}\widetilde{D}^{-\frac{1}{2}}]\widetilde{A}\|_{\mathrm{Fr}}^2 + \|[\check{E}^{\frac{1}{2}}\widehat{Q}Q^{-1}\widetilde{E}^{-\frac{1}{2}}]\widetilde{B}\|_{\mathrm{Fr}}^2\right] \\ &\leqslant \max(\|[\check{D}^{\frac{1}{2}}\widehat{P}P^{-1}\widetilde{D}^{-\frac{1}{2}}]\|, \|[\check{E}^{\frac{1}{2}}\widehat{Q}Q^{-1}]\|)^2\frac{1}{2}\left[\|\widetilde{A}\|_{\mathrm{Fr}}^2 + \|\widetilde{B}\|_{\mathrm{Fr}}^2\right] \\ &\leqslant \max(\|[\check{D}^{\frac{1}{2}}\widehat{P}P^{-1}\widetilde{D}^{-\frac{1}{2}}]\|, \|[\check{E}^{\frac{1}{2}}\widehat{Q}Q^{-1}\widetilde{E}^{-\frac{1}{2}}]\|)^2\|\widetilde{M}\|_*.\end{aligned} \tag{E.27}$$

Hence, we need to bound the quantity $\max(\|[\check{D}^{\frac{1}{2}}\widehat{P}P^{-1}\widetilde{D}^{-\frac{1}{2}}]\|, \|[\check{E}^{\frac{1}{2}}\widehat{Q}Q^{-1}\widetilde{E}^{-\frac{1}{2}}]\|)$. Using similar notation to proposition E.1 we have $\mathcal{T}_1 = [\check{D}^{\frac{1}{2}}\widehat{P}P^{-1}\widetilde{D}^{-\frac{1}{2}}][\check{D}^{\frac{1}{2}}\widehat{P}P^{-1}\widetilde{D}^{-\frac{1}{2}}]^\top$ and $\mathcal{T}_2 = [\check{E}^{\frac{1}{2}}\widehat{Q}Q^{-1}\widetilde{E}^{-\frac{1}{2}}]$

Picking up the proof of proposition (E.1) at equation (E.16), we obtain (for all $\tau \leqslant 1$):

$$\begin{aligned}\mathbb{P}(\|[\check{D}^{\frac{1}{2}}\widehat{P}P^{-1}\widetilde{D}^{-\frac{1}{2}}]\|^2 \geqslant 1 + \tau) &\leqslant \mathbb{P}(\|\mathcal{T}_1 - I\| \geqslant \tau) \\ &\leqslant (2d_1)\exp\left(-\frac{\tau^2/2}{\nu^2 + M\tau/3}\right) \\ &\leqslant (2d_1)\exp\left(-\frac{N\tau^2/2}{[\gamma(d_1 + 2) + \gamma^2] + (\gamma + 1)\tau/3}\right) \\ &\leqslant (2d_1)\exp\left(-\frac{N\tau^2/2}{[\gamma(d_1 + 3) + \gamma^2]}\right).\end{aligned} \tag{E.28}$$

$\square$

Rewriting, this implies that with probablity greater than $1 - \delta$, we have

$$\|[\check{D}^{\frac{1}{2}}\widehat{P}P^{-1}\widetilde{D}^{-\frac{1}{2}}]\|^2 \leqslant 1 + \sqrt{\frac{2\log(\frac{2d_1}{\delta})[\gamma(d_1 + 3) + \gamma^2]}{N}}, \tag{E.29}$$

as long as $N \geqslant 2\log(\frac{2d_1}{\delta})[\gamma(d_1 + 3) + \gamma^2]$.

Similarly, (as long as $N \geqslant 2\log(\frac{2d_2}{\delta})[\gamma(d_2 + 3) + \gamma^2]$) we have (for any $\delta$) with probability $\geqslant 1 - \delta$,

$$\|[\check{E}^{\frac{1}{2}}\widehat{Q}Q^{-1}\widetilde{Q}^{-\frac{1}{2}}]\|^2 \leqslant 1 + \sqrt{\frac{2\log(\frac{2d_2}{\delta})[\gamma(d_2 + 3) + \gamma^2]}{N}}. \tag{E.30}$$

Putting the above two results together and plugging them into equation (E.27), we obtain (as long as $N \geqslant 2\log(\frac{2d}{\delta})[\gamma(d + 3) + \gamma^2]$) with probability greater than $1 - \delta$:

$$\|\widetilde{M}\|_* \leqslant \|\widetilde{M}\|_* \left[1 + \sqrt{\frac{2\log(\frac{4d}{\delta})[\gamma(d + 3) + \gamma^2]}{N}}\right], \tag{E.31}$$

as expected.

**Lemma E.3.** *Fix $M_*$ such that $\|\widetilde{M_*}\| = \|\widetilde{D}^{\frac{1}{2}}PMQ^{-1}\widetilde{E}^{\frac{1}{2}}\| = \sqrt{r_*}\Gamma \leqslant \sqrt{r}\Gamma$. Define*

$$C(S) = \max\left(0, \left\|\frac{1}{\sqrt{r_*}\Gamma}\widetilde{M_*}\right\|_* - 1\right), \tag{E.32}$$

*where $\widetilde{M} = \check{D}^{\frac{1}{2}}\widehat{P}M\widehat{Q}^{-1}\check{E}^{\frac{1}{2}}$.*

*Writing $Z_* = XM_*Y^\top = \widetilde{X}\widetilde{M_*}\widetilde{Y}^\top = \check{X}\widetilde{M_*}\check{Y}^\top$, as long as $N \geqslant 2\log(\frac{2d}{\delta})[\gamma(d + 3) + \gamma^2]$, with probability $\geqslant 1 - \delta$ over the draw of the training set:*

$$\mathbb{E}_{(i,j)\sim p}\left(l\left[(1 - C(S))[Z_*]_{i,j}, G_{(i,j)}\right] - l\left[[Z_*]_{i,j}, G_{(i,j)}\right]\right)$$

$$\leqslant \ell\|\widetilde{M_*}\|_*\left[\frac{1}{\underline{x}^2} + \frac{1}{\underline{y}^2}\right]\sqrt{\frac{2\log(\frac{4d}{\delta})[\gamma(d + 3) + \gamma^2]}{N}} \tag{E.33}$$

*Proof.* We have, writing $\Theta$ for the matrix with $\Theta_{i,j} = p_{i,j}$ and using the notation $|A|$ for the matrix obtained from $A$ by replacing each entry by its absolute value:

$$\mathbb{E}_{(i,j)\sim p}\left(l\left[(1 - C(S))[Z_*]_{i,j}, G_{(i,j)}\right] - l\left[[Z_*]_{i,j}, G_{(i,j)}\right]\right)$$

$$= \sum_{i,j}p_{i,j}l\left[(1 - C(S))[Z_*]_{i,j}, G_{(i,j)}\right] - l\left[[Z_*]_{i,j}, G_{(i,j)}\right]$$

$$\leqslant \ell\sum_{i,j}p_{i,j}\left|(1 - C(S))[Z_*]_{i,j} - [Z_*]_{i,j}\right|$$

$$\leqslant \ell C(S)\sum_{i,j}p_{i,j}\left|[Z_*]_{i,j}\right| = \ell C(S)\langle\Theta, |Z_*|\rangle = C(S)\langle\Theta, |XM_*Y^\top|\rangle$$

$$= \ell C(S)\left\langle\widetilde{\Theta}, \widetilde{X}\widetilde{M_*}\widetilde{Y}^\top\right\rangle, \tag{E.34}$$

where we write $\widetilde{\Theta}$ for the matrix with $\widetilde{\Theta}_{i,j} = \Theta_{i,j}\,\mathrm{sign}([XM_*Y^\top]_{i,j})$ for all $i, j$.

Replacing the expressions $XP^{-1}\widetilde{D}^{-\frac{1}{2}}$ and $YQ^{-1}\widetilde{E}^{-\frac{1}{2}}$ for $\widetilde{X}$ and $\widetilde{Y}$ respectively and using the circular invariance of the trace we obtain:

$$\mathbb{E}_{(i,j)\sim p}\left(l\left[(1 - C(S))[Z_*]_{i,j}, G_{(i,j)}\right] - l\left[[Z_*]_{i,j}, G_{(i,j)}\right]\right)$$

$$\leqslant \ell C(S)\left\langle\widetilde{\Theta}, \widetilde{X}\widetilde{M_*}\widetilde{Y}^\top\right\rangle = C(S)\left\langle\widetilde{\Theta}, [XP^{-1}\widetilde{D}^{-\frac{1}{2}}]\widetilde{M_*}\widetilde{E}^{-\frac{1}{2}}QY^\top\right\rangle$$

$$= \ell C(S)\left\langle\widetilde{D}^{-\frac{1}{2}}PX^\top\widetilde{\Theta}YQ^{-1}\widetilde{E}^{-\frac{1}{2}}, \widetilde{M_*}\right\rangle$$

$$\leqslant \ell C(S)\|\widetilde{M_*}\|_*\|\widetilde{D}^{-\frac{1}{2}}PX^\top\widetilde{\Theta}YQ^{-1}\widetilde{E}^{-\frac{1}{2}}\|$$

$$= \ell C(S)\|\widetilde{M_*}\|_*\left\|\left[\widetilde{D}^{-\frac{1}{2}}PX^\top A^{-1}\right]A\widetilde{\Theta}B\left[B^{-1}YQ^{-1}\widetilde{E}^{-\frac{1}{2}}\right]\right\|, \tag{E.35}$$

where $A, B$ are arbitrary invertible matrices.

Now by Lemma E.5, setting $A = \text{diag}(\|x_1\|^2, \ldots, \|x_m\|^2)$ and $B = \text{diag}(\|y_1\|^2, \ldots, \|y_n\|^2)$, we obtain:

$$\left\| \left[ \widetilde{D}^{-\frac{1}{2}} P X^\top A^{-1} \right] A \widetilde{\Theta} B \left[ B^{-1} Y Q^{-1} \widetilde{E}^{-\frac{1}{2}} \right] \right\|$$

$$\leqslant \frac{1}{2} \left\| \left[ \widetilde{D}^{-\frac{1}{2}} P X^\top A^{-1} \right] \text{diag}(A \widetilde{\Theta} B 1_n) \left[ \widetilde{D}^{-\frac{1}{2}} P X^\top A^{-1} \right]^\top \right\|$$

$$+ \frac{1}{2} \left\| \left[ B^{-1} Y Q^{-1} \widetilde{E}^{-\frac{1}{2}} \right]^\top \text{diag}(1_m^\top A \widetilde{\Theta} B) \left[ B^{-1} Y Q^{-1} \widetilde{E}^{-\frac{1}{2}} \right] \right\| \qquad \text{(E.36)}$$

$$\leqslant \frac{1}{2\underline{x}^2} \left\| \left[ \widetilde{D}^{-\frac{1}{2}} P X^\top \right] \text{diag}(\widetilde{\Theta} B 1_n) \left[ \widetilde{D}^{-\frac{1}{2}} P X^\top \right]^\top \right\|$$

$$+ \frac{1}{2\underline{y}^2} \left\| \left[ Y Q^{-1} \widetilde{E}^{-\frac{1}{2}} \right]^\top \text{diag}(1_m^\top A \widetilde{\Theta}) \left[ Y Q^{-1} \widetilde{E}^{-\frac{1}{2}} \right] \right\| \qquad \text{(E.37)}$$

$$\leqslant \frac{1}{2\underline{x}^2} \left\| \left[ \widetilde{D}^{-\frac{1}{2}} P X^\top \right] \text{diag}(\Theta B 1_n) \left[ \widetilde{D}^{-\frac{1}{2}} P X^\top \right]^\top \right\|$$

$$+ \frac{1}{2\underline{y}^2} \left\| \left[ Y Q^{-1} \widetilde{E}^{-\frac{1}{2}} \right]^\top \text{diag}(1_m^\top A \Theta) \left[ Y Q^{-1} \widetilde{E}^{-\frac{1}{2}} \right] \right\| \qquad \text{(E.38)}$$

$$= \frac{1}{2\underline{x}^2} \left\| \left[ \widetilde{D}^{-\frac{1}{2}} P X^\top \right] \text{diag}(q) \left[ \widetilde{D}^{-\frac{1}{2}} P X^\top \right]^\top \right\|$$

$$+ \frac{1}{2\underline{y}^2} \left\| \left[ Y Q^{-1} \widetilde{E}^{-\frac{1}{2}} \right]^\top \text{diag}(\kappa) \left[ Y Q^{-1} \widetilde{E}^{-\frac{1}{2}} \right] \right\| \qquad \text{(E.39)}$$

$$= \frac{1}{2\underline{x}^2} \left\| \widetilde{D}^{-\frac{1}{2}} P P^{-1} D P P^{-1} \widetilde{D}^{-\frac{1}{2}} \right\| + \frac{1}{2\underline{y}^2} \left\| \widetilde{E}^{-\frac{1}{2}} Y Q^{-1} Q E Q^{-1} Q \widetilde{E}^{-\frac{1}{2}} \right\|$$

$$\leqslant \frac{1}{\underline{x}^2} + \frac{1}{\underline{y}^2} \qquad \text{(E.40)}$$

where at line (E.36), we have used Lemma E.5 and at line (E.38) we have used that $\text{diag}(\widetilde{\Theta} B 1_n) \leqslant \text{diag}(\Theta B 1_n)$ (i.e. $\text{diag}(\Theta B 1_n) - \text{diag}(\widetilde{\Theta} B 1_n)$ is positive semi-definite).

Now, using Lemma E.2 together with equation (E.40) above plugged into equation (E.35), we finally obtain that as long as $N \geqslant 2 \log(\frac{2d}{\delta})[\gamma(d+3) + \gamma^2]$, we have with probability $\geqslant 1 - \delta$:

$$\mathbb{E}_{(i,j) \sim p} \left( l \left[ (1 - C(S))[Z_*]_{i,j}, G_{(i,j)} \right] - l \left[ [Z_*]_{i,j}, G_{(i,j)} \right] \right)$$

$$\leqslant \ell C(S) \| \widetilde{M_*} \|_* \left\| \left[ \widetilde{D}^{-\frac{1}{2}} P X^\top A^{-1} \right] A \widetilde{\Theta} B \left[ B^{-1} Y Q^{-1} \widetilde{E}^{-\frac{1}{2}} \right] \right\|$$

$$\leqslant \ell C(S) \| \widetilde{M_*} \|_* \left[ \frac{1}{\underline{x}^2} + \frac{1}{\underline{y}^2} \right]$$

$$\leqslant \ell \| \widetilde{M_*} \|_* \left[ \frac{1}{\underline{x}^2} + \frac{1}{\underline{y}^2} \right] \sqrt{\frac{2 \log(\frac{4d}{\delta})[\gamma(d+3) + \gamma^2]}{N}}, \qquad \text{(E.41)}$$

as expected. $\qquad \square$

**Lemma E.4.** *For any $r > 0$ and $\delta \in (0, 1)$, as long as $N \geqslant 8\gamma^2 + \gamma[8d + 20][\log(2d) + \log(\frac{2}{\delta})]$, we have with probability $\geqslant 1 - \delta$ over the draw of the training set:*

$$\sup_{Z \in \widetilde{\mathcal{F}}_r} \left[ |l(Z) - \hat{l}_S(Z)| \right] \leqslant \sup_{Z \in \widetilde{\mathcal{F}}_{4r\gamma^2}} \left[ |l(Z) - \hat{l}_S(Z)| \right] \qquad \text{(E.42)}$$

*Proof.* This follows from Lemma E.1 upon noticing that if $\left\| \widetilde{D}^{\frac{1}{2}} P \widehat{P}^{-1} \widecheck{D}^{-\frac{1}{2}} \right\| \leqslant \sqrt{2}$, and $XMY^\top \in \widecheck{\mathcal{F}}_r$ and $\left\| \widetilde{E}^{\frac{1}{2}} Q \widehat{Q}^{-1} \widecheck{E}^{-\frac{1}{2}} \right\| \leqslant \sqrt{2}$, and $XMY^\top \in \widecheck{\mathcal{F}}_r$:

$$\|\widetilde{M}\| = \|\widecheck{D}^{\frac{1}{2}} \widehat{P} P^{-1} \widetilde{D}^{-\frac{1}{2}} \widetilde{M} \widetilde{E}^{-\frac{1}{2}} Q \widehat{Q}^{-1} \widecheck{E}^{\frac{1}{2}}\|_* \leqslant 2\|\widetilde{M}\|. \tag{E.43}$$

Using this and the fact that $\Gamma/\widehat{\Gamma} \leqslant \gamma$ yields the result immediately. $\qquad\square$

**Lemma E.5.** *Let $U \in \mathbb{R}^{d_1 \times m}, K \in \mathbb{R}^{m \times n}, V \in \mathbb{R}^{n \times d_2}$ be matrices and let $1_m$ (resp. $1_n$) denote a column vector in $\mathbb{R}^m$ (resp. $\mathbb{R}^n$) all of whose entries are equal to 1.*

*We have the following bound on the spectral norm of $UKV$:*

$$\|UKV\| \leqslant \frac{1}{2} \left[ \|U \operatorname{diag}(K1_{d_1})U^\top\| + \|V^\top \operatorname{diag}(1_{d_2}^\top K)V\| \right]. \tag{E.44}$$

*Proof.* The result essentially follows from the Cauchy-Schwarz inequality. Indeed, let $u \in \mathbb{R}^{d_1}$ and $v \in \mathbb{R}^{d_2}$ be two unit vectors. We have, using Cauchy-Schwarz at the second line:

$$
\begin{aligned}
u^\top U K V v &= \sum_{i=1}^m \sum_{j=1}^n [u^\top U]_i K_{i,j} [Vv]_j \\
&\leqslant \sum_{i=1}^m \sum_{j=1}^n \frac{1}{2} \left[ [u^\top U]_i^2 K_{i,j} + [Vv]_j^2 K_{k,j} \right] \\
&= \frac{1}{2} u^\top U \operatorname{diag}(K1_{d_1}) U^\top u + \frac{1}{2} v^\top V^\top \operatorname{diag}(1_{d_2}^\top K) V v \\
&\leqslant \frac{1}{2} \left[ \|U \operatorname{diag}(K1_{d_1})U^\top\| + \|V^\top \operatorname{diag}(1_{d_2}^\top K)V\| \right].
\end{aligned}
\tag{E.45}
$$

Since $u$ and $v$ were arbitrary unit vectors, the result follows. $\qquad\square$

## F   Low-level lemmas

Here collect Lemmas from the literature that are useful for our proofs. Sometimes we need to prove them purely to obtain explicit constants, but everything in this section is known.

**Lemma F.1** (Non commutative Khinchine inequality [8, 9, 10]). *Let $X \in \mathbb{R}^{d \times d}$ be a matrix with jointly Gaussian, centred real-valued entries. There exists a universal constant $C_k$ such that the following bound holds on the expectation of the spectral norm of $X$:*

$$E\left(\|X\|\right) \leqslant C_k \sqrt{\log(d)} \left[ \|E(X^\top X)\|^{\frac{1}{2}} + \|E(XX^\top)\|^{\frac{1}{2}} \right] \tag{F.1}$$

Recall the following classic theorem [11, 12, 4]:

**Theorem F.1.** *Let $Z, Z_1, \ldots, Z_n$ be i.i.d. random variables taking values in a set $\mathcal{Z}$, and let $a < b$. Consider a set of functions $\mathcal{F} \in [a, b]^\mathcal{Z}$. $\forall \delta \in (0, 1)$, we have with probability $\geqslant 1 - \delta$ over the draw of the sample $S$ that*

$$\forall f \in \mathcal{F}, \quad \mathbb{E}(f(Z)) \leqslant \frac{1}{n} \sum_{i=1}^n f(z_i) + 2\mathbb{E}_S(\mathfrak{R}_S(\mathcal{F})) + (b-a)\sqrt{\frac{\log(2/\delta)}{2n}}.$$

*We also have that with probability $\geqslant 1 - \delta$, the following data-dependent bound holds:*

$$\forall f \in \mathcal{F}, \quad \mathbb{E}(f(Z)) \leqslant \frac{1}{n} \sum_{i=1}^n f(z_i) + 2\mathfrak{R}_S(\mathcal{F}) + 3(b-a)\sqrt{\frac{\log(4/\delta)}{2n}}.$$

**Proposition F.2** (Bernstein inequality, cf. [13], Corollary 2.11). *Let $X_1, X_2, \ldots, X_N$ be independent real valued random variables with the following properties for some real numbers $\nu, M$*

- $X_i \leqslant M$ *almost surely*

- $\sum_{i=1}^{N} \mathbb{E}(X_i^2) \leqslant \nu^2$.

*Let $S = \sum_{i=1}^{N} X_i - \mathbb{E}(X_i)$, we have (for all $t \geqslant 0$)*

$$\mathbb{P}(S \geqslant t) \leqslant \exp\left(-\frac{t^2/2}{\nu^2 + Mt/3}\right). \tag{F.2}$$

The inequality can be extended to the matrix-wise case as follows:

**Proposition F.3** (Non commutative Bernstein inequality, Cf. [14])**.** *Let $X_1, \ldots, X_S$ be independent, zero mean random matrices of dimension $m \times n$. For all $k$, assume $\|X_k\| \leqslant M$ almost surely, and denote $\rho_k^2 = \max(\|\mathbb{E}(X_k X_k^\top)\|, \|\mathbb{E}(X_k^\top X_k)\|)$ and $\nu^2 = \sum_k \rho_k^2$. For any $\tau > 0$,*

$$\mathbb{P}\left(\left\|\sum_{k=1}^{S} X_k\right\| \geqslant \tau\right) \leqslant (m+n)\exp\left(-\frac{\tau^2/2}{\sum_{k=1}^{S}\rho_k^2 + M\tau/3}\right). \tag{F.3}$$

**Proposition F.4.** *Under the assumptions of Proposition F.3, writing $\sigma^2 = \sum_{k=1}^{S}\rho_k^2$, we have*

$$\mathbb{E}\left(\left\|\sum_{k=1}^{S} X_k\right\|\right) \leqslant \sqrt{8/3}\sigma(1 + \sqrt{\log(m+n)}) + \frac{8M}{3}(1 + \log(m+n)). \tag{F.4}$$

*Proof.* The result in O notation is an exercise from [15], and a similar result is also mentioned in both [7] and [16].

For completeness and to get the exact constants, we include a proof as follows.

Let $Y = \left\|\sum_{k=1}^{S} X_k\right\|$. By Proposition F.3, splitting into two cases depending on whether $\tau M \leqslant \sigma^2$ or $\tau M \geqslant \sigma^2$ we have

$$\mathbb{P}(Y \geqslant \tau) \leqslant \min\left(1, (m+n)\exp\left[-\frac{3\tau^2}{8\sigma^2}\right]\right) + \min\left(1, (m+n)\exp\left[-\frac{3\tau}{8M}\right]\right) \tag{F.5}$$

Now note that writing $\kappa$ for $\log(m+n)8M/3$, we have

$$\int_0^\infty 1 \wedge (m+n)\exp\left(-\frac{3\tau}{8M}\right)d\tau \tag{F.6}$$

$$\leqslant \int_0^\kappa 1 \wedge (m+n)\exp\left(-\frac{3\tau}{8M}\right)d\tau + \int_\kappa^\infty (m+n)\exp\left(-\frac{3\tau}{8M}\right)d\tau$$

$$\leqslant \kappa + \left[\frac{-8M}{3}(m+n)\exp\left(-\frac{3\tau}{8M}\right)\right]_\kappa^\infty = \kappa + \frac{8M(m+n)}{3}\exp\left(-\frac{3\kappa}{8M}\right)$$

$$= \kappa + \frac{8M(m+n)}{3} = \frac{8M}{3}(1 + \log(m+n)). \tag{F.7}$$

We also have, writing $\psi$ for $\sigma\sqrt{\log(m+n)8/3}$,

$$\int_0^\infty 1 \wedge (m+n)\exp\left(-\frac{3\tau^2}{8\sigma^2}\right)d\tau \leqslant \int_0^\psi 1\,d\tau + \int_\psi^\infty (m+n)\exp\left(-\frac{3\tau^2}{8\sigma^2}\right)d\tau$$

$$\leqslant \psi + \int_\psi^\infty \exp\left(-\frac{3(\tau^2 - \psi^2)}{8\sigma^2}\right)d\tau \leqslant \psi + \int_\psi^\infty \exp\left(-\frac{3(\tau - \psi)^2}{8\sigma^2}\right)d\tau$$

$$\leqslant \psi + \sigma\sqrt{2\pi/3} = \sigma\left[\sqrt{\log(m+n)8/3} + \sqrt{2\pi/3}\right] \leqslant \sqrt{8/3}\sigma(1 + \sqrt{\log(m+n)}). \tag{F.8}$$

Plugging inequalities (F.6) and (F.8) into equation (F.5), we obtain:

$$\mathbb{E}(Y) \leqslant \int_0^\infty \mathbb{P}(Y \geqslant \tau)d\tau \leqslant \sqrt{8/3}\sigma(1 + \sqrt{\log(m+n)})\frac{8M}{3}(1 + \log(m+n)), \tag{F.9}$$

as expected. $\qquad\square$

**Lemma F.5.** *Let $F$ be a random variable that depends only on the draw of the training set. Assume that with probability $\geqslant 1 - \delta$,*

$$\mathbb{E}(F) \leqslant f(\delta), \tag{F.10}$$

*for some given monotone increasing function $f$. Then we have, in expectation over the training set:*

$$\mathbb{E}(F) \leqslant \sum_{i=1}^{\infty} f(2^{-i}) 2^{1-i}, \tag{F.11}$$

*In particular, if $f(\delta) = C_1 \sqrt{\log(\frac{1}{\delta})} + C_2$, then we have in expectation over the draw of the training set:*

$$\mathbb{E}(F) \leqslant \frac{C_1}{\sqrt{2} - 1} + C_2. \tag{F.12}$$

*Proof.* By assumption we have for any $\delta$:

$$\mathbb{P}\left(X \geqslant f(\delta)\right) \leqslant \delta \tag{F.13}$$

Let us write $A_i$ for the event $A_i = \{F \leqslant f(\delta_i)\}$ where we set $\delta_i = 2^{-i}$ for $i = 1, 2, \dots$ . We also set $\tilde{A}_i = A_i \backslash A_{i-1}$ for $i = 1, 2, \dots$ with the convention that $A_0 = \varnothing$ so that $\tilde{A}_1 = A_1$.

We have, for $i \geqslant 2$, $\mathbb{P}(\tilde{A}_i) \leqslant \mathbb{P}(A_{i-1}^c) \leqslant \delta_{i-1}$, and for $i = 1$, $\mathbb{P}(\tilde{A}_1) \leqslant 1 = \delta_{i-1}$. Thus we can write

$$\mathbb{E}(F) \leqslant \sum_{i=1}^{\infty} \mathbb{E}(X|\tilde{A}_i) \mathbb{P}(\tilde{A}_i) \leqslant \sum_{i=1}^{\infty} \mathbb{E}(X|\tilde{A}_i) \delta_{i-1} \leqslant \sum_{i=1}^{\infty} f(\delta_i) \delta_{i-1}, \tag{F.14}$$

yielding identity (F.11) as expected.

Next, assuming $f(\delta) = C_1 \sqrt{\log(\frac{1}{\delta})} + C_2$, we can continue as follows:

$$\mathbb{E}(F - C_2) \leqslant \sum_{i=1}^{\infty} f(\delta_i) \delta_{i-1} \leqslant \sum_{i=1}^{\infty} [C_1 \sqrt{\log(2^i)}] 2^{1-i} \tag{F.15}$$

$$\leqslant \sum_{i=1}^{\infty} [C_1 \sqrt{i}] 2^{1-i} \leqslant C_1 \sum_{i=1}^{\infty} \sqrt{2}^{1-i} = \frac{C_1}{\sqrt{2} - 1} \tag{F.16}$$

where at the second line we have used the fact that for any natural number $i$, $\sqrt{i} \leqslant \sqrt{2}^{i-1}$. $\qquad\square$

As an immediate consequence we obtain the following Rademacher type theorem in expectation:

**Theorem F.2.** *Let $Z, Z_1, \dots, Z_N$ be i.i.d. random variables taking values in a set $\mathcal{Z}$, and let $a < b$. Consider a set of functions $\mathcal{F} \in [a, b]^{\mathcal{Z}}$. $\forall \delta \in (0, 1)$, we have in expectation over the draw of the sample $S$ that*

$$\inf_{f \in \mathcal{F}} \left( \mathbb{E}(f(Z)) - \frac{1}{N} \sum_{i=1}^{n} f(z_i) \right) \leqslant 2\mathbb{E}(\mathfrak{R}_S(\mathcal{F})) + 5(b - a)\sqrt{\frac{1}{N}}. \tag{F.17}$$

**Proposition F.6** ( [17, 18, 19]). *Let $\mathcal{F}$ be a real-valued function class taking values in $[0, 1]$, and assume that $0 \in \mathcal{F}$. Let $S$ be a finite sample of size $n$. For any $2 \leqslant p \leqslant \infty$, we have the following relationship between the Rademacher complexity $\mathfrak{R}(\mathcal{F}|_S)$ and the covering number $\mathcal{N}(\mathcal{F}|S, \epsilon, \|\cdot\|_p)$.*

$$\mathfrak{R}(\mathcal{F}|_S) \leqslant \inf_{\alpha > 0} \left( 4\alpha + \frac{12}{\sqrt{n}} \int_{\alpha}^{1} \sqrt{\log \mathcal{N}(\mathcal{F}|S, \epsilon, \|\cdot\|_p)} d\epsilon \right),$$

*where the norm $\|\cdot\|_p$ on $\mathbb{R}^m$ is defined by $\|x\|_p^p = \frac{1}{n}(\sum_{i=1}^{m} |x_i|^p)$.*

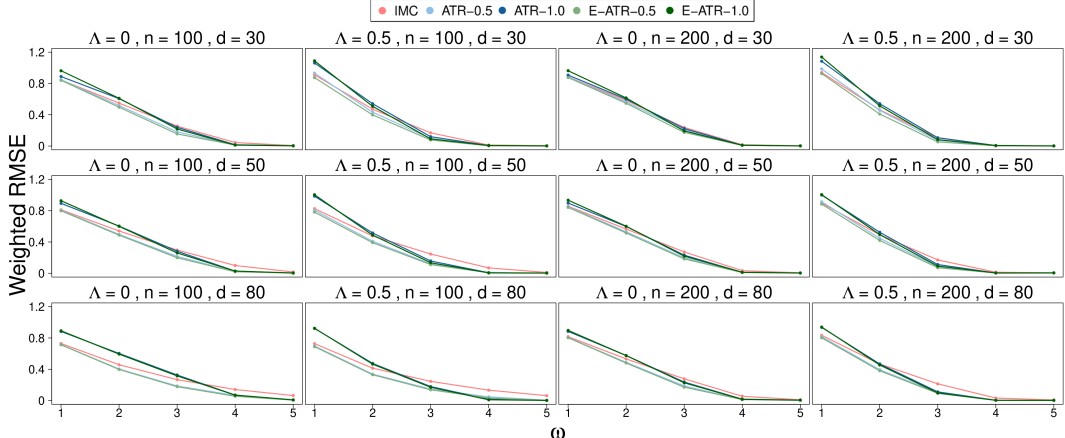

Figure G.1: Weighted RMSE as a function of $\omega$ .

# G   More detailed discussion of the experimental setting

## G.1   Synthetic data

**Generation and training procedure:** First we sample matrices $A$ and $B$ in $\mathbb{R}^{m \times d}$ with i.i.d. $N(0, 1)$ entries. We also sample $K_1$ and $K_2$ in $\mathbb{R}^{d \times r}$. We then compute $F = AK_1K_2^{\top}B^{\top}$ and set $G = m \, \mathrm{normalize}(F)$, $X = \sqrt{md} \, \mathrm{normalize}(A)$ and $Y = \sqrt{md} \, \mathrm{normalize}(B)$ where the operator normalize normalises the matrix to have unit Frobenius norm. Regarding the sampling distribution, we set $p_{i,j} \propto \exp(\Lambda |G_{i,j}|)$ where $\Lambda$ is a hyperparameter. In particular, when $\Lambda = 0$ we have uniform sampling. For each $n \in \{100, 200\}$ we evaluate the following $(d, r)$ combinations: $(30, 4)$, $(50, 6)$ and $(80, 10)$. In order to study a meaningful data-sparsity regime, in each case we sampled $dr\omega$ entries where $\omega \in \{1, 2, 3, 4, 5\}$. Each $(n, d, r)$ configuration was tested on 50 matrices. Training details: the $\lambda$s were chosen in the range $[10^{-6}, 2 \times 10^2]$, each configuration was run to convergence without warm starts.

**More detailed results:** Below are detailed results of the synthetic data experiments. The first graph G.1 shows the performance as a function of our data sparsity paramameter $\omega$ in different configurations, whilst Figure G.3 provides the corresponding boxplots documenting the variance with respect to the draw of the random matrix. Figure G.2 shows, in many different situations, the progression of performance as the size of the side information increases. Corresponding boxplots are provided in Figure G.4.

We observe that our methods (especially the smoothed version) generally outperform standard IMC in the meaningful sparsity regimes. Interestingly, when data is too sparse to make any meaningful prediction, standard IMC frequently outperforms our method (though our methods become better as more data becomes available), suggesting that $\alpha$ could be tuned depending on the sparsity of the observations.

## G.2   Description of real-life datasets

- **Douban**[2] ($R \in \mathbb{R}^{4999 \times 4577}$): Douban is a social network where users can produce content related to movies, music, and events. Douban users are members of the social network and Douban items are a subset of popular movies. The rating range is $\{1, 2, \ldots, 5\}$ and the entry $(i, j)$ corresponds the rating of user $i$ to movie $j$. To construct side information, we collected the following data from the Douban website: each movies' genres, its number of views, the number of people who rated the movie, and the number of reviews written.

- **LastFM** ($R \in \mathbb{R}^{1875 \times 4354}$): Last.fm is a British music website that builds a detailed profile of each user's musical taste. Differently from the other datasets an entry $(i, j)$ represents the number of views of user $i$ to band/artist $j$. We expressed the number of views in a log scale.

---

[2]Rating matrix available in https://doi.org/10.7910/DVN/JGH1HA

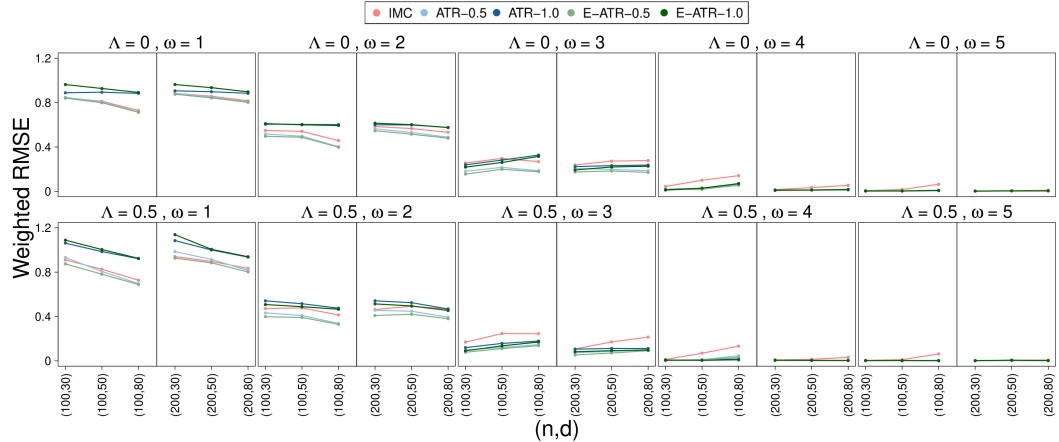

Figure G.2: Weighted RMSE as a function of the size of the side information.

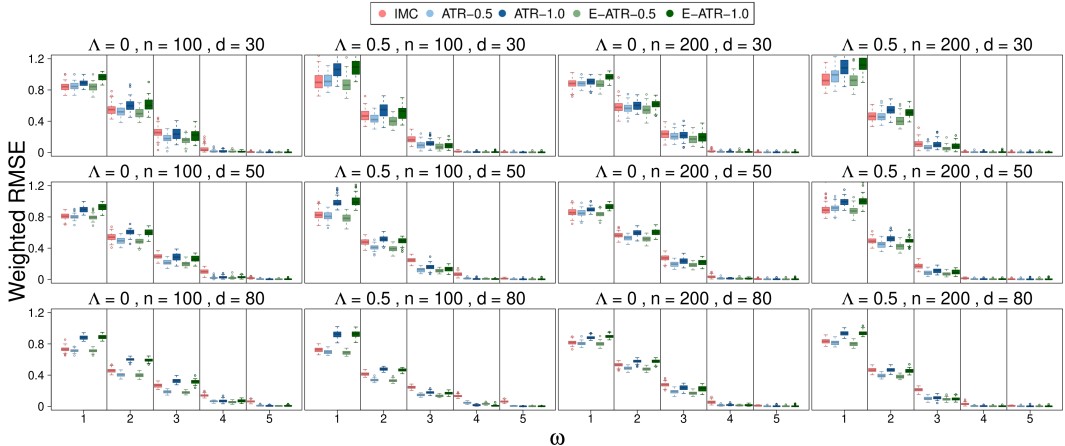

Figure G.3: Weighted RMSE as a function of $\omega$, boxplots.

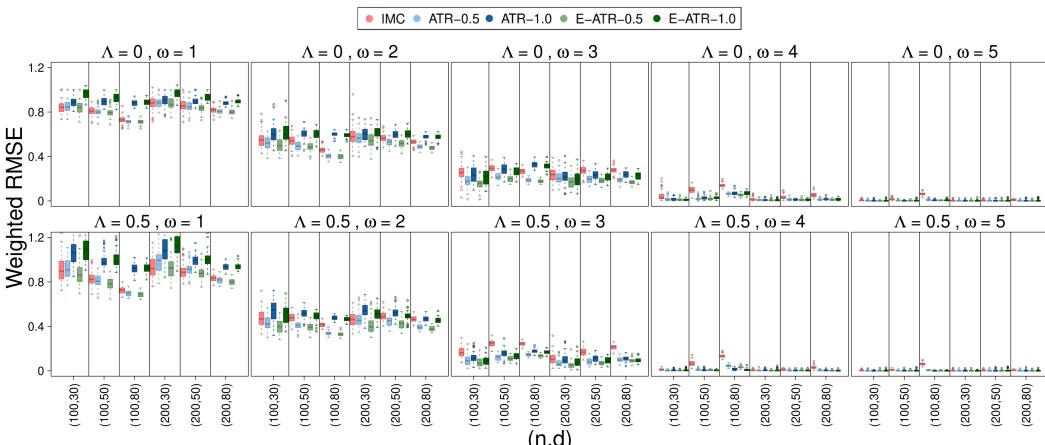

Figure G.4: Weighted RMSE as a function of the size of the side information, boxplots.

The website allows users to tag artists, which provides us with the opportunity to group the items (artists) by their associated tags.

- **MovieLens** ($R \in \mathbb{R}^{6040 \times 3382}$): We consider the MovieLens 1M dataset, which is a broadly used and stable benchmark dataset. MovieLens is a non-commercial website for movie recommendations. Just as in Douban, an entry $(i, j)$ represents the rate of user $i$ to movie $j$ on a scale from 1 to 5. We used movies' genres and gender as item and user side information respectively.

**Training details:** In all real data experiments, we used 85% of the data for training, 10% for validation and 5% for the test set.

We optimized the model (18) via the accelerated subgradient method of [20], alternating the optimization between each term with only two iterations per term.

To choose a suitable hyper parameter range, the matrices $\breve{X}$ and $\breve{Y}$ were normalised to have Frobenius norm $\sqrt{m}$ and $\sqrt{n}$ respectively, and values in the range $[1, 200]$ were explored for both $\lambda_1, \lambda_2$. Initially, twenty alternations were run for each tested hyper parameter combination. We then ran the model to convergence for the final hyperparameter configuration. For the real data experiments, we used a rank-restricted version of the SVD's with rank 30.

We performed the experiments in a cluster with 72 CPUs (3GHz) and 750GB of RAM. We relied on warm starts to reach convergence faster. For a given $X, Y$, and given a solution $Z_1 + Z_2$ (with $Z_1$ (resp. $Z_2$) corresponding to the inductive (resp. non inductive) term), a warm start $X M_0 Y^\top + Z_0$ can be constructed as follows: Set $Z_0 = Z_2$. Set $M_0 = (X^\top X)^{-1} X^\top Z_1 Y (Y^\top Y)^{-1}$. If $X$ or $Y$ is not full rank the above inverses can be replaces by pseudoinverses.

# H   Variations on the optimization problems and loss functions

**Models involving a non-inductive term** We first note that using the subadditivity of the Rademacher complexity, it is trivial to obtain results for a combined function class corresponding to the regulariser (18):

**Proposition H.1.** *Suppose for simplicity that* $m = n$, $d_1 = d_2 = d$, *and* $\frac{\mathbf{x}^2 \mathbf{y}^2}{\underline{x}^2 \underline{y}^2} = \gamma \leqslant K$ *for some constant* $K = O(1)$ *and define the function class* $\widetilde{\mathcal{G}}_{r_1, r_2} := \{ XMY^\top + Z : \| \widetilde{D}^{\frac{1}{2}} PMQ^{-1} \widetilde{E}^{\frac{1}{2}} \|_* \leqslant \Gamma \sqrt{r_1} \ \wedge \ \| \widetilde{D}_I^{\frac{1}{2}} Z \widetilde{E}_I^{\frac{1}{2}} \|_* \leqslant \sqrt{r_2} \}$. *As long as* $N \geqslant T$ *where* $T$ *is* $O(n)$, *w.p.* $\geqslant 1 - \delta$ *we have for all* $F \in \widetilde{\mathcal{G}}_{r_1, r_2}$:

$$l(F) - \hat{l}_S(F) \leqslant \widetilde{O}\left( (\ell + b) \frac{\sqrt{\Gamma r_1 d} + \sqrt{r_2 n}}{\sqrt{N}} \right). \tag{H.1}$$

*Proof.* Follows from the Rademacher complexity bound from Proposition 3.3 (cf. also Prop.B.1) applied to both side information pairs $(X, Y)$ and $(I, I)$, together with the subadditivity of the Rademacher complexity. Note that the condition on $n$ is only necessary to get rid of $O(1/N)$ terms for cosmetic purposes. □

**Lagrangian Formulation and Square Loss**

Similarly to other work ([21, 2] etc.) we expressed our results in terms of bounds on the expected loss of the empirical risk minimizers subject to *explicit norm constraints*. However, it is easy to express similar results for the solution to a regularised optimization problem in "Lagrangian formulation" such as the ones we propose[3]. We have also relied on a *bounded* loss function. However, in most practical situations, the values of the entries are restricted by domain knowledge (for instance, in the Recommender Systems field, ratings are typically restricted to the range $[1, 5]$). This effectively renders any Lipschitz loss bounded, including the square loss, as long as one also truncates the output of the algorithm to fit the required range.

We begin by completing the (trivial) proof of Corollary 3.4.

---

[3]just as in the case of exact norm constraints, the hyperparameters must be assumed to have been properly tuned

*Proof of Corollary 3.4.* Since $G$ satisfies the optimization constraints on the training set $\Omega$, we must have $\|\widetilde{D}^{\frac{1}{2}}PM_{\#}Q^{-1}\widetilde{E}^{\frac{1}{2}}\|_* \leq \sqrt{r_G}\Gamma$, which allows us to apply proposition (C.1) to the loss function $\Phi_{2C} \circ l$, which coincides with $l$ when applied to the matrix $\Phi_C(Z_{\#}) - G$. $\qquad\square$

We now show further how to adapt our result C.1 to make it apply to the solution to a Lagrangian formulation involving the square loss. Further similar manipulations can be applied to our other results.

**Proposition H.2.** *Assume that the noise $\zeta$ is bounded by a fixed constant $C$ almost surely, and that so are all of the entries of the ground truth matrix $G$. Let*

$$M_\lambda = \arg\min_M \frac{1}{N} \sum_{(i,j)\in\Omega} [XMY^\top - G_{i,j} - \zeta_{i,j}]^2_{i,j} + \lambda\|\widetilde{D}^{\frac{1}{2}}PMQ^{-1}\widetilde{E}^{\frac{1}{2}}\|_*,$$

*and $Z_\lambda := XM_\lambda Y^\top$ denote the solutions to a Lagrangian formulation of the problem with the square loss $l$ (which is unbounded).*

*Furthermore, we also write $\Phi(x) = \Phi_{2C}(x) = \text{sign}(x)\min(|x|, 2C)$, $\mathcal{E} = l(G)$ for the expected square loss at the ground truth (i.e. the variance of the noise) and $\Delta := C^2\sqrt{\frac{\log(4/\delta)}{2N}}$. We assume that $\lambda$ is tuned so that $\frac{\mathcal{E}+\Delta}{2\sqrt{r_G}\Gamma} \leq \lambda \leq 2\frac{\mathcal{E}+\Delta}{\sqrt{r_G}\Gamma}$[4].*

*We have the following bound on the expected $L^2$ risk of $\Phi_{2C}(XM_\lambda Y^\top)$:*

$$\mathbb{E}_\xi\left(\left|\Phi_{2C}[XM_\lambda Y^\top]_\xi - G_\xi - \zeta_\xi\right|^2\right) = l(\Phi_{2C}(XM_\lambda Y^\top)) \tag{H.2}$$

$$\leq 3\mathcal{E} + \frac{48\ell\sqrt{\Gamma}\sqrt{r}\sqrt{d}(1+\sqrt{\log(2d)})}{\sqrt{N}} + \frac{72\ell\mathbf{xy}\sqrt{d_1 d_2 r}(1+\log(2d))}{N} + 19C^2\sqrt{\frac{\log(4/\delta)}{2N}}.$$

*Proof.* Since $l(\zeta) \leq C^2$ for any $|\zeta| \leq C$ we have by Hoeffding's lemma that with probability $\geq 1 - \delta/2$,

$$|l_S(G) - l(G)| \leq C^2\sqrt{\frac{\log(4/\delta)}{2N}}. \tag{H.3}$$

Then (with the same probability) we have

$$l_S(Z_\lambda) + \lambda\|\widetilde{D}^{\frac{1}{2}}PM_\lambda Q^{-1}\widetilde{E}^{\frac{1}{2}}\|_* \leq l_S(G) + \lambda\sqrt{r_G}\Gamma$$

$$\leq l(G) + \lambda\sqrt{r_G}\Gamma + C^2\sqrt{\frac{\log(4/\delta)}{2N}}.$$
$$= l(G) + \Delta + \lambda\sqrt{r_G}\Gamma$$
$$\leq 3[\mathcal{E} + \Delta], \tag{H.4}$$

where at the last line, we have used the constraint on $\lambda$.

It follows that

$$\|\widetilde{D}^{\frac{1}{2}}PM_\lambda Q^{-1}\widetilde{E}^{\frac{1}{2}}\|_* \leq \frac{3[\mathcal{E}+\Delta]}{\lambda}$$
$$\leq 6\sqrt{r_G}\Gamma, \tag{H.5}$$

where we have made another use of the constraint on $\lambda$.

It follows that $Z_\lambda \in \widetilde{\mathcal{F}}_{36r_G}$. Let $\tilde{l} = \Phi_{4C} \circ l$ be the truncated square loss: $\tilde{l}(a,b) = \min(|a-b|, 4C)^2$. By Proposition C.1 we now have with probability $\geq 1 - \frac{\delta}{2}$ over the draw of the training set:

$$\mathbb{E}\left[\tilde{l}((XM_\lambda Y^\top)_\xi, G_\xi + \zeta_\xi)\right] - \frac{1}{N}\sum_{\xi\in\Omega}\tilde{l}((XM_\lambda Y^\top)_\xi, G_\xi + \zeta_\xi) \tag{H.6}$$

---
[4]Although this tuning depends on the sample size $N$ slightly, it converges as $N$ tends to infinity and is there for purely cosmetic purposes (to avoid extra logarithmic terms in the final formula).

$$\leqslant \frac{48\ell\sqrt{\Gamma}\sqrt{r}\sqrt{d}(1+\sqrt{\log(2d)})}{\sqrt{N}} + \frac{72\ell\mathbf{xy}\sqrt{d_1 d_2 r}(1+\log(2d))}{N} + 16C^2\sqrt{\frac{\log(4/\delta)}{2N}}.$$

Writing $\bar{\Delta}$ for the quantity

$$\frac{48\ell\sqrt{\Gamma}\sqrt{r}\sqrt{d}(1+\sqrt{\log(2d)})}{\sqrt{N}} + \frac{72\ell\mathbf{xy}\sqrt{d_1 d_2 r}(1+\log(2d))}{N} + 16C^2\sqrt{\frac{\log(4/\delta)}{2N}},$$

it now follows that (w.p. $\geqslant 1-\delta$)

$$
\begin{aligned}
l(\Phi_{2C}(XM_\lambda Y^\top)) &= \mathbb{E}_\xi\left[l((\Phi_{2C}(XM_\lambda Y^\top)_\xi, G_\xi + \zeta_\xi)\right] \\
&= \mathbb{E}_\xi\left[\tilde{l}((XM_\lambda Y^\top)_\xi, G_\xi + \zeta_\xi)\right] = \tilde{l}((XM_\lambda Y^\top)) \\
&\leqslant \tilde{l}_S(XM_\lambda Y^\top) + \bar{\Delta} \quad\quad\quad\quad\quad\quad\quad\quad\quad\quad\quad (\text{H.7}) \\
&\leqslant l_S(XM_\lambda Y^\top) + \bar{\Delta} \leqslant 3[\mathcal{E} + \Delta] + \bar{\Delta}, \quad\quad\quad (\text{H.8})
\end{aligned}
$$

where at equation (H.7) we have used equation (H.6) and at equation (H.8) we have used equation (H.4). The result follows.

$\square$

# I Further discussion

## I.1 Deeper comparison to related works

Here we discuss some related works in more detail than in the main paper.

One very interesting other work is [22] which introduces a joint model that imposes a nulcear norm based constraint on both $M$ and $XMY^\top$ through a modification of the objective: first, the matrices $X$ and $Y$ are augmented by columns of ones resulting in the matrices $\bar{X} = [X, \mathbf{1}]$ and $\bar{Y} = [Y, \mathbf{1}]$. Predictors then take the form $E = \bar{X}M(\bar{Y})^\top + \Delta$, with nuclear norm regularisation imposed on both $E$ and $M$, and Frobenius norm regularization imposed on $\Delta$, with the constraint that $P_\Omega(E) = R_\Omega$ where $R_\Omega$ denotes the observed entries. Thus the model achieves a similar aim as [21] through a different and more original approach. The authors then provide an efficient algorithm for their model and prove some theoretical guarantees: for exact recovery, they obtain a rate of $O(rd\log(d)\log(n))$ in the uniform sampling case. This is the same as [20], except that the assumptions on $X$ and $Y$ are weaker (no orthogonality assumption). Of course, both [22] and [20] require a realisability assumption for exact recovery to be possible. In addition to that, the authors of [22] also show distribution-free bounds for the approximate recovery case which scale as $O(\gamma^2 \log(n))$ where $\gamma$ is an upper bound on the ground truth spectral norm of the matrix $M$ ($G$ in their notation). That bound is comparable to the bounds of the form (3) from [21, 23, 24], though the precise results are different in formulation (and rely on a different optimizer). Note that in addition to pertaining to a completely different optimization problem, our results for approximate recovery lack any dependence on $n$, even logarithmic, and also do not have the implicit dependence on $d_1 d_2$ present in that paper. Note that although it is claimed in the paper that the rate is "$\log(n)$", this is because in that informal presentation of the results the authors are treating their "$\gamma$" (which scales at least as $\sqrt{d_1 d_2 r}$) as a constant, which amounts to treating the size of the side information as a constant. This type of formulation is standard and also used in [20], but corresponds to a different perspective as in this work we want to remove the dependence on $d_1, d_2$. Note also that although it is not explicitly stated in the paper that the exact recovery results rely on a uniform sampling assumption, such an assumption is implicit. Indeed, such an assumption is standard in all exact recovery results: there is no known exact recovery result for arbitrary distributions for either MC or IMC. Further, the results would be clearly wrong without such an assumption (assume for instance identity side information and a sampling distribution which only samples the top left quadrant, all of which is perfectly compatible with the coherence assumptions on $X, Y$ and the ground truth matrix $G$ ($F$ in their notation)). The first obvious implicit use of the uniform sampling assumption is in line 70 of the supplementary material. As we explain later, even defining the concept of exact recovery in the non uniform sampling case has not been done explicitly to the best of our knowledge, and no results exist for this for either inductive matrix completion or matrix completion in general.

In [25], the authors explicitly study a disentangled version of [21] specifically tailored to the case of community side information. Whilst generalisation bounds are provided which scale similarly to ours in the case of community side information, those are obtained through a direct application of the matrix completion results from [2] to the auxiliary problem where each community is treated as a single user. In particular, the results are not applicable in a more general context and they did not introduce any of the novel proof techniques we rely on here.

[26] proves rates of $d^2 r^3 \log(d)$ in the case of *exact recovery*, as well as abstract conditions for the possibility of exact recovery in a more general context and results for other problems closely related to inductive matrix completion (such as matrix regression, see also [27, 28]); [29], which proved a similar sample complexity rate together with an efficient optimization strategy with favourable convergence rates; and of course [20], which *both* introduced the MaxIDE algorithm (an involved form of projected gradient method with an integrated line search over the step sizes) to solve problem (2), *and* proved sample complexity bounds of order $rd \log(d) \log(n)$ for exact (noiseless) recovery under the assumption of uniform sampling. Recently, convergence and generalisation guarantees were shown for an exciting model which functions as inductive matrix completion with unknown "side information matrices $X, Y$ which must be learned by a two layer neural network from some raw user and item side information, jointly with the low rank problem [30]. We note that this applies to a fixed rank problem and does not rely on a nuclear norm regulariser.

**Further remarks on related works:** In Table 1 and Table 2, we are only concerned with sample complexity. It is worth noting that many important gains were also achieved in the direction of improving computational complexity through better algorithms [29, 31].

We also do not compare here with results obtained for other regularisation strategies including the max norm [32, 33, 34] etc., all of which apply exclusively to matrix completion without side information. We do note in passing that rates of $O(nr \log(n))$ were obtained very early for matrix completion with an *explicit low-rank assumption* [32]. In both MC and IMC, the relevance of the more recent branch of the literature is tied to the impractical nature of explicitly minimizing the rank and the fact that the low rank assumption is not satisfied *exactly*, justifying the use of nuclear norm based methods and the soft relaxations of the rank that they bring into the theoretical analysis.

## J   Discussion and future directions

### J.1   On transductive Rademacher complexity:

Some results in [2] and [7] are formulated in the transductive [35] setting. In this context, we assume that the set of observed entries is sampled *without replacement*, and the training and test sets are divided uniformly. There is a parallel theory in this case with a concept of transductive Rademacher complexity at the key. In some cases the bounds can be better in some aspects. For instance, the transductive bound in [2] scales like $O(nr \log(n))$ in the case of a distribution where the probabilities of each entries are within a ratio of each other. Such a bound follows in our iid setting from Proposition 3.3, and indeed similar results had been otherwise obtained (for the non inductive case) in [32], as the authors of [2] mention. As another significant advantage, the *transductive* bounds in [7] involve a smaller power of the log term.

There are two reasons why we didn't prove transductive bounds in our setting: (1) The transductive Rademacher complexity is bounded above by the standard Rademacher complexity up to a constant of 4 [5]. *In particular, all of our results also hold up to a constant in a transductive setting.* [6]. (2) Contrary to the MC case, we do not believe that we would get better bounds in this context. Indeed, the main reason the transductive setting improves the bounds is because it prevents the oversampling of single entries (see how in the proof of the main theorem in [2], one must distinguish between the oversampled entries and the moderately sampled entries). It is easy to see by comparing to our proof of Theorem 3.1, especially consolidating the intuition via the example of community side information, that the benefits would *not* carry over to the inductive case: even if the *entries* are sampled without replacement, the *combinations of communities* can still be sampled many times. Thus we do not expect significant gains from this approach.

---

[5]See Footnote 1 on page 3407 of [2], and Lemma 1 in [35]

[6]This remark also applies to earlier work, they merely proved the transductive bounds because in the matrix case, this provides an actual improvement.

## J.2 Open directions

There are many possible open problems related to this work and to distribution-free matrix completion in general:

- Is it possible to provide a rigorous theoretical explanation why the empirically weighted trace norm outperforms the exactly weighted version in the synthetic data experiments?

- Can we make the bounds even more sensitive to the alignement of the side information vectors?

- In what situations can one remove the $\sqrt{\log(d)}$ term in Proposition 3.1?

Regarding the extra log term in Theorem 3.1, we would like to note that although we do not see how to remove it in general, it is straightforward to remove it (at the cost of higher order dependence on the coherence of $X$ and $Y$) in the specific case where the columns of $X$ and $Y$ each have distinct support (i.e. the columns of $X^2$ and $Y^2$, defined as matrices whose entries are the squares of those of $X$ and $Y$ respectively, are orthogonal), in which particular case a proof with more similarities to that in [2] still holds.

# K Table of notations

Table K.1: Table of notations for quick reference

| Notation | Meaning |
|---|---|
| $\|A\|$ | spectral norm of matrix $A$ |
| $A \preccurlyeq B$ | $B - A$ is positive semi-definite |
| $\|A\|_*$ | nuclear norm of matrix $A$ |
| $I$ | Identity matrix |
| $G \in \mathbb{R}^{m \times n}$ | ground truth matrix |
| $\xi_1, \ldots, \xi_N$ $(\in \{1, \ldots, m\} \times \{1, \ldots, n\})$ | sampled entries |
| $\zeta_\xi$ | Noise observed at sample $\xi$ |
| $X \in \mathbb{R}^{m \times d}$ (resp. $Y \in \mathbb{R}^{n \times d}$) | Row (resp. column) side information matrix |
| $M$ | matrix to optimize (predictors: $XMY^\top$) |
| $S = \Omega = \{\xi_1, \ldots, \xi_N\}$ | (training) set of observed entries |
| $x_i = X_{i,\cdot}$ | side information vector for $i$th user (row) |
| $y_j = X_{j,\cdot}$ | side information vector for $j$th item (column) |
| $\mathbf{x}$ (resp. $\mathbf{y}$) | $\max_i \|x_i\|^2$ (resp. $\max_j \|x_j\|^2$) |
| $\underline{x}$ (resp. $\underline{y}$) | $\min_i \|x_i\|^2$ (resp. $\min_j \|x_j\|^2$) |
| $\gamma$ | $\frac{\mathbf{x}^2 \mathbf{y}^2}{\underline{x}^2 \underline{y}^2}$ |
| $d$ | $\max(d_1, d_2)$ |
| $p_{i,j}$ | Probability of sampling $(i,j)$ $=\mathbb{P}(\xi = (i,j))$ |
| $p$ | sampling distribution |
| $\mathcal{M}$ | constraint on $\|M\|_*$ |
| $h_{i,j} = \sum_{\xi \in \Omega} 1_{\xi = (i,j)}$ | Number of times entry $(i,j)$ was sampled |
| $l$ | loss function |
| $b$ | global upper bound on $l$ |
| $\ell$ | Lipschitz constant of $l$ |
| $l(Z)$ | $\mathbb{E}_{(i,j) \sim p}(l([XMY^\top]_{i,j}, G_{i,j} + \zeta_{i,j}))$ |
| (or more rigorously) | $\mathbb{E}_{\xi, \bar{\xi}} l([XMY^\top]_{\xi_1, \xi_2}, \bar{\xi}_o)$ |
| $\hat{l}(Z)$ | $\frac{1}{N} \sum_{(i,j) \in \Omega} l([XMY^\top]_{i,j}, G_{i,j} + \zeta_{i,j})$ |
| (or more rigorously) | $\frac{1}{N} \sum_{o=1}^{N} l([XMY^\top]_{\xi_1, \xi_2}, \bar{\xi}_o)$ |
| $\Gamma$ | $\sum_{i,j} p_{i,j} \|x_i\|^2 \|y_j\|^2$ |
| $\widehat{\Gamma}$ | $\frac{1}{N} \sum_{i,j} h_{i,j} \|x_i\|^2 \|y_j\|^2$ |
| $q_i$ (resp. $\hat{q}_i$) | $\sum_{j=1}^{n} p_{i,j} \|y_j\|^2$ (resp. $\frac{1}{N} \sum_{j=1}^{n} h_{i,j} \|y_j\|^2$) |
| $\kappa_j$ (resp. $\hat{q}_i$) | $\sum_{i=1}^{m} p_{i,j} \|x_i\|^2$ (resp. $\frac{1}{N} \sum_{i=1}^{m} h_{i,j} \|x_i\|^2$) |
| $\langle v, w \rangle_l$ (resp. $\langle v, w \rangle_r$) | $\sum_{i=1}^{m} v_i q_i w_i$ (resp. $\sum_{j=1}^{n} v_j h_j w_j$) |
| $\langle v, w \rangle_{\hat{l}}$ (resp. $\langle v, w \rangle_{\hat{r}}$) | $\sum_{i=1}^{m} v_i \hat{q}_i w_i$ $\sum_{j=1}^{n} v_j \hat{\kappa}_j w_j$ |
| $L$ | $X^\top \operatorname{diag}(q) X = \sum_{i,j} p_{i,j} x_i x_i^\top \|y_j\|^2$ |
| $\widehat{L}$ | $X^\top \operatorname{diag}(\hat{q}) X = \sum_{i,j} \frac{h_{i,j}}{N} x_i x_i^\top \|y_j\|^2$ |
| $R$ | $Y^\top \operatorname{diag}(\kappa) Y = \sum_{i,j} p_{i,j} y_j y_j^\top \|x_i\|^2$ |
| $\widehat{R}$ | $Y^\top \operatorname{diag}(\hat{\kappa}) Y = \sum_{i,j} \frac{h_{i,j}}{N} y_j y_j^\top \|x_i\|^2$ |
| $D$ (resp. $\widehat{D}$) | Eigenvalues of $L$ (resp. $\widehat{L}$) |
| $E$ (resp. $\widehat{E}$) | Eigenvalues of $R$ (resp. $\widehat{R}$) |
| $P$ | orth. matrix diagonalising $L$ so $L = P^{-1} D P$ |
| $Q$ | orth. matrix diagonalising $R$ so $R = Q^{-1} E Q$ |
| $\tilde{D}$ | $\alpha D + (1 - \alpha) \frac{\Gamma}{d_1} I$ (In theorems, $\alpha = \frac{1}{2}$) |
| $\tilde{E}$ | $\alpha E + (1 - \alpha) \frac{\Gamma}{d_2} I$ |
| $\check{D}$ | $\alpha \widehat{D} + (1 - \alpha) \frac{\Gamma}{d_1} I$ |
| $\check{E}$ | $\alpha \widehat{E} + (1 - \alpha) \frac{\Gamma}{d_2} I$ |

| | |
|---|---|
| $\widetilde{X}$ (resp. $\widetilde{Y}$) | $XP^{-1}\widetilde{D}^{-\frac{1}{2}}$ (resp. $YQ^{-1}\widetilde{E}^{-\frac{1}{2}}$) |
| $Y'$ (resp. $Y'$) | $XP^{-1}D^{-\frac{1}{2}}$ (resp. $YQ^{-1}E^{-\frac{1}{2}}$) |
| $\widehat{X}$ (resp. $\widehat{Y}$) | $X\widehat{P}^{-1}\widehat{D}^{-\frac{1}{2}}$ (resp. $Y\widehat{Q}^{-1}\widehat{E}^{-\frac{1}{2}}$) |
| $\check{X}$ (resp. $\check{Y}$) | $X\widehat{P}^{-1}\check{D}^{-\frac{1}{2}}$ (resp. $Y\widehat{Q}^{-1}\check{E}^{-\frac{1}{2}}$) |
| $M'$ | $D^{\frac{1}{2}}PMQ^{-1}E^{\frac{1}{2}}$ |
| $\widehat{M}$ | $\widehat{D}^{\frac{1}{2}}\widehat{P}M\widehat{Q}^{-1}\widehat{E}^{\frac{1}{2}}$ |
| $\widetilde{M}$ | $\widetilde{D}^{\frac{1}{2}}PMQ^{-1}\widetilde{E}^{\frac{1}{2}}$ |
| $\check{\widetilde{M}}$ | $\check{D}^{\frac{1}{2}}\widehat{P}M\widehat{Q}^{-1}\check{E}^{\frac{1}{2}}$ |
| $\sigma^1 \in \mathbb{R}^{d_1}$ (resp. $\sigma^2 \in \mathbb{R}^{d_1}$) | singular values of $X$ (resp. $Y$) wrt $\langle \cdot, \cdot \rangle_l$ (resp. $\langle \cdot, \cdot \rangle_r$) |
| equivalently: | $\sigma_u^1 = \sqrt{D_{u,u}}$ ($\sigma_v^2 = \sqrt{D_{v,v}}$) for all $u \leqslant d_1$ (resp. $v \leqslant d_2$) |
| $\sigma_*^1$ (resp. $\sigma_*^2$) | $\max(\sigma^1)$ (resp. $\max(\sigma^2)$) |
| $c_U(i)$ (resp. $c_I(j)$) | community to which user $i$ (resp. item $j$) belongs |
| $\check{D}_I$ (resp. $\check{E}_I$) | same as $\check{D}$ (resp. $\check{E}$) |
| | (with identity side info) |
| Hence: $[\check{D}_I]_{i,i} =$ | $\alpha[\sum_{j=1}^n \frac{h_{i,j}}{N}] + (1-\alpha)\frac{1}{d_1}$ |
| and: $[\check{E}_I]_{j,j} =$ | $\alpha[\sum_{i=1}^m \frac{h_{i,j}}{N}] + (1-\alpha)\frac{1}{d_2}$ |
| $\widetilde{\mathcal{F}}_r$ | $\left\{XMY^\top : \|\widetilde{M}\|_* \leqslant \sqrt{r}\Gamma\right\}$ |
| $\check{\mathcal{F}}_r$ | $\left\{XMY^\top : \|\check{\widetilde{M}}\|_* \leqslant \sqrt{r}\widehat{\Gamma}\right\}$ |
| $\check{Z}_*$ | $\arg\min_{Z \in \check{\mathcal{F}}_r} \mathbb{E}l(Z_\xi, G_\xi + \zeta_\xi)$ |
| $\check{Z}_S$ | $\arg\min(\check{l}_S(Z) : Z \in \check{\mathcal{F}}_r)$ |
| $\widetilde{Z}_*$ | $\arg\min_{Z \in \widetilde{\mathcal{F}}_r} \mathbb{E}l(Z_\xi, G_\xi + \zeta_\xi)$ |
| $\widetilde{Z}_S$ | $\arg\min_{Z \in \widetilde{\mathcal{F}}_r} \mathbb{E}l_S(Z)$ |
| If $G \in \widetilde{F}_r$ | $G = \widetilde{Z}_*$ |
| If $G \in \check{F}_r$ | $G = \check{Z}_*$ |
| $\mathcal{E}$ | $l(G) = \mathbb{E}_{\xi \sim p}l((XM_SY^\top)_\xi, G_\xi + \zeta_\xi)$ |
| $\widetilde{\mathcal{G}}_{r_1,r_2}$ | $\left\{XMY^\top + Z \quad \text{s.t.} \right.$ |
| | $\left. \|\widetilde{D}^{\frac{1}{2}}PMQ^{-1}\widetilde{E}^{\frac{1}{2}}\|_* \leqslant \Gamma\sqrt{r_1} \wedge \|\check{D}_I^{\frac{1}{2}}Z\check{E}_I^{\frac{1}{2}}\|_* \leqslant \sqrt{r_2}\right\}$ |