# OpenReview forum: "Fine-grained Generalization Analysis of Inductive Matrix Completion"
_NeurIPS.cc/2021/Conference — NeurIPS 2021 Poster_

### Official Review · Reviewer_N3AC · 2021-07-07

**Rating:** 6
**Confidence:** 1

**Summary:**

The paper improves convergence rates for Inductive Matrix completion in the distribution-free setting. Authors introduced a new version of the IMC objective and experimentally demonstrate that it may performs better than the standard one.

**Limitations And Societal Impact:**

Yes

**Main Review:**

The paper is well written and many explanations are displayed to readers that are not familiar with the domain. The paper is outside of my area of expertise and I cannot speak confidently about its correctness.

The empirical section seems to be light.


**Time Spent Reviewing:**

4

---

> ### Author Response · Authors · 2021-08-10
> **Thank you for your positive review.**
>
>
> Thank you for your positive review. One of our main contributions is to extend the concept of the "weighted trace norm" to the case where side information is available. The idea of the weighted trace norm is to force the algorithm to focus especially on the rows and columns with more observations to prevent it from being cheated by rare and aberrant observations. In our inductive situation, the *orientations of the directions* which are more or less strongly regularized depend on the (empirical) sampling distribution, adding significant difficulty.
>
> The work is mostly theoretical and fills an important gap in the learning theory of IMC. The main aim of the inductive weighted trace norm regulariser we introduce is to prove the theoretical fact that the non uniformity of the sampling distribution can be corrected via a reweighting procedure which then reduces the sample complexity to the same rates as those we would obtain for a uniform distribution. Our experiments successfully demonstrate
> that our reweighting strategy significantly improves performance compared to the standard IMC objective. There are also plenty of additional experimental results reported in Appendix G2, and we stand ready to run any extra experiments you would like to ask us to include in the revised version.

---

> > ### Comment · Reviewer_N3AC · 2021-09-10
> > **Response**
> >
> > Thank you for these clarifications.

---

### Official Review · Reviewer_7dax · 2021-07-16

**Rating:** 7
**Confidence:** 2

**Summary:**

This paper studies generalisation bounds of Inductive Matrix Completion (IMC), a variant of the classic matrix completion (MC) problem. The main contribution of the paper is improved bounds in the distribution-free setting. Moreover, the authors introduce the (smoothed) adjusted trace-norm minimization strategy, for which they show non-trivial guarantees under arbitrary sampling. These two results generalize similar results previously proved for standard matrix completion to IMC.

**Limitations And Societal Impact:**

Yes

**Main Review:**

Strength:
1. The problem studied is fundamental and well-motivated.
2. Interesting theoretical results are proved, and the proof techniques seems quite non-trivial.
3. Empirical results show that the proposed methods outperform previous IMC on various synthetic datasets and real problems.

Weakness:
The notation and technical proofs are quite dense, and the theoretical results are not as clean as those for standard matrix completion/sensing problems. Thus I think potential audience of this paper might be limited.

I am not familiar with the literature of generalization bounds of IMC, so it’s difficult for me to give a precise evaluation on the technical novelty and depth of the proofs.

**Time Spent Reviewing:**

3

---

> ### Author Response · Authors · 2021-08-10
> **Thank you for your time and positive review.**
>
>
> Thank you for your careful and appreciative review.
>
> Regarding the "denseness"  of the proofs and the difficulty to assess their novelty: We agree that (similarly to the relevant literature) the proofs are relatively dense, and this is a technically difficult topic. To improve the readability, we have already sprinkled the supplementary with thorough descriptions of the intuition and motivations behind some of the proofs, but we promise to improve this aspect further in the revision.  Although for many results there is a parallel with the results of [1], the adaptation of the proof techniques is highly non-trivial because in our situation, the  *orientation* captured by the transformations $P, \widehat{P}, Q, \widehat{Q}$ actually changes depending on the data, whereas only a fixed set of coefficients varies in the existing literature. As a representative example, Lemma E1 is the equivalent of Lemma 2 in page 8 of the supplementary in [1] (whose proof is far, far shorter).  We have tried to explain this in the main paper as well, but we remain open to adding further explanations in whichever direction you suggest.

---

### Official Review · Reviewer_XdTE · 2021-07-16

**Rating:** 6
**Confidence:** 3

**Summary:**

This paper analyzes inductive matrix completion theoretically and conducts numerical experiments to demonstrate the results.

**Limitations And Societal Impact:**

This paper focuses on theoretical analysis. It's expected that the paper does not have direct negative societal impact.

**Main Review:**

1. In the first paragraph of Section 3, it says "We have one fixed loss function $l$ used through the paper which is both Lipschitz with constant $\ell$ and bounded by $b$". This choice of loss function $l$ seems quite unusual since the commonly used squared loss function does not satisfy this condition clearly. What's more, in Section 4, it says square loss is adopted in all experiments. But square loss does not fit in the theoretical results proved in this paper. There seems no explanations about this in the paper.

2. Regarding the choice of loss function $l$, can you give some examples of practical and popular choices?

3. There are some typos. For example, on page 2 below equation (3), since $\boldsymbol{x}$ is defined to be the maximum norm of row of $X$, the definition of $\boldsymbol{x}$ should be $\boldsymbol{x}:=\Vert X\Vert_{2,\infty}$ rather than $\boldsymbol{x}:=\Vert X^{\top}\Vert_{2,\infty}$.


**Time Spent Reviewing:**

3

---

> ### Author Response · Authors · 2021-08-10
> **Many thanks for your comments. We address the loss function issue in detail below.**
>
>
> Many thanks for your careful review.
>
> **Questions 1,2: boundedness of the loss function, Lipschitzness, what about the square loss...?**
>
>
> Thank you for your comment.
>
>
> A. As also explained in our answer to reviewer FcbJ, note that in many applications of matrix completion (most notably in Recommender Systems), the entries are known to be constrained to lie in a given interval (for instance, [1,5]). This has two consequences: (1) Since the *test  points* are known to lie in the given interval, if the recommender system gives off a completely aberrant rating (for instance, 100), it is not any worse than it would be to give a wrong (but within-range) rating, since the answer is clearly out of scale. (2) Since all the *observed* ratings are also within the given interval, optimizing the square loss on the training set is exactly the same as optimizing a truncated square loss, e.g. $l(x,y)=\min(|x-y|^2,25)$ (note this loss function is $10$-Lipschitz and bounded by 25).
> In conclusion, in such typical situations, one can use the square loss whilst still enjoying both Boundedness and Lipschitzness. Other popular losses include the absolute value, which is also bounded assuming the entries are known to be bounded.
>
> B. Our approach (assuming in theoretical results that the loss is bounded and Lipschitz) is in line with most of the related literature. The following works (as cited in our answer to reviewer FcbJ) also take this approach:
>
>
>
> [1] Learning with the Weighted Trace-norm under Arbitrary Sampling Distributions, Rina Foygel, Ruslan Salakhutdinov, Ohad Shamir and Nathan Srebro, **NeurIPS 2011**.
>
> (Theorem 2 page 4, Theorem 5 page 7, etc.)
>
> [2] Matrix Completion with the Trace Norm: Learning, Bounding, and Transducing, Ohad Shamir, Shai Shalev-Shwartz, **JMLR 2014**.
>
> （Theorem 2 on page 3405, Theorem 3 on page 3406)
>
> [3] Matrix Completion with Noisy Side Information, Kai-Yang Chiang, Cho-Jui Hsieh and Inderjit S. Dhillon, **NeurIPS 2015**.
>
> (Theorem 1 on page 5)
>
> [4] Using Side Information to Reliably Learn Low-Rank Matrices from Missing and Corrupted Observations,  Kai-Yang Chiang, Cho-Jui Hsieh and Inderjit S. Dhillon, **JMLR 2018**.
>
> (Theorem 5 on page 13 etc. )
>
> **Typos: definition of $\mathbf{x},\mathbf{y}$**
>
> Thank you for your help in improving our manuscript. Of course, we agree to carefully reread the whole of our work and fix typos in the revised version.
>
> Regarding the definition of $\mathbf{x}, \mathbf{y}$, we tend to use the notation |A|$\_{2,\infty}$ for the maximum $L^2$ norm of any of the columns of $A$ or more generally, |A|$\_{p,q}$ is the $L^q$ norm of the vector of the $L^p$ norms of the columns of $A$. We have observed that both this notation and the reverse notation are used in the literature.
>
>
> Our notation follows from the same notation system as in this paper:
>
> [5] Spectrally-normalized margin bounds for neural networks. Peter Bartlett, Dylan J. Foster, Matus Telgarsky, NeurIPS 2017.
>
>
> We also note the reverse notations are used in the following paper:
>
> [6] Matrix Completion with Model-free Weighting. Jiayi Wang, Raymond K. W. Wong, Xiaojun Mao, Kwun Chuen Gary Chan. ICML 2021

---

> > ### Comment · Reviewer_XdTE · 2021-08-26
> > **Discussion**
> >
> > Thank you for detailed response to my reviews. I've updated my score accordingly.

---

### Official Review · Reviewer_FcbJ · 2021-08-03

**Rating:** 7
**Confidence:** 3

**Summary:**

This paper develops theoretical results for IMC in a uniform and distribution free setting. It also develops an algorithm for the adjusted regularisation in (11).

**Limitations And Societal Impact:**

Some future work is mentioned in the conclusion.

**Main Review:**

I think the paper is solid. Below I have a few questions and would like to hear the authors' responses.
1. From line 43 to 45, they mentioned the sample complexity in a approximate recovery setting. This guarantees has epsilon ^2 dependence. Is all the results mentioned in the paper (Table 1 and 2) has this epsilon dependence?

2. The formulation (1) (2) (4) and (11) are all penalized optimization problem. Yet theorems and propositions in Section 3 are all constraint version, e.g., min sth subject to some constraints. Even though there is a general relation between the two via the lagrangian multiplier, the multiplier will depend on the data in general rather than a fixed choice. Can the author comment on how to choose the parameter lambda in (1)(2)(4) and (11) in order to achieve meaningful theoretical results?

3. Most of the theorem requires the loss to be bounded and Lipschitz. But the loss in (1) (2) and (4) are not lipschitz and bounded if we consider the whole space of the matrices M or Z. With the constraint set in theorems and propositions in Section 3, can the author comment on how large the bounded parameter b of the loss l and the lipschitz constant of the loss l will be if we use the square loss? Will the results in section 3 become very bad if we use square loss?

4. What is the weighted version in Table 2? Is it the setting in Section 3.4 and 3.5? Can I understand it as the case that this is also the distribution free setting but we use a different optimization problem to recover the matrix?

5.In line 54 what is O(r^2)? or you mean rd^2?

6. In the tables, I think it worth mentioning the following results
Chen, Yudong. "Incoherence-optimal matrix completion." IEEE Transactions on Information Theory 61, no. 5 (2015): 2909-2923.
This paper discusses the MC and IMC setting with a better incoherence dependence.

Ding, Lijun, and Yudong Chen. "Leave-one-out approach for matrix completion: Primal and dual analysis." IEEE Transactions on Information Theory 66, no. 11 (2020): 7274-7301.
This one has the best sample complexity O(drlog(d)log(r)) for exact matrix completion, which is better than the results in  Table 1. So this is the state-of-art.

7. Why r = M/(Sd1d2) is considered as the relaxation of rank in line 165?  I think r = M/S in general is the rank. And in line 168, why |M|_*/S<=sqrt(d1d2*r)? You defined rho in line 167 and it is not used.

8. Perhaps in Line 244, you can mention q and kappa are defined in (9)

**Time Spent Reviewing:**

5

---

> ### Author Response · Authors · 2021-08-10
> **Thank you for your positive review and relevant comments. We address the issue of the constrained optimization and bounded loss function in detail below.**
>
> Many thanks for your careful and positive review, which will surely improve the quality and readability of our work.
>
> **$\epsilon^2$ dependence in approximate recovery bounds in Tables 1 and 2.**
>
> Yes, we confirm that all bounds for approximate recovery in both Tables 1 and 2 have an implicit factor of $1/\epsilon^2$. This is true for both our own results and the state-of-the-art results we mention. Indeed, approximate recovery results typically consist in a bound on the expected loss of the form $\sqrt{\frac{f(r,d,m)}{N}},$ where $N$ is the number of samples. Translating this into a sample complexity bound yields a factor of $1/\epsilon^2$ where $\epsilon$ is the tolerance.
>
>
> We thank you again for your comment and we promise to clarify this aspect better in the revised version.
>
>
>
> **"Theoretical results use explicit constraints while the practical algorithms use a (soft) regulariser"**
>
> Thank you for your highly relevant comment and observations.
>
> The parameters $\lambda,\lambda_1,\lambda_2$ in the practical algorithms (1), (2), (4), (11), just as the parameter $r$ in Prop. 3.3 and Theorem 3.2, must be tuned with cross validation. From the statistical point of view, the cross validation requirements are similar in both cases.
>
> Next, note the *generalization gap* can be computed from a trained model with our simplest results based on Rademacher complexity. For instance, Propositions B1, B2 and C1 can easily, via a union bound, be made to hold (up to constants and logarithmic terms) for *any* values of $\mathcal{M},r$.
>
>
> You make a good observation that a priori, the correct regularisation parameters will depend not just on the ground truth but also on the *data*. However, (as long as the loss term is normalized by $1/N$), suitable assumptions ensure the choice of the optimal parameter doesn't depend too much on the data.
>
>
> Consider first the simple case of an application of Proposition 3.1 in the *noiseless case*. If we were to rely on the (penalized) optimization problem (2), the optimal value of $\lambda$ would be $0$, and solving  the problem boils down to minimizing the nuclear norm $n_*(M)$ subject to a perfect match of the observed entries. Adopting analogous notation to Proposition 3.1, let
> $M_S$ = argmin ($n_*(M)$: $\sum_{\xi\in\Omega}$  l(
> ($XMY^{\top}$)$\_\xi$, G$ \_\xi$)=0). Since the ground truth $M_*$ satisfies $\sum_{\xi\in\Omega}$ l([$XMY^{\top}$]$\_{\xi}$,G$\_{\xi}$)=0, we must have that $n_*(M\_S) \leq$ $n_*(M_*)$, and we can apply Proposition 3.1 with $\mathcal{M}$ = $n_*(M_*)$.
>
>
> Similar arguments can be applied in the presence of (well-behaved) noise: write $\hat{l}(Z)$ for the empirical loss evaluated at a solution $Z$ and $\mathcal{R}(Z)$ for the regularizer considered. Write $Z_*$ for the ground truth and $Z_S$ for the solution to the penalized optimization problem (e.g. equation (1),(2),(4),(11) from the paper). If the noise distribution is well-behaved (e.g., subgaussian), we have that for (absolutely) large enough $N$, $\hat{l}(Z_*)$ concentrates to the Bayes risk  $l(Z_*)$. Setting $\lambda$ $\sim$  $\mathcal{E}/\mathcal{R}(Z_*)$, the objective function (of the penalized formulation), evaluated at the ground truth $Z_*$, is close to $l(Z_*)+[l(Z_*)/\mathcal{R}(Z_*)]\mathcal{R}(Z_*)=2l(Z_*)$. This implies that the solution $Z_S$ satisfies $\lambda \mathcal{R}(Z_S)\lesssim 2l(Z_*)$, which in turn shows that $\mathcal{R}(Z_S)\lesssim C\mathcal{R}(Z_*)$ for a constant $C$, which allows us to apply results of the paper pertaining to the explicitly restricted function class.
>
>
>
>
>
> **Most theorems require the loss function to be Lipschitz and bounded, what about the square loss?**
>
>
>
>
> Thank you for this relevant observation.
>
>
> In most natural applications of matrix completion (including Recommender Systems), the entries are (known to be) bounded by an absolute constant (for instance, ratings are often given on a scale from 1 to 5). Thus, it makes sense to use a truncated version of the square loss: $l(x,y)=\min(|x-y|^2,B^2)$ where $B$ is the known absolute bound on the entries. Our results then apply in this case (the Lipschitz constant is bounded by $2B$ and the loss function itself is bounded by $B^2$).
> Note that optimization with this loss is identical to optimization with the square loss (since the observations satisfy the constraint).
> Note also that even in the context of *exact recovery*, some (weaker) form of boundedness assumption is still required: the entries of the ground truth matrix (at least) are indirectly assumed bounded via incoherence assumptions.
>
>
>
>
> Finally, we observe that **our formulation** including a bounded loss function and  the optimization problem in terms of explicit norm constraints is **a standard in the theoretical literature**. In particular, **all of the works below**, which are amongst the most influential and core works in the theory of approximate recovery of MC and IMC **take a similar approach**.
>
>
>
>
> **[1] Learning with the Weighted Trace-norm under Arbitrary Sampling Distributions, Rina Foygel, Ruslan Salakhutdinov, Ohad Shamir and Nathan Srebro, NeurIPS 2011.**
>
> (Theorem 2 page 4, Theorem 5 page 7, etc.)
>
> **[2] Matrix Completion with the Trace Norm: Learning, Bounding, and Transducing, Ohad Shamir, Shai Shalev-Shwartz, JMLR 2014.**
>
> （Theorem 2 on page 3405, Theorem 3 on page 3406)
>
> **[3] Matrix Completion with Noisy Side Information, Kai-Yang Chiang, Cho-Jui Hsieh and Inderjit S. Dhillon, NeurIPS 2015.**
>
> (Theorem 1 on page 5)
>
> **[4] Using Side Information to Reliably Learn Low-Rank Matrices from Missing and Corrupted Observations,  Kai-Yang Chiang, Cho-Jui Hsieh and Inderjit S. Dhillon, JMLR 2018.**
>
> (Theorem 5 on page 13 etc. )
>
> We agree to explain the implications of the boundedness and Lipschitzness assumptions in more detail in the revision, as per your suggestions. Thank you for improving the quality of our work.
>
>
> **What is the weighted version in tables...?**
> Yes, we confirm the weighted version column refers to the distribution-free case with the ability to alter the regularizer via a weighting strategy such as ours or its equivalent in standard MC. We will make this clearer in the revision, thank you.
>
> **Line 54...**
>
> Yes, we mean $O(rd^2)$, thank you!
>
>
> **Citation missing: Chen, Yudong "Incoherence-optimal matrix completion"**
>
> We warmly thank you for the reference, which seems very interesting technically and which we will certainly cite and include in the table in the revision.
>
> After a preliminary look at it, we agree that it should be included in the top left of Table 1 since it has a rate of $O(nrlog(n)log(r))$ for standard matrix completion with the trace norm, which is better (by a logarithmic term) than the rate in the table (currently $O(nrlog^2(n))$). Of course, its better dependence on the incoherence coefficients is also worth mentioning, even though this aspect of the dependence is not taken into account in our tables.
>
> We were unable to find results for inductive matrix completion in that reference, so it is not clear whether it should be included in Table 2.
>
> Thank you again for this highly relevant comment.
>
>
>
>
>
> **Why r = M/(Sd1d2) is considered as the relaxation of rank in line 165?... You defined rho in line 167 and it is not used.**
>  We agree to reword the informal description of the behavior of the result (lines 165 to 171) , as $\mathcal{M}/\mathcal{S}$ is indeed a soft analogue of the rank, as you point out.
> We will reformulate our descriptions to take these facts into account. Thank you again for your helpful comments.
>
>
>
>
> **Perhaps in Line 244, you can mention q and kappa are defined in (9)**
>
>
> Thank you, we promise to do so in the revision.

---

> > ### Comment · Reviewer_FcbJ · 2021-08-31
> > **Two small questions**
> >
> > Hi Authors,
> >
> > Thanks for addressing my question. I have two small questions to ask.
> >
> > 1. What is $\mathcal{E}$ when you say how to set $\lambda$.
> >
> > 2. For the loss, if I still use the square loss but in the constraint set, I add a constraint so that the maximum of entries of the matrices is bounded. Will this modification having a similar effect as the truncated loss you proposed?

---

> > > ### Author Response · Authors · 2021-08-31
> > > **Answer to your questions.**
> > >
> > > Thank you for your further comments.
> > >
> > >
> > > **Answer to question 1:**
> > >
> > >  $\mathcal{E}$ refers to the Bayes risk, i.e., the expected value of the loss evaluated at the ground truth. Thus, if there is no noise, then $\mathcal{E}=0$. For the square loss it would be the variance of the noise.
> > >
> > > In summary, in the main parts of the paper where we consider a constrained optimization problem, the constraint must be chosen sufficiently loose to ensure that the ground truth satisfies the constraint, but not so large as to make the bound become too loose. Similarly, in our response above, where we consider regularised versions of our algorithms, $\lambda$ must be set within a constant factor of the ratio between $\mathcal{E}$ and the regularizer evaluated at the ground truth. In both cases, the choice must be done via *cross-validation* but it is only required to approximate the correct parameter up to a multiplicative constant.
> > >
> > > All these concerns are similar to the existing literature (cf. [1,2,3,4]).
> > >
> > >
> > >
> > >
> > >
> > >
> > >
> > > **Answer to question 2:**
> > >
> > > Yes, if we introduce an entry-wise boundedness constraint in the constraint set, this will allow us to bound the *Rademacher complexity* of the relevant function class exactly as if the loss itself were bounded.
> > > This is also discussed in some of the relevant literature (c.f. [2] below, Theorem 3).
> > >
> > > Although it is commonly not discussed explicitly in the literature, it is also true that if the noise is also bounded, we can also immediately translate the bounds into *excess risk bounds*.
> > > If the noise is not bounded but is (for instance) $\nu$-subgaussian, then this can still be done at the cost of a few more trivial manipulations and an extra log factor.
> > >
> > > We agree to add a discussion of those aspects in the revised version.
> > >
> > >
> > >
> > >
> > > **References:**
> > >
> > > [1] Learning with the Weighted Trace-norm under Arbitrary Sampling Distributions, Rina Foygel, Ruslan Salakhutdinov, Ohad Shamir and Nathan Srebro, NeurIPS 2011.
> > >
> > >
> > > [2] Matrix Completion with the Trace Norm: Learning, Bounding, and Transducing, Ohad Shamir, Shai Shalev-Shwartz, JMLR 2014.
> > >
> > >
> > > [3] Matrix Completion with Noisy Side Information, Kai-Yang Chiang, Cho-Jui Hsieh and Inderjit S. Dhillon, NeurIPS 2015.
> > >
> > >
> > > [4] Using Side Information to Reliably Learn Low-Rank Matrices from Missing and Corrupted Observations, Kai-Yang Chiang, Cho-Jui Hsieh and Inderjit S. Dhillon, JMLR 2018.

---

> > > > ### Comment · Reviewer_FcbJ · 2021-09-10
> > > > **Bounded loss or entrywise norm bound**
> > > >
> > > > Hi authors,
> > > >
> > > > I now understand the assumption of bounded and lipschitz loss probably is adopted
> > > > in many literatures when dealing with matrix with or without side information.
> > > >
> > > > However, thinking back, this assumption makes the optimization problem computationally intractable (though
> > > > theoretically tractable). The truncated square loss you mentioned is not convex and hence the minimization problem
> > > > can not be guaranteed to solved to near optimality.
> > > >
> > > > Thus, I would like really like to see a guarantee, at least some discussion, for the square loss with a constraint on the entrywise norm.

---

> > > > > ### Author Response · Authors · 2021-09-11
> > > > > **Minimizing the square loss and then truncating the entries yields a solution with similar guarantees as long as the ground truth and noise are bounded.**
> > > > >
> > > > >  Thank you for your further comment.
> > > > >
> > > > > **Summary**: Truncating the entries results in a manageable Rademacher complexity. If one assumes that both the ground truth and the noise are bounded by a constant $C$, one can use this to prove that the following strategy yields a solution with similar guarantees to the ones we proved. *It is not necessary to optimize the truncated loss*.
> > > > >
> > > > >
> > > > >
> > > > >
> > > > > Step 1: Minimize the square loss in the given function class (no max entry constraint, no truncation of the loss).
> > > > >
> > > > >
> > > > > Step 2: Truncate the entries by feeding them into a function like $\phi(x)=\text{sign}(x)\min(|x|,2C)$.
> > > > >
> > > > >
> > > > > (Since the ground truth and noise are bounded, the result of the above operations will have even smaller population average square loss than if we omitted step 2.) The constants showing up in the bounds are those corresponding to the square loss truncated at $4C$, but the theorem applies to the square loss. $C$ can be cross validated, or be known (e.g. most recommender systems settings, where $C=4$). The entire Rademacher complexity argument is exactly the same. We will include a full discussion in the final version if accepted.
> > > > >
> > > > >
> > > > >
> > > > > **A quick note on the noiseless case**:  Note that as hinted at before the *noiseless* case is easier in every aspect. Taking Prop 3.3 as an example, in that case, if we define the solution $Z_1$ as the minimizer of the regulariser subject to a perfect match with the observed entries (a fully tractable optimization problem, which doesn't involve any loss function at all), then for *any* given threshold $C$, the (population) expected truncated square loss $l_C(Z_1)$ will with high probability be bounded by the right hand side of the the equation in Prop 3.3 with $b=C^2$ and $\ell=2C$.
> > > > >
> > > > >
> > > > >  The bounded noise case is similar but requires a few more simple manipulations.
> > > > >
> > > > >
> > > > >
> > > > >  **Details about the case of bounded noise:**
> > > > >
> > > > >
> > > > >
> > > > >  Assume that the entries of the ground truth $G$ are bounded by $C$ and the noise is bounded also by $C$.
> > > > >
> > > > > Write $l_2$ for the square loss and $l_{4C}$ for its truncated version with threshold $4C$ (thus $\|l_{4C}\|_{\infty}=16C^2$). As usual, we add hats to loss functions to designate their empirical versions.
> > > > >
> > > > > Let $\phi(x)=\text{sign}(x)\min(|x|,2C)$.  We can consider  the function class $\[\phi(XMY^\top) : n(M) \leq \mathcal{M}   \]$ where  $\phi$ is applied elementwise and $n(M)$ denotes the relevant norm of $M$ depending on which of our theorems one applies (in Proposition 3.3 for instance, $n(M)=\|\tilde{M}\|_*$).
> > > > > *All the arguments on the Rademacher complexity carry over, and all of our bounds on Rademacher complexity hold for this function class*.
> > > > >
> > > > >   Furthermore, by the bounds we have for the entries, for any $A\in F_{\phi}$ we have $l_2(A)=l_{4C}(A)$ and
> > > > >
> > > > > $\hat l_2(A)=\hat l_{4C}(A)$
> > > > >
> > > > >
> > > > > Thus with high probability $\geq 1-\delta$, for any element $Z=\phi(XMY^\top)$ in the function class $F_{\phi}$ the generalisation gap
> > > > > $l_{4C}(Z)-\hat l_{4C}(Z)$
> > > > >  satisfies
> > > > >  $|l_{4C}(Z)-\hat l_{4C}(Z)|\leq B_\delta =B $ where "B" is the bound from our paper (e.g. the right hand side of the equation in proposition 3.3) with $b=16C^2$ and $\ell=8C$.
> > > > >
> > > > >
> > > > > Now, let $Z_1=XM_1Y^\top$ be the minimiser of the **square loss**        $l_2$ within the function class $F=\[XMY^\top : n(M) \leq \mathcal{M}   \]$ (no $\phi$ is  applied here). I.e. $Z_1\in\text{argmin}(\hat l_2(Z): Z\in F)$.   We certainly have
> > > > > $\hat l_2(Z_1)\leq \hat l_2(G) =\hat l_{4C}(G) $
> > > > >
> > > > > Note also that since the maximum possible observation is $\leq 2C$, we have $\hat l_2(\phi(Z_1))\leq \hat l_2(Z_1)$. Thus $\hat l_2(\phi(Z_1))\leq \hat l_{4C}(G) $.
> > > > >
> > > > > Thus we have (with probability $\geq 1-\delta$):
> > > > >
> > > > > $l_2(\phi(Z_1))=l_{4C}(\phi(Z_1))\leq \hat l_{4C}(\phi(Z_1))+B_\delta =$
> > > > > $\hat l_2(\phi(Z_1))+B_\delta\leq \hat l_{4C}(G)+B_\delta\leq l_{4C}(G)+ 2B_{\delta} =l_2(G)+2B_{\delta}$.
> > > > >
> > > > >
> > > > >
> > > > > As mentioned previously, similar results could be achieved for subgaussian noise with a few extra manipulations and log terms. This can also be combined with the idea in our previous comment above to obtain bounds for the solution to the lagrangian formulation (with the square loss) as long as the noise is bounded.
> > > > >
> > > > >
> > > > >
> > > > >
> > > > >
> > > > >
> > > > > **Concluding remarks**: In the main paper, we followed the established optimization setting from references [1,2,3,4] with regards to the loss function, opting to focus on the aspects which are specific to our own work. However, we enjoyed further discussing this more general question of the loss function and promise to include a full discussion in the final version if accepted. Thank you.

---

### Author Response · Authors · 2021-08-10
**Summary of the main points of the rebuttal.**

We thank the reviewers for their careful and appreciative reviews.

The main concerns raised were (1) the modified form of the optimization problem (which uses explicit bounds on the relevant norms rather than the penalized form used in experiments) and (2) the assumption that the loss function is bounded and Lipschitz when proving the relevant theorems.

As we explain in more detail in our responses to reviewers FcbJ and XdTE, this approach is standard in the literature: indeed, (1) there is a close relationship between the constrained and penalized versions of the algorithm, and (2) it is reasonable to assume that the entries/ratings are bounded, in which case the square loss (restricted to the relevant range) will be bounded and Lipschitz. We hope that we have clarified all the reviewers' doubts and look forward to further feedback.

---

### Decision · Program_Chairs · 2021-09-27

**Decision:**

Accept (Poster)

**Comment:**

This paper studies inductive matrix completion with side information. They provide distribution-free bounds that improve upon previous work. New guarantees are also obtained for weighted nuclear norm minimization under arbitrary sampling. The reviewers agree that this paper makes solid contributions and involves interesting techniques. On the other hand, the reviewers also find that the authors should better explain the Lipschitz/boundedness assumptions as well as the novelties for the weighted trace norm. and the clarity of the presentation can be improved.